# Universal Inverse Distillation for Matching Models with Real-Data Supervision (No GANs)

**Nikita Kornilov**[*]
Applied AI Institute, Moscow, Russia
MIRAI, Moscow, Russia
BRAIn Lab, Moscow, Russia
`jhomanik14@gmail.com`

**David Li**[*]
AI Foundation lab, Moscow, Russia
MBZUAI, Abu Dhabi, UAE
`David.Li@mbzuai.ac.ae`

**Tikhon Mavrin**[*]
Applied AI Institute, Moscow, Russia
`tixonmavrin@gmail.com`

**Aleksei Leonov**
AI Foundation lab, Moscow, Russia
MIRAI, Moscow, Russia

**Nikita Gushchin**
Applied AI Institute, Moscow, Russia
AXXX, Moscow, Russia

**Evgeny Burnaev**
Applied AI Institute, Moscow, Russia
AXXX, Moscow, Russia

**Iaroslav Koshelev**
AI Foundation lab, Moscow, Russia

**Alexander Korotin**
Applied AI Institute, Moscow, Russia
AXXX, Moscow, Russia
`iamalexkorotin@gmail.com`

## Abstract

While achieving exceptional generative quality, modern diffusion, flow, and other matching models suffer from slow inference, as they require many steps of iterative generation. Recent distillation methods address this problem by training efficient one-step generators under the guidance of a pre-trained teacher model. However, these methods are often constrained to only one specific framework, e.g., only to diffusion or only to flow models. Furthermore, these methods are originally data-free, and to benefit from the usage of real data, it is required to use an additional complex adversarial training with an extra discriminator model. In this paper, we present **RealUID**, a universal distillation framework for all matching models that seamlessly incorporates real data into the distillation procedure without GANs. Our **RealUID** approach offers a simple theoretical foundation that covers previous distillation methods for Flow Matching and Diffusion models, and can be also extended to their modifications, such as Bridge Matching and Stochastic Interpolants. The code can be found in `https://github.com/David-cripto/RealUID`.

## 1 Introduction

In generative modeling, the goal is to learn to sample from complex data distributions (e.g., images), and two powerful paradigms for it are the **diffusion models** (DM) and the **flow matching** (FM) models. While they share common principles and are even equivalent under certain conditions (Holderrieth et al., 2024; Gao et al., 2025), they are typically studied separately. Diffusion models (Sohl-Dickstein et al., 2015; Ho et al., 2020; Song et al., 2021) transform data into noise through a forward process and then learn a reverse-time stochastic differential equation (SDE) to recover the data distribution. Training minimizes score-matching objectives, yielding unbiased estimates of intermediate scores. Sampling requires simulating the reverse dynamics, which is computationally heavy but delivers high-quality and diverse results. Flow Matching (Lipman et al., 2023; Liu, 2022)

---
[*]Equal contribution

instead interpolates between source and target distributions by learning the vector field of an ordinary differential equation (ODE). The field is estimated through unbiased conditional objectives, but the resulting ODE often has curved trajectories, making sampling costly due to expensive integration. Beyond these, **Bridge Matching** (Peluchetti, 2023; Liu et al., 2022) and **Stochastic Interpolants** (Albergo et al., 2023) generalize the framework and naturally support *data couplings*, which are crucial for data-to-data translation. Since all of the above optimize *conditional matching* objectives to recover an ODE/SDE for generation, we refer to them collectively as *matching models*.

Despite their success, matching models share a major drawback: sampling is slow, as generation requires integrating many steps of an SDE or ODE. To address this problem, a range of distillation techniques have been proposed to compress multi-step dynamics into efficient one-step or few-step generators. Although matching models follow a similar mathematical framework, many distillation works consider only one particular framework, e.g., only diffusion models (Zhou et al., 2024a;b), Flow Matching (Huang et al., 2024), or Bridge Matching (Gushchin et al., 2025). Furthermore, these distillation methods are data-free by construction and cannot benefit from the utilization of real data without using additional GAN-based losses. *Thus, the following issues remain:*

1. Similar distillation techniques developed separately for similar matching models frameworks.

2. Absence of a natural way to incorporate real data in distillation procedures (without GANs).

**Contributions.** In this paper, we address these issues and present the following **main contributions**:

1. We present the *Universal Inverse Distillation with real data (RealUID)* framework for matching models, including diffusion and flow matching models (§3) as well as Bridge Matching and Stochastic Interpolants (Appendix C). It unifies previously introduced Flow Generator Matching (FGM), Score Identity Distillation (SiD) and Inverse Bridge Matching Distillation (IBMD) methods (§3.2) for flow, score and bridge matching models respectively, provides simple yet rigorous theoretical explanations based on a linearization technique, and reveals the connections between these methods and inverse optimization (§3.3).

2. Our RealUID introduces a novel and natural way to incorporate real data directly into the distillation loss, eliminating the need for extra adversarial losses which require additional discriminator networks used in GANs from the previous works (§3.4).

## 2 BACKGROUNDS ON TRAINING AND DISTILLING MATCHING MODELS

Here, we describe diffusion, flow (§2.1) and other types of matching models (§2.2) along with the distillation methods for them (§2.3). Then, we discuss how GANs can be used to add real data into the distillation procedure (§2.4).

**Preliminaries.** We work on the $D$-dimensional Euclidean space $\mathbb{R}^D$. This space is equipped with the standard scalar product $\langle x, y \rangle = \sum_{d=1}^D x_d y_d$, the $\ell_2$-norm $\|x\| = \sqrt{\langle x, x \rangle}$ and the $\ell_2$-distance $\|x - y\|, \forall x, y \in \mathbb{R}^D$. We consider probability distributions from the set $\mathcal{P}(\mathbb{R}^D)$ of absolutely continuous distributions with finite variance and support on the whole $\mathbb{R}^D$.

### 2.1 DIFFUSION AND FLOW MODELS

**Diffusion models** (Sohl-Dickstein et al., 2015; Ho et al., 2020; Song et al., 2021) consider a forward noising process $p_t$ that gradually transforms clean data $p_0$ into a noise $p_T$ on the time interval $[0, T]$:

$$dx_t = f_t \cdot x_t \cdot dt + g_t \cdot dw_t, \quad x_0 \sim p_0,$$

where $f_t$ and $g_t$ are time-dependent scalars and $w_t$ is a standard Wiener process. This process defines a conditional distributions $p_t(x_t|x_0) = \mathcal{N}(\alpha_t x_0 | \sigma_t^2 \mathbf{I})$, where

$$\alpha_t = \exp\left(\int_0^t f_s \, ds\right), \quad \sigma_t = \left(\int_0^t g_s^2 \exp\left(-2\int_0^s f_u \, du\right) ds\right)^{1/2}.$$

Each conditional distribution admits a conditional score function, describing it:

$$s_t(x_t|x_0) := \nabla_{x_t} \log p_t(x_t|x_0) = -(x_t - \alpha_t x_0)/\sigma_t^2.$$

The reverse dynamics from the noise distribution $p_T$ to the data distribution $p_0$ is provided by the following *reverse-time* SDE with a reverse-time Wiener process $\bar{w}_t$:

$$dx_t = (f_t \cdot x_t - g_t^2 \cdot s_t(x_t))dt + g_t d\bar{w}_t,$$

where $s_t(x_t) = \mathbb{E}_{x_0 \sim p_0(\cdot|x_t)}[s_t(x_t|x_0)]$ is the score function of $p_t(x_t) = \int p(x_t|x_0)p(x_0)dx_0$. This unconditional score function is learned via minimizing the denoising score matching (DSM) loss:

$$\mathcal{L}_{\text{DSM}}(s', p_0) = \mathbb{E}_{t \sim [0,T], x_0 \sim p_0, x_t \sim p_t(\cdot|x_0)}\left[\gamma_t \|s'_t(x_t) - s_t(x_t|x_0)\|_2^2\right], \qquad (1)$$

where $\gamma_t$ are some positive weights. The reverse dynamics admits a probability flow ODE (PF-ODE):

$$dx_t = u_t(x_t)dt, \quad u_t(x_t) := (f_t \cdot x_t - g_t^2 \cdot s_t(x_t)/2),$$

which provides faster inference than the SDE formulation.

**Flow Matching** framework (Lipman et al., 2023; Liu et al., 2023) constructs the flow directly by learning the drift $u_t(x_t)$. Specifically, for each data point $x_0 \sim p_0$, one defines a conditional flow $p_t(x_t|x_0)$ with the corresponding conditional vector field $u_t(x_t|x_0)$ generating it via ODE:

$$dx_t = u_t(x_t|x_0)dt.$$

Then, to construct the flow between the data $p_0$ and noise $p_T$, one needs to compute the unconditional vector field $u_t(x_t) = \mathbb{E}_{x_0 \sim p_0(\cdot|x_t)}[u_t(x_t|x_0)]$ which generates the flow $p_t(x_t) = \int p(x_t|x_0)p(x_0)dx_0$. It can be done by minimizing the following Conditional Flow Matching (CFM) loss:

$$\mathcal{L}_{\text{CFM}}(v, p_0) = \mathbb{E}_{t \sim [0,T], x_0 \sim p_0, x_t \sim p_t(\cdot|x_0)}\left[\|v_t(x_t) - u_t(x_t|x_0)\|_2^2\right]. \qquad (2)$$

In practice, the most popular choice is the Gaussian conditional flows $p_t(x_t|x_0) = \mathcal{N}(\alpha_t x_0, \sigma_t^2 \mathbf{I})$. For this conditional flow samples can be obtained as $x_t = \alpha_t x_0 + \sigma_t \epsilon$, $\epsilon \sim \mathcal{N}(0, \mathbf{I})$ and the conditional drift can be calculated as $u_t(x_t|x_0) = \dot{\alpha}_t x_0 + \dot{\sigma}_t \epsilon$.

We recall data-to-data models working with data couplings, such as **Bridge Matching** and **Stochastic Interpolants**, in Appendices C.1 and C.2, respectively.

## 2.2 Universal loss for matching models

From a mathematical point of view, it was shown in (Holderrieth et al., 2024; Gao et al., 2025) that flow and diffusion models basically share the same loss structure. We recall this structure, but use our own notation. We call diffusion and flow models and their extensions as *matching models*.

A matching model constructs a probability path $p_t$ on the time interval $[0, T]$, transforming the desired data $p_0 \in \mathcal{P}(\mathbb{R}^D)$ to the noise $p_T \in \mathcal{P}(\mathbb{R}^D)$. This path is built as a mixture of simple conditional paths $p_t(\cdot|x_0)$ conditioned on samples $x_0 \sim p_0$, i.e., $p_t(x_t) = \int_{\mathbb{R}^D} p_t(x_t|x_0)p_0(x_0)dx_0, \forall x_t \in \mathbb{R}^D$. The path $p_t$ determines the *function* $f^{p_0} : [0, T] \times \mathbb{R}^D \to \mathbb{R}^D$ which recovers it (e.g., score function or drift). The conditional paths also determine their own simple *conditional functions* $f^{p_0}(\cdot|x_0)$ so that they express $f_t^{p_0}(x_t) = \mathbb{E}_{x_0 \sim p_0(\cdot|x_t)}[f_t^{p_0}(x_t|x_0)]$, where $p_0(\cdot|x_t)$ is a data distribution $p_0$ conditioned on sample $x_t$ at time $t$. Since the function $f^{p_0}$ cannot be computed directly, it is approximated by a *trainable function* $f : [0, T] \times \mathbb{R}^D \to \mathbb{R}^D$ at each time $t$ from $[0, T]$ and point $x_t \sim p_t$ using known analytically conditional functions:

$$\|f_t(x_t) - f_t^{p_0}(x_t)\|^2 = \|f_t(x_t) - \mathbb{E}_{x_0 \sim p_0(\cdot|x_t)}[f_t^{p_0}(x_t|x_0)]\|^2 \propto \mathbb{E}_{x_0 \sim p_0(\cdot|x_t)}[\|f_t(x_t) - f_t^{p_0}(x_t|x_0)\|^2].$$

We also change the sampling order $x_t \sim p_t, x_0 \sim p_0(\cdot|x_t)$ to more natural $x_0 \sim p_0, x_t \sim p_t(\cdot|x_0)$.

**Definition 1.** *We define **Universal Matching (UM)** loss $\mathcal{L}_{UM}(f, p_0)$ that takes trainable function $f$ and distribution $p_0 \in \mathcal{P}(\mathbb{R}^D)$ as arguments and upon minimization over $f$ returns the function $f^{p_0}$:*

$$\mathcal{L}_{UM}(f, p_0) := \mathbb{E}_{t \sim [0,T]}\mathbb{E}_{x_0 \sim p_0, x_t \sim p_t(\cdot|x_0)}[\|f_t(x_t) - f_t^{p_0}(x_t|x_0)\|^2], \quad f^{p_0} := \arg\min{}_f \mathcal{L}_{UM}(f, p_0). \quad (3)$$

*The notation $t \sim [0, T]$ hides time sampling and loss weighting inherent to the given matching model.*

## 2.3 Distillation of matching-based models

To solve the long inference problem of matching models, a line of distillation approaches sharing similar principles was introduced: **Score Identity Distillation** (Zhou et al., 2024b;a, **SiD**), **Flow Generator Matching** (Huang et al., 2024, **FGM**), and **Inverse Bridge Matching Distillation** (Gushchin et al., 2025, **IBMD**), for diffusion, flow, and bridge matching models, respectively.

The **SiD** approach trains a *student generator* $G_\theta : \mathcal{Z} \to \mathbb{R}^D$ (parameterized by $\theta$) that produces a distribution $p_0^\theta$ from a latent distribution $p^\mathcal{Z}$ on $\mathcal{Z}$. This approach minimizes the squared $\ell_2$-distance

between the known *teacher score function* $s^* := \arg\min_{s'} \mathcal{L}_{\text{DSM}}(s', p_0^*)$ on real data $p_0^*$ and the unknown *student score function* $s^\theta$:

$$\min_\theta \mathbb{E}_{t\sim[0,T]}\mathbb{E}_{x_t^\theta \sim p_t^\theta}[\|s_t^\theta(x_t^\theta) - s_t^*(x_t^\theta)\|^2], \quad \text{s.t. } s^\theta = \arg\min_{s'} \mathcal{L}_{\text{DSM}}(s', p_0^\theta), \quad (4)$$

where $p_t^\theta$ is the forward noising process for the generated data $p_0^\theta$. The authors propose the tractable loss with parameter $\alpha_{\text{SiD}}$ to approximate the real gradients of (4) :

$$\begin{aligned}
\mathcal{L}_{\text{SiD}}(\theta) \quad := \quad & \mathbb{E}_{t\sim[0,T]}\mathbb{E}_{z\sim p^\mathcal{Z}, x_0^\theta=G_\theta(z), x_t^\theta\sim p_t^\theta}[-2\omega_t \cdot \alpha_{\text{SiD}}\|s_t^*(x_t^\theta) - s_t^{sg[\theta]}(x_t^\theta)\|^2 \\
+ \quad & 2\omega_t\langle s_t^*(x_t^\theta) - s_t^{sg[\theta]}(x_t^\theta), s_t^*(x_t^\theta) - s_t^\theta(x_t^\theta|x_0^\theta)\rangle], \ s^\theta = \arg\min_{s'} \mathcal{L}_{\text{DSM}}(s', p_0^\theta), (5)
\end{aligned}$$

where $w_t$ are normalizing weights and gradients w.r.t. $\theta$ are not calculated for the variables under stop-gradient $sg[\cdot]$ operator. The SiD pipeline is two alternating steps: first, refine the *fake score* $s^{sg[\theta]}$ by minimizing DSM loss (1) on new $p_0^\theta$ from the previous step. Then, update the generator $G_\theta$ using the gradient of (5) with frozen $s^{sg[\theta]}$. The $\alpha_{\text{SiD}}$ parameter is chosen from the range $[0.5, 1.2]$, although theoretically only the value $\alpha_{\text{SiD}} = 0.5$ restores true gradient as we show in our paper.

The authors of **FGM** propose a similar approach, but for the flow matching models. Specifically, they also use a generator $G_\theta$ to produce a distribution $p_0^\theta$, but instead of DSM loss (1), consider CFM loss (2). The method minimizes the squared $\ell_2$-distance between the student and teacher drifts:

$$\min_\theta \mathbb{E}_{t\sim[0,T]}\mathbb{E}_{x_t\sim p_t^\theta}[\|u_t^\theta(x_t) - u_t^*(x_t)\|^2], \quad \text{s.t. } u^\theta := \arg\min_v \mathcal{L}_{\text{CFM}}(v, p_0^\theta), \quad (6)$$

where the interpolation path $p_t^\theta$ is constructed between the noise $p_T$ and generator $p_0^\theta$ distributions. To avoid the same problem of differentiating through $\arg\min$ operator as in SiD, the authors derive a tractable loss whose gradients match those of (6):

$$\begin{aligned}
\mathcal{L}_{\text{FGM}}(\theta) \quad := \quad & \mathbb{E}_{t\sim[0,T]}\mathbb{E}_{z\sim p^\mathcal{Z}, x_0^\theta=G_\theta(z), x_t^\theta\sim p_t^\theta}[-\|u_t^*(x_t^\theta) - u_t^{sg[\theta]}(x_t^\theta)\|^2 \quad (7) \\
+ \quad & 2\langle u_t^*(x_t^\theta) - u_t^{sg[\theta]}(x_t^\theta), u_t^*(x_t^\theta) - u_t^\theta(x_t^\theta|x_0^\theta)\rangle], \ \text{s.t. } u^\theta = \arg\min_v \mathcal{L}_{\text{CFM}}(v, p_0^\theta).
\end{aligned}$$

For data-to-data bridge matching models, the **IBMD** method applies the same idea of minimizing the difference between student and teacher drifts using a similar loss. Notably, all these approaches (SiD, FGM, IBMD) are *data-free*, i.e., they do not use any real data from $p_0^*$ to train a generator.

## 2.4 GANS FOR REAL DATA INCORPORATION

Although FGM and SiD methods exhibit strong performance in one-step generation, the generator in these methods is trained under the guidance of the teacher model alone. It means that the generator cannot get more information about the real data that the teacher has learned. For example, it is not expected to correct the teacher's errors. To address this problem, recent works (Yin et al., 2024a; Zhou et al., 2024a) propose adding real data via GANs (Goodfellow et al., 2014). In such approaches, the encoder of fake model is typically augmented with an extra discriminator head $D$ that distinguishes between the generated and real data noising processes via the following adversarial loss:

$$\mathcal{L}_{\text{adv}} = \mathbb{E}_{t\sim[0,T]}\big[\mathbb{E}_{x_t^*\sim p_t^*}\big[\ln D_t\big(x_t^*\big)\big] + \mathbb{E}_{x_t^\theta\sim p_t^\theta}\big[\ln[1 - D_t\big(x_t^\theta\big)]\big]\big]. \quad (8)$$

The overall objective in such hybrid frameworks (Zhou et al., 2024a) consists of *generator loss:*

$$\mathcal{L}_{G_\theta} = \mathcal{L}_{\text{FGM/SiD}}(\theta) + \lambda_{\text{adv}}^{G_\theta}\mathcal{L}_{\text{adv}}(\theta),$$

*And fake model/discriminator loss:*

$$\mathcal{L}_D = \mathcal{L}_{\text{CFM/DSM}} + \lambda_{\text{adv}}^D\mathcal{L}_{\text{adv}}.$$

Here, $\lambda_{\text{adv}}^{G_\theta}$ and $\lambda_{\text{adv}}^D$ are weighting coefficients for the adversarial components. Despite empirical gains, the GAN augmentation entails nontrivial costs: it necessitates architectural modifications, such as an auxiliary discriminator head, and inherits the well-known optimization problems of adversarial training, such as non-stationary objectives, mode collapse, and sensitivity to training dynamics.

## 3 UNIVERSAL DISTILLATION OF MATCHING MODELS WITH REAL DATA

In this section, we present our novel RealUID approach for matching models enhanced by real data. First, we show that the previous data-free distillation methods can be unified under the single UID framework (§3.1). Then, we describe how this framework is connected to prior works (§3.2) and inverse optimization (§3.3). Using this intuition, we propose and discuss the real data modified UID framework (RealUID) with a natural way to incorporate real data without GANs (§3.4).

### 3.1 UNIVERSAL INVERSE DISTILLATION

To learn a complex real data distribution $p_0^*$, one usually trains a *teacher function* $f^* := \arg\min_f \mathcal{L}_{\mathrm{UM}}(f, p_0^*)$ that is then used in a multi-step sampling procedure (Def. 1). To avoid time-consuming sampling, one can train a simple *student generator* $G_\theta : \mathcal{Z} \to \mathbb{R}^D$ with parameters $\theta$ to reproduce the real data $p_0^*$ from the distribution $p^{\mathcal{Z}}$ on the latent space $\mathcal{Z}$. The teacher function serves as a guide that shows how close the student distribution $p_0^\theta$ and the real data $p_0^*$ are. FGM and SiD methods (§2.3) train such generator via minimizing the squared $\ell_2$-distance between the known teacher function $f^*$ and an unknown *student function* $f^\theta := \arg\min_f \mathcal{L}_{\mathrm{UM}}(f, p_0^\theta)$:

$$\mathbb{E}_{t\sim[0,T]}\mathbb{E}_{x_t^\theta\sim p_t^\theta}[\|f_t^*(x_t^\theta) - f_t^\theta(x_t^\theta)\|^2] = \mathbb{E}_{t\sim[0,T]}\mathbb{E}_{x_t^\theta\sim p_t^\theta}[\|f_t^*(x_t^\theta) - \mathbb{E}_{x_0^\theta\sim p_0^\theta(\cdot|x_t^\theta)}[f_t^\theta(x_t^\theta|x_0^\theta)]\|^2]$$

$$= \mathbb{E}_{t\sim[0,T]}\mathbb{E}_{x_t^\theta\sim p_t^\theta}[\|f_t^*(x_t^\theta)\|^2] - 2\cdot\mathbb{E}_{t\sim[0,T]}\mathbb{E}_{x_t^\theta\sim p_t^\theta, x_0^\theta\sim p_0^\theta(\cdot|x_t^\theta)}[\langle f_t^*(x_t^\theta), f_t^\theta(x_t^\theta|x_0^\theta)\rangle]$$

$$+ \underbrace{\mathbb{E}_{t\sim[0,T]}\mathbb{E}_{x_t^\theta\sim p_t^\theta}[\|\mathbb{E}_{x_0^\theta\sim p_0^\theta(\cdot|x_t^\theta)}[f_t^\theta(x_t^\theta|x_0^\theta)]\|^2]}_{\text{not tractable}}, \tag{9}$$

where $p_t^\theta$ is the probability path constructed between generated data $p_0^\theta$ and noise $p_T$. The problem is that the final term (9) cannot be calculated directly, since it involves the math expectation inside the squared norm, unlike the other terms which are linear in the expectations. It means that a simple estimate of $\|f_t^\theta(x_t^\theta|x_0^\theta)\|^2$ using samples $x_0^\theta$ and $x_t^\theta$ will be *biased*. Moreover, to differentiate through the math expectation inside the norm, an explicit dependence of $p_0^\theta$ on $\theta$ is required, while, in practice, usually only dependence of samples $x_0^\theta$ on $\theta$ is known.

**Making loss tractable via linearization.** To resolve this problem, we use the identity $\|a\|^2 = \max_{b\in\mathbb{R}^D}\{-\|b\|^2 + 2\langle b, a\rangle\}, \forall a \in \mathbb{R}^D$. For a fixed time $t$ and point $x_t^\theta$, we reformulate the squared norm (9) as this identity and parametrize vector $b$ via an auxiliary function $\delta : [0,T] \times \mathbb{R}^D \to \mathbb{R}^D$:

$$\mathbb{E}_{\substack{t\sim[0,T],\\ x_t^\theta\sim p_t^\theta}}[\|f_t^*(x_t^\theta) - f_t^\theta(x_t^\theta)\|^2] = \max_{\delta_t(x_t^\theta)} \mathbb{E}_{\substack{t\sim[0,T],\\ x_t^\theta\sim p_t^\theta}} \left[-\|\delta_t(x_t^\theta)\|^2 + 2\langle\delta_t(x_t^\theta), f_t^*(x_t^\theta) - f_t^\theta(x_t^\theta)\rangle\right]$$

$$= \max_{\delta_t(x_t^\theta)} \mathbb{E}_{\substack{t\sim[0,T],\\ x_t^\theta\sim p_t^\theta}}\left[-\|\delta_t(x_t^\theta)\|^2 + 2\langle\delta_t(x_t^\theta), f_t^*(x_t^\theta)\rangle - \underbrace{2\langle\delta_t(x_t^\theta), \mathbb{E}_{x_0^\theta\sim p_0^\theta(\cdot|x_t^\theta)}[f_t^\theta(x_t^\theta|x_0^\theta)]\rangle}_{\text{linear and tractable}}\right]. \tag{10}$$

Now, all loss terms are linear and can be sampled. The parameterization $\delta = f^* - f$ with a *fake function* $f : [0,T] \times \mathbb{R}^D \to \mathbb{R}^D$ allows us to get an elegant form:

$$(10) = \max_{f_t(x_t^\theta)} \mathbb{E}_{\substack{t\sim[0,T], x_0^\theta\sim p_0^\theta,\\ x_t^\theta\sim p_t^\theta(\cdot|x_0^\theta)}}\left\{-\|f_t^*(x_t^\theta) - f_t(x_t^\theta)\|^2 + 2\langle f_t^*(x_t^\theta) - f_t(x_t^\theta), f_t^*(x_t^\theta) - f_t^\theta(x_t^\theta|x_0^\theta)\rangle\right\} \tag{11}$$

$$= \max_{f_t(x_t^\theta)}\left\{\underbrace{\mathbb{E}_{\substack{t\sim[0,T], x_0^\theta\sim p_0^\theta,\\ x_t^\theta\sim p_t^\theta(\cdot|x_0^\theta)}}[\|f_t^*(x_t^\theta) - f_t^\theta(x_t^\theta|x_0^\theta)\|^2]}_{=\mathcal{L}_{\mathrm{UM}}(f^*, p_0^\theta)} - \underbrace{\mathbb{E}_{\substack{t\sim[0,T], x_0^\theta\sim p_0^\theta,\\ x_t^\theta\sim p_t^\theta(\cdot|x_0^\theta)}}[\|f_t(x_t^\theta) - f_t^\theta(x_t^\theta|x_0^\theta)\|^2]}_{=\mathcal{L}_{\mathrm{UM}}(f, p_0^\theta)}\right\}. \tag{12}$$

**Summary.** We build a universal distillation framework as a single min-max optimization (13), implicitly minimizing squared $\ell_2$-distance between teacher and student functions. When real and generated probability paths match, these functions match as well, and the distance attains its minimum.

**Theorem 1** (**Real data generator minimizes UID loss**). *Let teacher $f^* := \arg\min_f \mathcal{L}_{UM}(f, p_0^*)$ be the minimizer of UM loss (Def. 1) on real data $p_0^* \in \mathcal{P}(\mathbb{R}^D)$. Then, real data generator $G_{\theta^*}$ s.t. $p_0^{\theta^*} = p_0^*$ is a solution to the min-max optimization of **Universal Inverse Distillation (UID) loss** $\mathcal{L}_{UID}(f, p_0^\theta)$ over fake function $f$ and generator distribution $p_0^\theta$:*

$$\min_\theta \max_f \left\{\mathcal{L}_{UID}(f, p_0^\theta) := \mathcal{L}_{UM}(f^*, p_0^\theta) - \mathcal{L}_{UM}(f, p_0^\theta)\right\}. \tag{13}$$

**Lemma 1** (**UID loss minimizes squared $\ell_2$-distance**). *Maximization of UID loss (13) over fake function $f$ represents the squared $\ell_2$-distance between the student function $f^\theta := \arg\min_f \mathcal{L}_{UM}(f, p_0^\theta)$ and the teacher $f^* := \arg\min_f \mathcal{L}_{UM}(f, p_0^*)$:*

$$\max_f \mathcal{L}_{UID}(f, p_0^\theta) = \mathbb{E}_{t\sim[0,T]}\mathbb{E}_{x_t^\theta\sim p_t^\theta}[\|f_t^*(x_t^\theta) - f_t^\theta(x_t^\theta)\|^2]. \tag{14}$$

In UID framework, the trained fake model simply learns the current student function $f^\theta$ by minimizing UM loss $\mathcal{L}_{\mathrm{UM}}(f, p_0^\theta)$. Note that for points $x_t^\theta$ out of the generator's domain s.t. $p_t^\theta(x_t^\theta) \approx 0$, the distance (14) vanishes, and the generator cannot receive feedback from the uncovered real data. Moreover, if the teacher function is inaccurate, the generator will learn it with all inaccuracies.

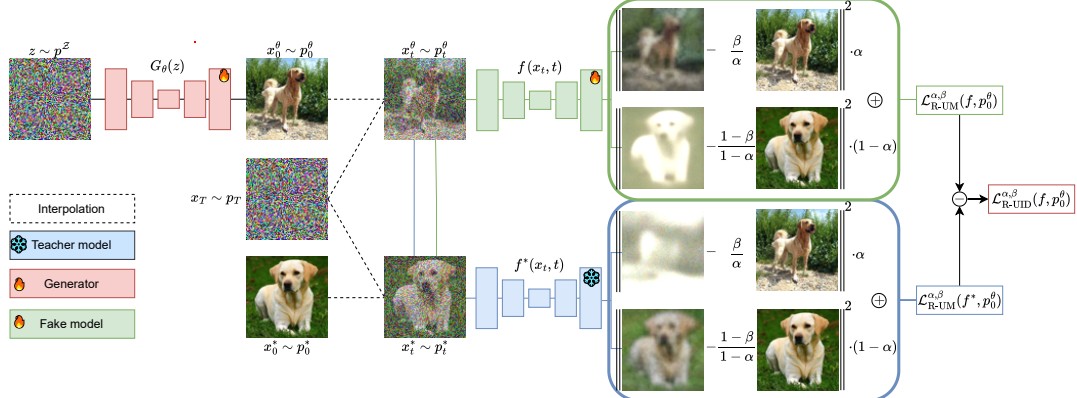

Figure 1: Pipeline of **our RealUID distillation framework** (§3) with the direct incorporation of real data $p_0^*$ adjusted by parameters $\alpha, \beta \in (0, 1]$. The figure depicts flow matching models predicting denoised samples. It distills a costly frozen teacher model $f^*$ (blue) into a one-step generator $G_\theta$ (red) upon min-max optimization of $\mathcal{L}_{\text{R-UID}}^{\alpha,\beta}(f, p_0^\theta)$ loss over fake model $f$ (green) and generator distribution $p_0^\theta$ with parameters $\theta$. It updates the fake model several times per one generator update for stability. Algorithm's pseudocode is located in Appendix B.

### 3.2  RELATION TO PRIOR DISTILLATION WORKS

FGM and SiD approaches formulate distillation as a constraint minimization of generator loss subject to the optimal fake model. For generator updates, the explicit UID loss (11) matches FGM loss (7) and SiD loss (5) with $\alpha_{\text{SiD}} = 0.5$. For a fake model, it also minimizes the UM loss on the generated data. The work (Gushchin et al., 2025) was the first to formulate the distillation of bridge matching models in their IBMD framework as a min-max optimization of the single loss (12).

Although previous works derive the same losses, we give a new, simple explanation using a linearization technique. *This technique is more powerful and general for handling intractable terms than complex proofs for concrete models from FGM, SiD or IBMD.* It allows us to build other distillations, e.g., a loss for minimizing the $\ell_2$-distance instead of the squared one (see Appendix A.4).

### 3.3  CONNECTION WITH INVERSE OPTIMIZATION

We derive UID loss (13) by minimizing the squared $\ell_2$-distance between teacher and student functions. However, this loss admits another interpretation: its structure is typical for inverse optimization (Chan et al., 2025). In this framework, one considers a parametric family of optimization problems $\min_f \mathcal{L}(f, \theta)$ with objective loss $\mathcal{L}(f, \theta)$ depending on argument $f$ and parameters $\theta$. The goal is to find the parameters $\theta^*$ that yield a known, desired solution $f^* = \arg\min_f \mathcal{L}(f, \theta^*)$. One standard way to recover the required parameters is to solve the same min-max problem as (13):

$$\min_\theta \max_f \left\{ \mathcal{L}(f^*, \theta) - \mathcal{L}(f, \theta) \right\} \sim \min_\theta \left\{ \mathcal{L}(f^*, \theta) - \min_f \{ \mathcal{L}(f, \theta) \} \right\}. \tag{15}$$

The inverse problem (15) always has minimum 0 which is attained when $\theta = \theta^*$.

Although the inverse optimization can handle arbitrary losses $\mathcal{L}$, it does not describe the properties of the optimized functions or how to find solutions. In our case, we show that all losses are tractable and minimize the distances between teacher and student functions (Lemmas 1 and 2).

### 3.4  REALUID: NATURAL APPROACH FOR REAL DATA INCORPORATION

Previous distillation methods add real data during training only via GANs with extra discriminator and adversarial loss. We propose a simpler, more natural way that requires no extra models or losses.

Based on intuition from inverse optimization (§3.3), we see that the min-max inverse problem (15) is compatible with other losses. It allows us to redesign the UM loss (3) to incorporate real data into it. A key constraint is that the loss must still yield the same teacher upon minimization on the real data. Thus, we derive a novel Unified Matching loss with real data - a weighted sum of two UM-like losses on generated and real data parameterized by $\alpha, \beta \in (0, 1]$ which control the weights.

**Definition 2.** *We define* **Universal Matching loss with real data (RealUM)** $\mathcal{L}_{R\text{-}UM}^{\alpha,\beta}(f, p_0^\theta)$ *that is parametrized by* $\alpha, \beta \in (0, 1]$ *and takes trainable function* $f$ *and generated data* $p_0^\theta$ *as arguments:*

$$
\mathcal{L}_{R\text{-}UM}^{\alpha,\beta}(f, p_0^\theta) = \underbrace{\alpha \cdot \mathbb{E}_{t\sim[0,T]}\mathbb{E}_{x_0^\theta\sim p_0^\theta, x_t^\theta\sim p_t^\theta(\cdot|x_0^\theta)}\left[\|f_t(x_t^\theta) - \frac{\beta}{\alpha}f^\theta(x_t^\theta|x_0^\theta)\|^2\right]}_{\text{generated data } p_0^\theta \text{ term}}
$$

$$
+ \underbrace{(1-\alpha) \cdot \mathbb{E}_{t\sim[0,T]}\mathbb{E}_{x_0^*\sim p_0^*, x_t^*\sim p_t^*(\cdot|x_0^*)}\left[\|f_t(x_t^*) - \frac{1-\beta}{1-\alpha}f^*(x_t^*|x_0^*)\|^2\right]}_{\text{real data } p_0^* \text{ term}}. \quad (16)
$$

*For* $\alpha = 1$*, we consider only* $\beta = 1$*, i.e., the pure generated data term.*

RealUM loss (16) for all $\alpha, \beta$ and UM loss (3) yield the same teacher when input is real data $p_0^*$, since if we consider only the $f$-dependent terms in the losses, we have:

$$
\mathcal{L}_{R\text{-}UM}^{\alpha,\beta}(f, p_0^*) \propto \mathbb{E}_{t, x_0^*, x_t^*}[\underbrace{[\alpha + (1-\alpha)]}_{=1} \cdot \langle f_t(x_t^*), f_t(x_t^*)\rangle + 2\underbrace{[\alpha \cdot \frac{\beta}{\alpha} + (1-\alpha) \cdot \frac{1-\beta}{1-\alpha}]}_{=1}\langle f_t(x_t^*), f_t^*(x_t^*|x_0^*)\rangle]
$$

$$
\propto \mathbb{E}_{t, x_0^*, x_t^*}[\|f_t(x_t^*) - f_t^*(x_t^*|x_0^*)\|^2] \implies \arg\min {}_f \mathcal{L}_{R\text{-}UM}^{\alpha,\beta}(f, p_0^*) = \arg\min {}_f \mathcal{L}_{UM}(f, p_0^*) = f^*.
$$

Hence, the min-max inverse scheme (15) with RealUM loss and the old teacher $f^*$ still has a real data generator as a solution, but now real data is incorporated via the real data terms of $\mathcal{L}_{R\text{-}UM}^{\alpha,\beta}(f, p_0^\theta)$:

$$
\min {}_\theta\{\underbrace{\mathcal{L}_{R\text{-}UM}^{\alpha,\beta}(f^*, p_0^\theta) - \min {}_f\{\mathcal{L}_{R\text{-}UM}^{\alpha,\beta}(f, p_0^\theta)\}}_{\geq 0}\} \stackrel{p_0^\theta=p_0^*}{=} \mathcal{L}_{R\text{-}UM}^{\alpha,\beta}(f^*, p_0^*) - \underbrace{\min {}_f\{\mathcal{L}_{R\text{-}UM}^{\alpha,\beta}(f, p_0^*)\}}_{=\mathcal{L}_{R\text{-}UM}^{\alpha,\beta}(f^*, p_0^*)} = 0.
$$

**Theorem 2** (**Real data generator minimizes RealUID loss**). *Let teacher* $f^* := \arg\min_f \mathcal{L}_{UM}(f, p_0^*)$ *be the minimizer of UM loss (Def. 1) on real data* $p_0^* \in \mathcal{P}(\mathbb{R}^D)$. *Then, real data generator* $G_{\theta^*}$ *s.t.* $p_0^{\theta^*} = p_0^*$ *is a solution to the min-max optimization of* **Universal Inverse Distillation loss with real data (RealUID)** $\mathcal{L}_{R\text{-}UID}^{\alpha,\beta}(f, p_0^\theta)$ *over fake function* $f$ *and generator distribution* $p_0^\theta$ *(see Def. 2):*

$$
\min {}_\theta \max {}_f \left\{\mathcal{L}_{R\text{-}UID}^{\alpha,\beta}(f, p_0^\theta) := \mathcal{L}_{R\text{-}UM}^{\alpha,\beta}(f^*, p_0^\theta) - \mathcal{L}_{R\text{-}UM}^{\alpha,\beta}(f, p_0^\theta)\right\}. \quad (17)
$$

We provide analysis of RealUID in Appendix A.1, below we highlight the most important findings.

**Role of coefficients** $\alpha, \beta$. Our RealUID uses real data only to minimize $\mathcal{L}_{R\text{-}UM}^{\alpha,\beta}(f, p_0^\theta)$ loss over fake function $f$. Thus, the trained fake function memorizes both the real data and the generator's current state. In turn, the generator is influenced by the real data indirectly, only via this fake function. As shown in Lemma 2, RealUID implicitly minimizes the rescaled distance (18) between the teacher and generator functions. This distance is still minimal when $p_0^\theta = p_0^*$, alternatively proving Theorem 2.

**Lemma 2** (**Distance minimized by RealUID loss**). *Maximization of RealUID loss (16) over fake function* $f$ *represents the weighted squared* $\ell_2$*-distance between the student function* $f^\theta := \arg\min_f \mathcal{L}_{UM}(f, p_0^\theta)$ *and the teacher* $f^* := \arg\min_f \mathcal{L}_{UM}(f, p_0^*)$ :

$$
\max_f \mathcal{L}_{R\text{-}UID}^{\alpha,\beta}(f, p_0^\theta) = \mathbb{E}_{\substack{t\sim[0,T], \\ x_t^*\sim p_t^*}}\left[\frac{\|\frac{\beta}{\alpha} \cdot [p_t^*(x_t^*)f_t^*(x_t^*) - p_t^\theta(x_t^*)f_t^\theta(x_t^*)] + (p_t^\theta(x_t^*) - p_t^*(x_t^*)) \cdot f_t^*(x_t^*)\|^2}{p_t^*(x_t^*)((1-\alpha)p_t^*(x_t^*) + \alpha p_t^\theta(x_t^*))/\alpha^2}\right]. \quad (18)
$$

The proof of Lemma 2 is located in Appendix A.1.2. With the help of real data, our RealUID loss now provides the generator with the feedback on the real data domain it needs to cover, i.e., the distance (18) does not vanish for points $x_t$ s.t. $p^\theta(x_t) \approx 0, p^*(x_t) \gg 0$ (see Appendix A.1.3). Moreover, if teacher function is inaccurate, RealUID can now provably fix teacher's errors (see Appendix A.1.4).

**Choice of coefficients** $\alpha, \beta$. Lemma 2 shows that, instead of values $\alpha$ and $\beta$, actually the values $\alpha$ and $\beta/\alpha$ determine the balance between real and generated data in the minimized distance (18). Furthermore, coefficient $\alpha$ only sets the general scaling of the distance, while $\beta/\alpha$ *plays the most important role*, as it determines the relation between $f_t^\theta$ and $f_t^*$ inside the distance.

The value $\beta/\alpha = 1$ yields the distance identical to the data-free distance (14) up to scaling. Thus, even when $\alpha = \beta < 1$ and real data is formally added, it may have no effect on the generator. Excessively

low $\alpha$ and $\beta$ diminish the effect of the generated data terms in the trained fake function, leading to vanishing gradients. The same issue occurs with $\beta/\alpha \ll 1$ in (18), while $\beta/\alpha \gg 1$ diminish the effect of real data in the right term of (18). See complete distance analysis in Appendix A.1.3. Moreover, if teacher function is inaccurate, only the choice $\beta/\alpha \neq 1$ can fix teacher's errors (see Appendix A.1.4).

**Hence, good coefficients $\alpha, \beta \in (0, 1]$ can be chosen by first finding good $\beta/\alpha \neq 1$, as it has the largest impact, and then adjusting $\alpha < 1$. Both $\beta/\alpha$ and $\alpha$ should be close to** 1.

**Comparison with GAN-based methods.**   Unlike SiD and FGM with GANs (8), we do not use extra adversarial losses and discriminator to incorporate real data. We only modify UM loss, preserving its core structure and fake model architecture. While general adversarial loss is unrelated to the main distillation loss and has uninterpretable scaling hyperparameters, our RealUID loss and weighting coefficients $\alpha, \beta \in (0, 1]$ come naturally from the data-free UID loss. The original UID loss (13), equivalent to FGM (7) and SiD (5) with $\alpha_{\text{SiD}} = 0.5$, is obtained when $\alpha = \beta = 1$.

**Alternative loss form.**   Our RealUID is implicitly related to the linearization scheme used to obtain data-free UID (§3.1). *The loss (17) can be derived by splitting each term in the linearized UID loss (10) between real and generated data according to proportions $\alpha$ and $\beta$* (see Appendix A.1.1). This form helps to prove RealUID's properties and extend it beyond the inversion scheme (§5).

**Extension for Bridge Matching and Stochastic Interpolants frameworks.**   In Appendix C.3, we demonstrate that our framework can be easily extended to data-to-data matching models by parameterizing the generated data coupling $\pi^\theta(x_0, x_T)$ instead of the data distribution $p_0^\theta$.

## 4    EXPERIMENTS

All our PyTorch implementations and the latest checkpoints are publicly available in



https://github.com/David-cripto/RealUID.



This section provides an ablation study and evaluation of our RealUID, assessing both its performance and computational efficiency. We begin in (§4.1) by detailing the experimental setup based on flow matching models. In (§4.2), we show that our incorporation of real data via coefficients $\alpha, \beta$ improves performance, speeds up convergence, and enables effective fine-tuning. In (§4.3), we assess the benchmark performance and computational demands of RealUID relative to SOTA methods. Additional experimental details and results are provided in Appendix D.

### 4.1    EXPERIMENTAL SETUP

**Datasets and Evaluation Protocol.**   The experiments were conducted on the CIFAR-10 dataset with $32 \times 32$ resolution (Krizhevsky et al., 2009) and on the CelebA dataset with $64 \times 64$ resolution (Liu et al., 2015), see Appendix D.3. In line with the prior works (Karras et al., 2019; 2022), we report test FID scores (Heusel et al., 2017), computed using 50k generated samples.

**Implementation Details.**   We implement our RealUID framework for flow matching models from Appendix B. In contrast to prior studies (Zhou et al., 2024b;a; Huang et al., 2024) which employ the computationally demanding EDM architecture (Karras et al., 2022) our work adopts a more lightweight alternative (Tong et al., 2024). We also train our own flow matching teacher models using CFM loss (2). Further implementation details and efficiency analysis are provided in Appendix D.1.

### 4.2    BENCHMARKING METHODS UNDER A UNIFIED EXPERIMENTAL CONFIGURATION

We evaluate RealUID under a unified experimental protocol (fixed architecture and implementation). We begin by conducting an ablation over $\alpha, \beta$ to assess the influence of real-data incorporation. We then compare RealUID to a GAN-based alternative, showing that RealUID achieves comparable or superior accuracy. Furthermore, we analyze convergence, indicating that RealUID variants with real data train substantially faster than baselines without real-data. Finally, we explore a fine-tuning stage initialized from strong RealUID checkpoints, showing further performance gains.

**Ablation study of coefficients $\alpha, \beta$.**   We restrict the search for optimal parameters $\alpha$ and $\beta$ to values near 1, specifically $\alpha, \beta \in [0.85, 1.0]$ with increments of 0.02. Setting these parameters too low leads

to noisy generated samples. Following the analysis in (§3.4), we perform a grid search over the values $\alpha$ and $\beta/\alpha$ instead of the original $\alpha$ and $\beta$. The results are reported in Table 1. As a baseline, we highlight the UID model without data incorporation, i.e., our RealUID with $\alpha = 1.0, \beta = 1.0$.

As shown in the table, the ratio $\beta/\alpha$ has the largest impact on the final metrics, while $\alpha$ only adjusts them. Using real data with $\beta/\alpha = 1$ or with large values outside the range $[0.98, 1.02]$ consistently degrades performance. In contrast, values $\beta/\alpha = 0.98$ or $\beta/\alpha = 1.02$ outperform the baseline for a majority of $\alpha$. Note that these practical results match the theoretical description in (§3.4).

| Generation | $\alpha\backslash\frac{\beta}{\alpha}$ | 0.96 | 0.98 | 1.00 | 1.02 | 1.04 |
|---|---|---|---|---|---|---|
| Unconditional | 0.90 | 2.66 | 2.44 | 2.66 | 2.25 | 2.55 |
| | 0.92 | 2.73 | 2.36 | 2.65 | 2.23 | 2.66 |
| | 0.94 | 2.79 | 2.35 | 2.65 | 2.28 | 2.58 |
| | 0.96 | 2.85 | 2.37 | 2.58 | 2.29 | 2.65 |
| | 0.98 | 2.97 | 2.33 | 2.62 | 2.38 | - |
| | 1.0 | - | - | 2.58 | - | - |
| Conditional | 0.90 | 2.34 | 2.16 | 2.38 | 2.19 | 2.26 |
| | 0.92 | 2.28 | 2.12 | 2.35 | 2.21 | 2.23 |
| | 0.94 | 2.29 | 2.13 | 2.35 | 2.19 | 2.25 |
| | 0.96 | 2.36 | 2.09 | 2.32 | 2.13 | 2.27 |
| | 0.98 | 2.34 | 2.02 | 2.26 | 2.05 | - |
| | 1.0 | - | - | 2.21 | - | - |

| Generation | $\lambda_{\text{adv}}^{G_\theta}$ | $\lambda_{\text{adv}}^{D}$ | FID ($\downarrow$) |
|---|---|---|---|
| Unconditional | 0.1 | 0.3 | 2.42 |
| | 0.3 | 1 | 2.29 |
| | 1 | 3 | 2.39 |
| | 5 | 15 | 2.54 |
| Conditional | 0.1 | 0.3 | 2.22 |
| | 0.3 | 1 | 2.12 |
| | 1 | 3 | 2.15 |
| | 5 | 15 | 2.40 |

Table 1: Ablation studies of our $(\alpha, \frac{\beta}{\alpha})$ parameters in the left table and adversarial weighting parameters $(\lambda_{\text{adv}}^{G_\theta}, \lambda_{\text{adv}}^{D})$ in the right table for CIFAR-10. The baseline RealUID ($\alpha = 1.0, \beta = 1.0$) does not use real data. Configurations that sligtly and substantially outperform the baseline are highlighted. All values report FID $\downarrow$, where lower is better. The best configuration in each case is **bolded**. The mark "–" denotes infeasible parameters.

**Comparison with GAN-based method.** We integrate the GAN-based approach (8) proposed by (Zhou et al., 2024a) as an alternative method for incorporating real data, enabling a direct comparison with our RealUID formulation. We combine the GAN loss with the UID baseline. As shown in Table 1, the best-performing configurations are achieved with GAN losses ($\lambda_{\text{adv}}^{G_\theta} = 0.3, \lambda_{\text{adv}}^{D} = 1$). While this setup performs comparably to RealUID ($\alpha = 0.92, \beta = 0.94$) in the unconditional setting, it remains clearly inferior to RealUID ($\alpha = 0.98, \beta = 0.96$) in the conditional case.

Table 2: This table presents the results of ablation study of our RealUID framework, evaluated using the FID metric under both unconditional and conditional generation setups. The Teacher Flow model with 100 NFE is reported as a reference. The performance of the UID (FGM) baseline without real-data incorporation is indicated in *italic*. For emphasis, we underline the two counterparts that incorporate real data: the GAN-based and our RealUID methods. The best-performing configurations, obtained via an additional fine-tuning stage, are highlighted in **bold**. Qualitative results are presented in Appendix D.5.1.

| Model | FID ($\downarrow$) |
|---|---|
| Teacher Flow (NFE=100) | 3.57 |
| UID (FGM) | *2.58* |
| UID + GAN ($\lambda_{\text{adv}}^{G_\theta} = 0.3, \lambda_{\text{adv}}^{D} = 1 \mid \lambda_{\text{FT}}^{G_\theta} = 25, \lambda_{\text{FT}}^{D} = 75$) | 2.10 |
| RealUID ($\alpha = 0.92, \beta = 0.94 \mid \alpha_{\text{FT}} = 0.92, \beta_{\text{FT}} = 0.86$) **(Ours)** | **1.98** |

| Model | FID ($\downarrow$) |
|---|---|
| Teacher Flow (NFE=100) | 5.56 |
| UID (FGM) | *2.21* |
| UID + GAN ($\lambda_{\text{adv}}^{G_\theta} = 0.3, \lambda_{\text{adv}}^{D} = 1 \mid \lambda_{\text{FT}}^{G_\theta} = 25, \lambda_{\text{FT}}^{D} = 75$) | 1.88 |
| RealUID ($\alpha = 0.98, \beta = 0.96 \mid \alpha_{\text{FT}} = 0.96, \beta_{\text{FT}} = 0.92$) **(Ours)** | **1.87** |

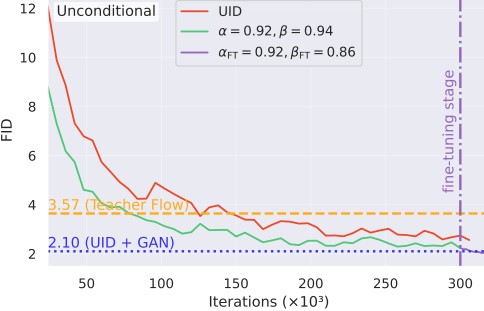
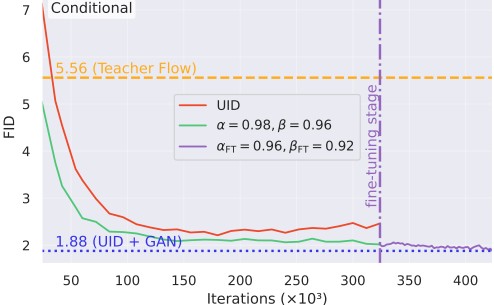

Figure 2: Evolution of FID during CIFAR-10 distillation for (i) the UID (FGM) baseline, (ii) the best-performing RealUID configurations, and (iii) subsequent fine-tuning, evaluated in both unconditional and conditional settings. The performances of Teacher Flow and UID+GAN are indicated by horizontal lines in their respective colors.

**Convergence Speed.** Our RealUID$(\alpha, \beta)$ with parameters which are highlighted in Table 1 achieves faster convergence than the UID baseline. For clarity, we present qualitative comparisons in Figure 2. The best RealUID configurations reach the saturated performance level of the baseline after $\sim$100k iterations, whereas the baseline requires $\sim$300k iterations to achieve comparable metrics.

**Fine-tuning stage.** We observe that RealUID and GAN frameworks offer substantial flexibility for fine-tuning. In this procedure, the generator is initialized from the best-performing checkpoint obtained during training from scratch of the corresponding framework, while the fake model is initialized from the teacher. Fine-tuning then proceeds with new values $\alpha_{FT}$ and $\beta_{FT}$ for our RealUID and $\lambda_{FT}^{G_\theta}$ and $\lambda_{FT}^{D}$ for GANs. We present the best-found fine-tuning configurations for both methods in Table 2. Ablation study analyzing the effect of loss coefficients is provided in Appendix D.2.

**Scaling to larger datasets.** In Appendix D.3, we provide the results of the same ablation study on the CelebA dataset with $64 \times 64$ resolution. Notably, our RealUID performance and the optimal values $\beta/\alpha$ remain the same across datasets.

### 4.3 BASELINE COMPARISON

As shown in Tables 3 and 4, our RealUID after fine-tuning consistently outperforms all prior flow-based models on CIFAR-10, significantly surpassing the strongest flow distillation baseline, FGM. Despite its compact and lightweight architecture (§4.1) with nearly 2× faster inference, it achieves performance comparable to leading diffusion distillation methods SiD ($\alpha_{SiD}{=}1.0\backslash1.2$), while falling short of adversarially enhanced models such as SiD$^2$A. We hypothesize that this performance gap is attributed to architectural and teacher capacity differences rather than the lack of adversarial loss.

**Our latest checkpoints and metrics (Appendix D.4) are available in our repository.**

Table 3: Comparison of *unconditional* generation on CIFAR-10. The best method under the FID metric in each section is highlighted with **bold**.

| Family | Model | NFE | FID ($\downarrow$) |
|---|---|---|---|
| | DDPM (Ho et al., 2020) | 1000 | 3.17 |
| | VP-EDM (Karras et al., 2022) | 35 | 1.97 |
| | StyleGAN2+ADA+Tune (Karras et al., 2020) | 1 | 2.92 |
| | StyleGAN2+ADA+Tune+DI (Luo et al., 2023) | 1 | 2.71 |
| | Diffusion ProjectedGAN (Wang et al., 2022) | 1 | 2.54 |
| | iCT-deep (Song & Dhariwal, 2023) | 1 | 2.51 |
| Diffusion & GAN | Diff-Instruct (Luo et al., 2023) | 1 | 4.53 |
| | DMD (Yin et al., 2024b) | 1 | 3.77 |
| | CTM (Kim et al., 2023) | 1 | 1.98 |
| | sCD (Lu & Song, 2024) | 1 | 3.66 |
| | sCT (Lu & Song, 2024) | 1 | 2.85 |
| | SiD, $\alpha_{SiD} = 1.0$ (Zhou et al., 2024b) | 1 | 2.03 |
| | SiD, $\alpha_{SiD} = 1.2$ (Zhou et al., 2024b) | 1 | 1.92 |
| | SiDA, $\alpha_{SiD} = 1.0$ (Zhou et al., 2024a) | 1 | 1.52 |
| | SiD$^2$A, $\alpha_{SiD} = 1.2$ (Zhou et al., 2024a) | 1 | 1.52 |
| | SiD$^2$A, $\alpha_{SiD} = 1.0$ (Zhou et al., 2024a) | 1 | **1.50** |
| | CFM (Yang et al., 2024) | 2 | 5.34 |
| | IMM (Zhou et al., 2025) | 1 | 3.20 |
| Flow-based | MeanFlow (Geng et al., 2025) | 1 | 2.92 |
| | FACM (Peng et al., 2025) | 1 | 2.69 |
| | 1-ReFlow (+Distill) (Liu et al., 2023) | 1 | 6.18 |
| | 2-ReFlow (+Distill) (Liu et al., 2023) | 1 | 4.85 |
| | 3-ReFlow (+Distill) (Liu et al., 2023) | 1 | 5.21 |
| | FGM (Huang et al., 2024) | 1 | 3.08 |
| | RealUID + FT (**Ours**) | 1 | **1.98** |

Table 4: Comparison of *conditional* generation on CIFAR-10. The best method under the FID metric in each section is highlighted with **bold**.

| Family | Model | NFE | FID ($\downarrow$) |
|---|---|---|---|
| | VP-EDM (Karras et al., 2022) | 35 | 1.79 |
| | GET-Base (Geng et al., 2023) | 1 | 6.25 |
| | BigGAN (Brock et al., 2018) | 1 | 14.73 |
| | BigGAN+Tune (Brock et al., 2018) | 1 | 8.47 |
| | StyleGAN2+ADA (Karras et al., 2020) | 1 | 3.49 |
| | StyleGAN2+ADA+Tune (Karras et al., 2020) | 1 | 2.42 |
| | StyleGAN2+ADA+Tune+DI (Luo et al., 2023) | 1 | 2.27 |
| | StyleGAN-XL (Sauer et al., 2022) | 1 | 1.85 |
| | StyleSAN-XL (Takida et al., 2023) | 1 | **1.36** |
| Diffusion & GAN | Diff-Instruct (Luo et al., 2023) | 1 | 4.19 |
| | DMD (Yin et al., 2024b) | 1 | 2.66 |
| | DMD (*w.o.* KL) (Yin et al., 2024b) | 1 | 3.82 |
| | DMD (*w.o. reg.*) (Yin et al., 2024b) | 1 | 5.58 |
| | GDD-I (Zheng et al., 2024) | 1 | 1.44 |
| | CTM (Kim et al., 2023) | 1 | 1.73 |
| | SiD, $\alpha_{SiD} = 1.0$ (Zhou et al., 2024b) | 1 | 1.93 |
| | SiD , $\alpha_{SiD} = 1.2$ (Zhou et al., 2024b) | 1 | 1.71 |
| | SiDA, $\alpha_{SiD} = 1.0$ (Zhou et al., 2024b) | 1 | 1.44 |
| | SiD$^2$A, $\alpha_{SiD} = 1.0$ (Zhou et al., 2024a) | 1 | 1.40 |
| | SiD$^2$A, $\alpha_{SiD} = 1.2$ (Zhou et al., 2024a) | 1 | 1.39 |
| Flow-based | FGM (Huang et al., 2024) | 1 | 2.58 |
| | RealUID + FT (**Ours**) | 1 | **1.87** |

## 5 DISCUSSION AND EXTENSIONS

**Extensions.** Our RealUID framework (§3.4 and Appendix C) can distill Flow/Bridge Matching, diffusion models, and Stochastic Interpolants enhanced by a novel natural way to incorporate real data. In Appendix A, we provide three extensions of our RealUID: more flexible General RealUID (Appendix A.2), General SiD framework for all matching models with real data and $\alpha_{SiD} \neq 1/2$ (Appendix A.3) and Normalized RealUID for minimizing non-squared $\ell_2$-distance (Appendix A.4).

**Relation to DMD.** Instead of minimizing the squared $\ell_2$-distance between the score functions, *Distribution Matching Distillation* (Luo et al., 2023; Wang et al., 2023; Yin et al., 2024b;a, **DMD**) approach minimizes the KL divergence between the real and generated data. Its gradients are computed using the generator and teacher score functions, leading to the similar alternating updates. *The DMD is a special case of our UID framework with the special KL loss (different from UM loss) and only one parameter $\alpha = \beta$ (Appendix A.5).*

## 6 BROADER IMPACT

This paper presents work whose goal is to advance the field of Artificial Intelligence, Machine Learning and Generative Modeling. There are many potential societal consequences of our work, none which we feel must be specifically highlighted here.

**Acknowledgements.** The work was supported by the grant for research centers in the field of AI provided by the Ministry of Economic Development of the Russian Federation in accordance with the agreement 000000C313925P4F0002 and the agreement №139-10-2025-033.

## 7 LLM USAGE

Large Language Models (LLMs) were used only to assist with rephrasing sentences and improving the clarity of the text. All scientific content, results, and interpretations in this paper were developed solely by the authors.

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

CONTENTS

## A    Theoretical proofs and extensions

In this appendix, we discuss our RealUID framework (Appendix A.1) in theoretical details and provide three extensions of it: *General RealUID* framework with 3 degrees of freedom (Appendix A.2), *General SiD framework with real data* (Appendix A.3) and *Normalized RealUID* framework for minimizing $\ell_2$-distance between teacher and student functions instead of the squared one (Appendix A.4). All proofs are based on the linearization technique and splitting terms in linearized decomposition between real and generated data. We also show that DMD approach is a special case of our UID framework and we can similarly incorporate real data into it (Appendix A.5).

### A.1    RealUID theoretical properties

In this section, we discuss our RealUID loss in detail. We begin by presenting its alternative form and how it connects linearization technique and real data incorporation (Appendix A.1.1). We then demonstrate that the loss minimizes a squared $\ell_2$-distance between the rescaled teacher and student functions (Appendix A.1.2). Finally, we provide the motivation of the best choice of coefficients $\alpha \neq \beta$ from the perspectives of the better distance (Appendix A.1.3) and the correction of the teacher's errors (Appendix A.1.4).

### A.1.1    Alternative RealUID split form

Let us recall the linearization trick that we apply to make the minimized squared norm between the student function $f^\theta$ and the teacher $f^*$ tractable. For each time $t$ and generated point $x_t^\theta$, we restate this squared norm as the identity $\|a\|^2 = \max_{b \in \mathbb{R}^D} \{-\|b\|^2 + 2\langle b, a \rangle\}, \forall a \in \mathbb{R}$ and use an auxiliary function $\delta_t(x_t)$ to parametrize a vector $b$. In the end, we substitute the student function $f_t^\theta(x_t^\theta)$ with its conditional and differentiable estimate $f_t^\theta(x_t^\theta | x_0^\theta)$:

$$\mathbb{E}_{\substack{t \sim [0,T], \\ x_t^\theta \sim p_t^\theta}}[\|f_t^*(x_t^\theta) - f_t^\theta(x_t^\theta)\|^2] = \max_{\delta_t(x_t^\theta)} \mathbb{E}_{\substack{t \sim [0,T], \\ x_t^\theta \sim p_t^\theta}} \left[ -\|\delta_t(x_t^\theta)\|^2 + 2\langle \delta_t(x_t^\theta), f_t^*(x_t^\theta) - f_t^\theta(x_t^\theta) \rangle \right]$$

$$= \mathbb{E}_{\substack{t \sim [0,T], x_0^\theta \sim p_0^\theta, \\ x_t^\theta \sim p_t^\theta(\cdot | x_0^\theta)}}[-\|\delta_t(x_t^\theta)\|^2 + 2\langle \delta_t(x_t^\theta), f_t^*(x_t^\theta) \rangle - 2\langle \delta_t(x_t^\theta), f_t^\theta(x_t^\theta | x_0^\theta) \rangle]. \tag{19}$$

In addition, we use parameterization $\delta = f^* - f$ with a fake model $f$ to obtain our UID loss which matches the previous distillation losses.

Originally, we derived our RealUID loss (17) from the idea of *splitting each term in the linearized form of data-free UID* (19) *between the generated and real data in proportions defined by coefficients $\alpha$ and $(1-\alpha)$, $\alpha$ and $(1-\alpha)$ and $\beta$ and $(1-\beta)$*. We present the split form of RealUID loss in Lemma 3, and this form completely matches the inverse optimization form defined in Theorem 2.

**Lemma 3** (**RealUID split form**). *The RealUID loss* (17) *can be restated as*

$$\mathcal{L}_{R\text{-}UID}^{\alpha,\beta}(f,p_0^\theta) = \mathbb{E}_{\substack{t\sim[0,T],x_0^\theta\sim p_0^\theta, \\ x_t^\theta\sim p_t^\theta(\cdot|x_0^\theta)}}[-\alpha\|\delta_t(x_t^\theta)\|^2 + 2\alpha\langle\delta_t(x_t^\theta),f_t^*(x_t^\theta)\rangle - 2\beta\langle\delta_t(x_t^\theta),f_t^\theta(x_t^\theta|x_0^\theta)\rangle]$$

$$+ \mathbb{E}_{\substack{t\sim[0,T],x_0^*\sim p_0^*, \\ x_t^*\sim p_t^*(\cdot|x_t^*)}}[-(1-\alpha)\|\delta_t(x_t^*)\|^2 + 2(1-\alpha)\langle\delta_t(x_t^*),f_t^*(x_t^*)\rangle - 2(1-\beta)\langle\delta_t(x_t^*),f_t^*(x_t^*|x_0^*)\rangle],$$

*with the parameterization $\delta = f^* - f$.*

The idea of splitting coefficients between two data types helps to prove properties of RealUID, and extend our real data incorporation technique to general form (Appendix A.2), SiD framework with $\alpha_{\text{SiD}} \neq \frac{1}{2}$ (Appendix A.3) and new distances (Appendix A.4).

*Proof.* Putting explicit values for RealUM loss (16) in RealUID loss (17), we get:

$$\mathcal{L}_{R\text{-}UID}^{\alpha,\beta}(f,p_0^\theta) = \mathcal{L}_{R\text{-}UM}^{\alpha,\beta}(f^*,p_0^\theta) - \mathcal{L}_{R\text{-}UM}^{\alpha,\beta}(f,p_0^\theta)$$

$$= \alpha\cdot\mathbb{E}_{t,x_0^\theta,x_t^\theta}\left[\|f_t^*(x_t^\theta) - \frac{\beta}{\alpha}f^\theta(x_t^\theta|x_0^\theta)\|^2\right] + (1-\alpha)\cdot\mathbb{E}_{t,x_0^*,x_t^*}\left[\|f_t^*(x_t^*) - \frac{1-\beta}{1-\alpha}f_t^*(x_t^*|x_0^*)\|^2\right]$$

$$- \alpha\cdot\mathbb{E}_{t,x_0^\theta,x_t^\theta}\left[\|f_t(x_t^\theta) - \frac{\beta}{\alpha}f^\theta(x_t^\theta|x_0^\theta)\|^2\right] - (1-\alpha)\cdot\mathbb{E}_{t,x_0^*,x_t^*}\left[\|f_t(x_t^*) - \frac{1-\beta}{1-\alpha}f_t^*(x_t^*|x_0^*)\|^2\right].$$

Then, we group the factors with the same data type and multipliers:

$$\mathcal{L}_{R\text{-}UID}^{\alpha,\beta}(f,p_0^\theta) = \mathcal{L}_{R\text{-}UM}^{\alpha,\beta}(f^*,p_0^\theta) - \mathcal{L}_{R\text{-}UM}^{\alpha,\beta}(f,p_0^\theta)$$

$$= \mathbb{E}_{t,x_0^\theta,x_t^\theta}\left[\alpha\cdot\|f_t^*(x_t^\theta) - \frac{\beta}{\alpha}f^\theta(x_t^\theta|x_0^\theta)\|^2 - \alpha\cdot\|f_t(x_t^\theta) - \frac{\beta}{\alpha}f^\theta(x_t^\theta|x_0^\theta)\|^2\right]$$

$$+ \mathbb{E}_{t,x_0^*,x_t^*}\left[(1-\alpha)\cdot\|f_t^*(x_t^*) - \frac{1-\beta}{1-\alpha}f_t^*(x_t^*|x_0^*)\|^2 - (1-\alpha)\cdot\|f_t(x_t^*) - \frac{1-\beta}{1-\alpha}f_t^*(x_t^*|x_0^*)\|^2\right]$$

$$= \mathbb{E}_{t,x_0^\theta,x_t^\theta}\left[\alpha\cdot\|f_t^*(x_t^\theta)\|^2 - 2\beta\cdot\langle f_t^*(x_t^\theta),f^\theta(x_t^\theta|x_0^\theta)\rangle - \alpha\cdot\|f_t(x_t^\theta)\|^2 + 2\beta\cdot\langle f_t(x_t^\theta),f^\theta(x_t^\theta|x_0^\theta)\rangle\right]$$

$$+ \mathbb{E}_{t,x_0^*,x_t^*}\left[(1-\alpha)\cdot\|f_t^*(x_t^*)\|^2 - 2(1-\beta)\cdot\langle f_t^*(x_t^*) - f_t(x_t^*),f^*(x_t^*|x_0^*)\rangle - (1-\alpha)\cdot\|f_t(x_t^*)\|^2\rangle\right]$$

$$= \mathbb{E}_{t,x_0^\theta,x_t^\theta}\left[\alpha\cdot(\|f_t^*(x_t^\theta)\|^2 - \|f_t(x_t^\theta)\|^2) - 2\beta\cdot\langle f_t^*(x_t^\theta) - f_t(x_t^\theta),f^\theta(x_t^\theta|x_0^\theta)\rangle\right]$$

$$+ \mathbb{E}_{t,x_0^*,x_t^*}\left[(1-\alpha)\cdot(\|f_t^*(x_t^*)\|^2 - \|f_t(x_t^*)\|^2) - 2(1-\beta)\cdot\langle f_t^*(x_t^*) - f_t(x_t^*),f^*(x_t^*|x_0^*)\rangle\right]$$

$$= \mathbb{E}_{t,x_0^\theta,x_t^\theta}\left[\alpha\cdot(-\|f_t^*(x_t^\theta) - f_t(x_t^\theta)\|^2 + 2\langle f_t^*(x_t^\theta) - f_t(x_t^\theta),f_t^*(x_t^\theta)\rangle)\right]$$

$$- \mathbb{E}_{t,x_0^\theta,x_t^\theta}\left[2\beta\cdot\langle f_t^*(x_t^\theta) - f_t(x_t^\theta),f^\theta(x_t^\theta|x_0^\theta)\rangle\right]$$

$$+ \mathbb{E}_{t,x_0^*,x_t^*}\left[(1-\alpha)\cdot(-\|f_t^*(x_t^*) - f_t(x_t^*)\|^2 + 2\langle f_t^*(x_t^*) - f_t(x_t^*),f_t^*(x_t^*)\rangle)\right]$$

$$- \mathbb{E}_{t,x_0^*,x_t^*}\left[2(1-\beta)\cdot\langle f_t^*(x_t^*) - f_t(x_t^*),f^*(x_t^*|x_0^*)\rangle\right].$$

Finally, denoting parameterization $\delta = f^* - f$, we obtain the required form:

$$\mathcal{L}_{R\text{-}UID}^{\alpha,\beta}(f,p_0^\theta) = \mathbb{E}_{t,x_0^\theta,x_t^\theta}[-\alpha\|\delta_t(x_t^\theta)\|^2 + 2\alpha\langle\delta_t(x_t^\theta),f_t^*(x_t^\theta)\rangle - 2\beta\langle\delta_t(x_t^\theta),f_t^\theta(x_t^\theta|x_0^\theta)\rangle]$$

$$+ \mathbb{E}_{t,x_0^*,x_t^*}[-(1-\alpha)\|\delta_t(x_t^*)\|^2 + 2(1-\alpha)\langle\delta_t(x_t^*),f_t^*(x_t^*)\rangle - 2(1-\beta)\langle\delta_t(x_t^*),f_t^*(x_t^*|x_0^*)\rangle].$$

$\square$

### A.1.2 PROOF OF REALUID DISTANCE LEMMA 2

*Proof of Lemma 2.* In this proof, we use the split form of our ReaLUID loss from Lemma 3. First, we take math expectation over data points $x_0^*$. Since the expectation can be taken in a reverse order, i.e., $\mathbb{E}_{x_0^*\sim p_0^*,x_t^*\sim p_t^*(\cdot|x_t^*)} = \mathbb{E}_{x_t^*\sim p_t^*,x_0^*\sim p_0^*(\cdot|x_t^*)}$, we see that

$$\mathbb{E}_{x_0^*\sim p_0^*,x_t^*\sim p_t^*(\cdot|x_0^*)}[\langle\delta_t(x_t^*),f_t^*(x_t^*|x_0^*)\rangle] = \mathbb{E}_{x_t^*\sim p_t^*}[\langle\delta_t(x_t^*),\mathbb{E}_{x_0^*\sim p_0^*(\cdot|x_t^*)}[f_t^*(x_t^*|x_0^*)]\rangle]$$
$$= \mathbb{E}_{x_t^*\sim p_t^*}[\langle\delta_t(x_t^*),f_t^*(x_t^*)\rangle]. \quad (20)$$

For the generated data term $\mathbb{E}_{x_0^\theta\sim p_0^\theta,x_t^\theta\sim p_t^\theta(\cdot|x_0^\theta)}[\langle\delta_t(x_t^\theta),f_t^\theta(x_t^\theta|x_0^\theta)\rangle] = \mathbb{E}_{x_t^\theta\sim p_t^\theta}[\langle\delta_t(x_t^\theta),f_t^\theta(x_t^\theta)\rangle]$, the reasoning is similar. Thus, we can write down RealUID loss in an explicit form with $\delta_t = f_t^* - f_t$:

$$\mathcal{L}_{R\text{-}UID}^{\alpha,\beta}(\delta,p_0^\theta) = \mathbb{E}_{t\sim[0,T]}\mathbb{E}_{x_t^\theta\sim p_t^\theta}[-\alpha\|\delta_t(x_t^\theta)\|^2 + 2\alpha\langle\delta_t(x_t^\theta),f_t^*(x_t^\theta)\rangle - 2\beta\langle\delta_t(x_t^\theta),f_t^\theta(x_t^\theta)\rangle]$$

$$+\mathbb{E}_{t\sim[0,T]}\mathbb{E}_{x_t^*\sim p_t^*}[-(1-\alpha)\|\delta_t(x_t^*)\|^2 + 2(1-\alpha)\langle\delta_t(x_t^*), f_t^*(x_t^*)\rangle - 2(1-\beta)\langle\delta_t(x_t^*), f_t^*(x_t^*)\rangle]. \quad (21)$$

Then, we rescale the generated data terms in RealUID loss (21) using the equality $p_t^\theta(x_t) = \frac{p_t^\theta(x_t)}{p_t^*(x_t)}p_t^*(x_t)$ for $x_t \in \mathbb{R}^D$ (we assume $p_t^*(x_t) > 0, \forall x_t, t$) leaving only math expectation w.r.t. the real data, i.e,

$$\mathcal{L}_{\text{R-UID}}^{\alpha,\beta}(\delta, p_0^\theta) = \mathbb{E}_{\substack{t\sim[0,T]\\x_t^*\sim p_t^*}}\left[-[(1-\alpha) + \alpha\frac{p_t^\theta(x_t^*)}{p_t^*(x_t^*)}]\|\delta_t(x_t^*)\|^2\right]$$
$$- \mathbb{E}_{\substack{t\sim[0,T]\\x_t^*\sim p_t^*}}\left[2\beta\frac{p_t^\theta(x_t^*)}{p_t^*(x_t^*)}\langle\delta_t(x_t^*), f_t^\theta(x_t^*)\rangle + 2[(\beta-\alpha) + \alpha\frac{p_t^\theta(x_t^*)}{p_t^*(x_t^*)}]\langle\delta_t(x_t^*), f_t^*(x_t^*)\rangle\right].$$

Finally, we maximize the loss w.r.t. $\delta_t(x_t^*)$ for each $x_t^*$ and $t$ as a quadratic function. The maximum is achieved when

$$\delta_t(x_t^*) = \frac{[(\beta-\alpha) + \alpha\frac{p_t^\theta(x_t^*)}{p_t^*(x_t^*)}]f_t^*(x_t^*) - \beta\frac{p_t^\theta(x_t^*)}{p_t^*(x_t^*)}f_t^\theta(x_t^*)}{[(1-\alpha) + \alpha\frac{p_t^\theta(x_t^*)}{p_t^*(x_t^*)}]}$$

or in terms of the fake model $f = f^* - \delta$

$$\left(\arg\max_f \mathcal{L}_{\text{R-UID}}^{\alpha,\beta}(f, p_0^\theta)\right)(t, x_t) = \frac{f_t^*(x_t)\cdot(1-\beta) + f_t^\theta(x_t)\cdot\beta\frac{p_t^\theta(x_t)}{p_t^*(x_t)}}{(1-\alpha) + \alpha\frac{p_t^\theta(x_t)}{p_t^*(x_t)}}. \quad (22)$$

The maximum itself equals to

$$\max_f \mathcal{L}_{\text{R-UID}}^{\alpha,\beta}(f, p_0^\theta) = \mathbb{E}_{t\sim[0,T]}\mathbb{E}_{x_t^*\sim p_t^*}\left[\frac{\|f_t^*(x_t^*)\cdot((\beta-\alpha) + \alpha\frac{p_t^\theta(x_t^*)}{p_t^*(x_t^*)}) - f_t^\theta(x_t^*)\cdot\beta\frac{p_t^\theta(x_t^*)}{p_t^*(x_t^*)}\|^2}{(1-\alpha) + \alpha\frac{p_t^\theta(x_t^*)}{p_t^*(x_t^*)}}\right].$$

It is easy to see that when $p_0^\theta = p_0^*$ and $f^\theta = f^*$ this distance achieves its minimal value 0. Moreover, optimal fake model in this case matches the teacher $f^*$, i.e.,

$$\left(\arg\max_f \mathcal{L}_{\text{R-UID}}^{\alpha,\beta}(f, p_0^*)\right)(t, x_t) = \frac{f_t^*(x_t)\cdot(1-\beta) + f_t^*(x_t)\cdot\beta\frac{p_t^*(x_t)}{p_t^*(x_t)}}{(1-\alpha) + \alpha\frac{p_t^*(x_t)}{p_t^*(x_t)}} = f_t^*(x_t).$$

$\square$

### A.1.3 EXPLANATION OF THE CHOICE OF COEFFICIENTS $\alpha$ AND $\beta$

Here we show that the best way to incorporate real data during generator training is to set $\beta/\alpha \neq 1$.

Following Lemma 2, we know exactly what distance our RealUID loss implicitly minimizes. Below we examine it for various $\alpha, \beta \in (0, 1]$:

$$\max_f \mathcal{L}_{\text{R-UID}}^{\alpha,\beta}(f, p_0^\theta) = \int_{x_t} l_t(x_t, \beta, \alpha)dx_t,$$
$$l_t(x_t, \beta, \alpha) := \frac{\alpha^2\|(p_t^*(x_t)(\frac{\beta}{\alpha}-1) + p_t^\theta(x_t))\cdot f_t^*(x_t) - \frac{\beta}{\alpha}\cdot p_t^\theta(x_t)\cdot f_t^\theta(x_t)\|^2}{(1-\alpha)p_t^*(x_t) + \alpha p_t^\theta(x_t)},$$

where $l_t(x_t, \beta, \alpha)$ denotes the distance for the particular point $x_t$.

The total distance mostly sums up from the two groups of points: incorrectly generated points from the generator's main domain, i.e., $p_t^\theta(x_t) \gg 0, p^*(x_t) \approx 0$, and real data points which are not covered by the generator, i.e., $p_t^\theta(x_t) \approx 0, p^*(x_t) \gg 0$. For the points out of both domains $p_t^\theta(x_t) \approx 0, p_t^*(x_t) \approx 0$, the distance tends to 0, as well as for matching points $p_t^\theta(x_t) \approx p_t^*(x_t)$.

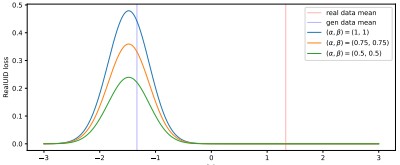 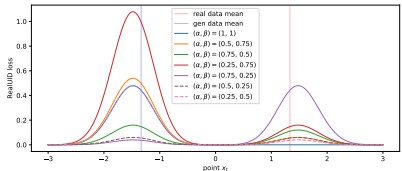

Figure 3: RealUID loss for $1D$-Gaussians under various coefficients $(\alpha, \beta)$.

**Choice of coefficients** $\alpha, \beta$. Next, we consider various coefficients $\alpha, \beta \in (0, 1]$ and how they affect two main groups of points.

- All configurations affect the incorrectly generated points $x_t : p_t^*(x_t) \approx 0, p^\theta(x_t) \gg 0$:

$$l_t(x_t, \beta, \alpha) \approx \frac{\|\alpha p_t^\theta(x_t) \cdot f_t^*(x_t) - \beta p_t^\theta(x_t) \cdot f_t^\theta(x_t)\|^2}{\alpha p_t^\theta(x_t)} \approx \frac{\beta^2 \|f_t^\theta(x_t)\|^2}{\alpha} p_t^\theta(x_t) \gg 0. \qquad (23)$$

Note that increasing $\beta/\alpha > 1$ will diminish the weight of the distance in comparison with $\alpha = \beta = 1$, while decreasing otherwise will lift the weight up.

- Configuration $\beta < \alpha = 1$ is unstable for uncovered real data points $x_t : p_t^\theta(x_t) \approx 0, p^*(x_t) \gg 0$:

$$l_t(x_t, \beta, \alpha) \approx \frac{\|p_t^*(x_t)(\beta - 1) \cdot f_t^*(x_t) - \beta p_t^\theta(x_t) \cdot f_t^\theta(x_t)\|^2}{p_t^\theta(x_t)} \approx \infty.$$

- Configuration $\beta = \alpha = 1$ (UID loss) does not affect uncovered real data points $x_t : p_t^\theta(x_t) \approx 0, p^*(x_t) \gg 0$:

$$l_t(x_t, \beta, \alpha) \approx \frac{\|p_t^\theta(x_t) \cdot f_t^*(x_t) - p_t^\theta(x_t) \cdot f_t^\theta(x_t)\|^2}{p_t^\theta(x_t)} = \|f_t^*(x_t) - f_t^\theta(x_t)\|^2 p_t^\theta(x_t) \approx 0.$$

- Configuration $\beta = \alpha < 1$ does not affect uncovered real data points $x_t : p_t^\theta(x_t) \approx 0, p^*(x_t) \gg 0$:

$$l_t(x_t, \beta, \alpha) \approx \frac{\|\alpha p_t^\theta(x_t) f_t^*(x_t) - \beta p_t^\theta(x_t) f_t^\theta(x_t)\|^2}{(1 - \alpha) p_t^*(x_t)} = \frac{\|\alpha f_t^*(x_t) - \beta f_t^\theta(x_t)\|^2}{(1 - \alpha)} \frac{(p_t^\theta(x_t))^2}{p_t^*(x_t)} \approx 0.$$

Notably, in this configuration, the distance drops even faster than when $\alpha = \beta = 1$, what makes it even less preferable.

- *Only configuration $\beta/\alpha \neq 1$ affects the uncovered real data points $x_t : p_t^\theta(x_t) \approx 0, p^*(x_t) \gg 0$:*

$$l_t(x_t, \beta, \alpha) \approx \frac{\|p_t^*(x_t)(\beta - \alpha) \cdot f_t^*(x_t) - \beta p_t^\theta(x_t) \cdot f_t^\theta(x_t)\|^2}{(1 - \alpha) p_t^*(x_t)} \gg 0.$$

**Visual illustration.** We analytically calculate the loss surface $l_t(x_t, \alpha, \beta)$ between the FM models transforming one-dimensional real data Gaussian $\mathcal{N}(\mu^*, 1)$ and generated Gaussian $\mathcal{N}(\mu^\theta, 1)$ to noise $\mathcal{N}(0, 1)$ on the time interval $[0, 1]$. In this case, the generated and real data interpolations are $p_t^\theta(x_t) = \mathcal{N}(x_t | \mu^\theta(1 - t), t^2 + (1 - t)^2)$ and $p_t^*(x_t) = \mathcal{N}(x_t | \mu^*(1 - t), t^2 + (1 - t)^2)$. The unconditional vector field $u = f$ between $\mathcal{N}(\mu, 1)$ and $\mathcal{N}(0, 1)$ can be calculated as

$$u_t(x_t) = \mathbb{E}_{x_0 \sim p_0(\cdot | x_t)} \left[ \frac{x_t - x_0}{t} \right] = \int_{x_0} \left( \frac{x_t - x_0}{t} \right) \cdot \mathcal{N}\left( \frac{x_t - x_0(1 - t)}{t} | 0, 1 \right) \cdot \mathcal{N}(x_0 | \mu, 1) dx_0$$

$$= \frac{a(2t^2 - 2t) - bt^2}{\sqrt{2\pi}(1 - 2t + 2t^2)^{\frac{3}{2}}} \exp\left( -\frac{(x_t - \mu(1 - t))^2}{2(1 - 2t + 2t^2)^2} \right). \qquad (24)$$

In Figure 3, we depict the loss surfaces for the fixed time $t = 1/3$, real data $\mu^* = 2$, generated data $\mu^\theta = -2$ and various pairs of $(\alpha, \beta)$. We can see that configurations $\beta/\alpha = 1$ do not detect the real data sample, even when $\alpha = \beta < 1$ and real data is formally used. while $\beta/\alpha \neq 1$ actually spots both domains, increasing the weight of generator domain when $\beta/\alpha > 1$ and decreasing it otherwise.

A.1.4 CORRECTION OF TEACHER'S ERRORS

In this chapter, we assume that instead of accurate teacher $f^* = \arg\min_f \mathcal{L}_{\text{UM}}(f, p_0^*)$ we have access only to the arbitrary corrupted teacher $\tilde{f}^*$. *We will show that adding real data via our approach with $\alpha \neq \beta$ provably mitigates the teacher's errors in the final generator.*

**Minimized distance.** With the corrupted teacher $\tilde{f}^*$ and $\tilde{\delta} = \tilde{f}^* - f$, our corrupted RealUID loss takes the split form from Lemma 3

$$\mathcal{L}_{\text{R-UID}}^{\alpha,\beta}(\tilde{\delta}, p_0^\theta) = \mathbb{E}_{\substack{t\sim[0,T], x_0^\theta \sim p_0^\theta, \\ x_t^\theta \sim p_t^\theta(\cdot|x_0^\theta)}}[-\alpha\|\tilde{\delta}_t(x_t^\theta)\|^2 + 2\alpha\langle\tilde{\delta}_t(x_t^\theta), \tilde{f}_t^*(x_t^\theta)\rangle - 2\beta\langle\tilde{\delta}_t(x_t^\theta), f_t^\theta(x_t^\theta|x_0^\theta)\rangle]$$

$$+\mathbb{E}_{\substack{t\sim[0,T], x_0^*\sim p_0^*, \\ x_t^*\sim p_t^*(\cdot|x_t^*)}}[-(1-\alpha)\|\tilde{\delta}_t(x_t^*)\|^2 + 2(1-\alpha)\langle\tilde{\delta}_t(x_t^*), \tilde{f}_t^*(x_t^*)\rangle - 2(1-\beta)\langle\tilde{\delta}_t(x_t^*), f_t^*(x_t^*|x_0^*)\rangle].$$

Note that sampled terms $f_t^*(x_t^*|x_0^*)$ and $f_t^\theta(x_t^\theta|x_0^\theta)$ are not affected by the corruption and give the accurate functions $f_t^*(x_t^*) = \mathbb{E}_{x_0^*\sim p_0^*(\cdot|x_t^*)}[f_t^*(x_t^*|x_0^*)]$ and $f_t^\theta(x_t^\theta) = \mathbb{E}_{x_0^\theta\sim p_0^\theta(\cdot|x_t^\theta)}[f_t^\theta(x_t^\theta|x_0^\theta)]$:

$$\mathcal{L}_{\text{R-UID}}^{\alpha,\beta}(\tilde{\delta}, p_0^\theta) = \mathbb{E}_{t\sim[0,T]}\mathbb{E}_{x_t^\theta\sim p_t^\theta}[-\alpha\|\tilde{\delta}_t(x_t^\theta)\|^2 + 2\alpha\langle\tilde{\delta}_t(x_t^\theta), \tilde{f}_t^*(x_t^\theta)\rangle - 2\beta\langle\delta_t(x_t^\theta), f_t^\theta(x_t^\theta)\rangle]$$

$$+\mathbb{E}_{t\sim[0,T]}\mathbb{E}_{x_t^*\sim p_t^*}[-(1-\alpha)\|\tilde{\delta}_t(x_t^*)\|^2 + 2(1-\alpha)\langle\tilde{\delta}_t(x_t^*), \tilde{f}_t^*(x_t^*)\rangle - 2(1-\beta)\langle\tilde{\delta}_t(x_t^*), f_t^*(x_t^*)\rangle].$$

Then, we rescale the generated data terms using the equality $p_t^\theta(x_t) = \frac{p_t^\theta(x_t)}{p_t^*(x_t)}p_t^*(x_t)$ for $x_t \in \mathbb{R}^D$ (we assume $p_t^*(x_t) > 0, \forall x_t, t$) leaving only math expectation w.r.t. the real data, i.e,

$$\mathcal{L}_{\text{R-UID}}^{\alpha,\beta}(\tilde{\delta}, p_0^\theta) = \mathbb{E}_{t\sim[0,T], x_t^*\sim p_t^*}\left[-[(1-\alpha) + \alpha\frac{p_t^\theta(x_t^*)}{p_t^*(x_t^*)}](\|\tilde{\delta}_t(x_t^*)\|^2 + 2\langle\tilde{\delta}_t(x_t^*), \tilde{f}_t^*(x_t^*)\rangle)\right]$$

$$- \mathbb{E}_{\substack{t\sim[0,T], \\ x_t^*\sim p_t^*}}\left[2\langle\tilde{\delta}_t(x_t^*), (1-\beta)f_t^*(x_t^*) + \beta\frac{p_t^\theta(x_t^*)}{p_t^*(x_t^*)}f_t^\theta(x_t^*)\rangle\right].$$

Finally, we maximize the loss w.r.t. $\tilde{\delta}_t(x_t^*)$ for each $x_t^*$ and $t$ as a quadratic function $\max_{\tilde{\delta}} \mathcal{L}_{\text{R-UID}}^{\alpha,\beta}(\tilde{\delta}, p_0^\theta) =$

$$\mathbb{E}_{t\sim[0,T]}\mathbb{E}_{x_t^*\sim p_t^*}\left[\frac{\|\tilde{f}_t^*(x_t^*) \cdot ((1-\alpha) + \alpha\frac{p_t^\theta(x_t^*)}{p_t^*(x_t^*)}) - (1-\beta)f_t^*(x_t^*) - \beta\frac{p_t^\theta(x_t^*)}{p_t^*(x_t^*)}f_t^\theta(x_t^*)\|^2}{(1-\alpha) + \alpha\frac{p_t^\theta(x_t^*)}{p_t^*(x_t^*)}}\right]. \quad (25)$$

Hence, max-min optimization of the corrupted RealUID loss implicitly minimizes expected distance (25). However, due to arbitrary function $\tilde{f}$, we now cannot guarantee that minimum is achived when the relation inside the norm equals 0. Previously, we could use the solution $p^\theta = p^*$ which obviously achieved a minimum of 0. Now, due to the implicit and complex relationship between $f^\theta$ and $p^\theta$, we can neither find an explicit form for the optimal $p^\theta$ nor guarantee the minimum of 0.

**Choice of coefficients $\alpha, \beta$.** Here we give an intuition on why coefficients $\beta/\alpha \neq 1$ can fix the teacher's errors, while $\beta/\alpha = 1$ cannot. For simplicity, we assume that the minimized distance (25) actually attains minimum of 0 when

$$((1-\alpha)p_t^*(x_t) + \alpha p_t^\theta(x_t)) \cdot \tilde{f}_t^*(x_t) - (1-\beta)p_t^*(x_t) \cdot f_t^*(x_t) - \beta p_t^\theta(x_t) \cdot f_t^\theta(x_t^*) = 0. \quad (26)$$

- In case of $\alpha = \beta = 1$, we have $\tilde{f}_t^* = f_t^\theta$, i.e., the generator learns the corrupted function.

- In case of $\alpha = \beta < 1$, we have

$$\tilde{f}_t^*(x_t) = \frac{(1-\alpha)p_t^*(x_t)}{(1-\alpha)p_t^*(x_t) + \alpha p_t^\theta(x_t)} \cdot f_t^*(x_t) + \frac{\alpha p_t^\theta(x_t)}{(1-\alpha)p_t^*(x_t) + \alpha p_t^\theta(x_t)} \cdot f_t^\theta(x_t^*).$$

In this convex combination, the corrupted function $\tilde{f}^*$ is always between the true teacher function $f^*$ and the optimal generator function $f^\theta$, i.e., the generator learns even worse function.

- *In case of $\beta/\alpha \neq 1$, there exist intervals of $\alpha, \beta$ which can give better generator function than the corrupted teacher.* For example, coefficients $\alpha \neq \beta$ close to 1 allow to neglect the terms $(1 - \alpha)p_t^*(x_t) \cdot \tilde{f}_t^*(x_t)$ and $(1 - \beta)p_t^*(x_t) \cdot f_t^*(x_t)$ in (26) to get $f_t^\theta(x_t) \approx \frac{\alpha}{\beta}\tilde{f}_t^*(x_t)$. Hence, we can steer $f^\theta$ towards the true teacher picking $\beta/\alpha < 1$ or $\beta/\alpha > 1$ depending on the corrupted and clean teacher's values. However, we cannot find all these intervals analytically due to complex distributions and functions.

Note that we derive the same recommendation $\beta/\alpha \neq 1$ from the perspective of correcting the teacher's errors and from the perspective of the minimized distance surface from Appendix A.1.3.

**Visual illustration.** For visual demonstration, we consider the FM models transforming one-dimensional real data Gaussian $\mathcal{N}(\mu^*, 1)$ and generated Gaussian $\mathcal{N}(\mu^\theta, 1)$ to noise $\mathcal{N}(0, 1)$ on the time interval $[0, 1]$. In this case, the generated and real data interpolations are $p_t^\theta(x_t) = \mathcal{N}(x_t|\mu^\theta(1 - t), t^2 + (1 - t)^2)$ and $p_t^*(x_t) = \mathcal{N}(x_t|\mu^*(1 - t), t^2 + (1 - t)^2)$. The unconditional vector field $u = f$ between $\mathcal{N}(\mu, 1)$ and $\mathcal{N}(0, 1)$ can be calculated as

$$u_t(x_t) = \mathbb{E}_{x_0 \sim p_0(\cdot|x_t)}\left[\frac{x_t - x_0}{t}\right] = \int_{x_0}\left(\frac{x_t - x_0}{t}\right) \cdot \mathcal{N}\left(\frac{x_t - x_0(1 - t)}{t}\Big|0, 1\right) \cdot \mathcal{N}(x_0|\mu, 1)dx_0$$

$$= \frac{a(2t^2 - 2t) - bt^2}{\sqrt{2\pi}(1 - 2t + 2t^2)^{\frac{3}{2}}}\exp\left(-\frac{(x_t - \mu(1 - t))^2}{2(1 - 2t + 2t^2)^2}\right). \tag{27}$$

In Figure 4, we depict the optimal generator mean $\mu^\theta$ and vector field $u^\theta$ satisfying (26) for various deviations $\tilde{u}^* - u^*$ and fixed time $t = 1/3$, real data $\mu^* = -2$ and point $x_t = -1$.

We can see that with $\alpha = \beta = 1$, the generator learns the corrupted vector field, and with $\alpha = \beta < 1$, the learned field and means are often even worse. In contrast, with $\beta/\alpha \neq 1$, the generator can learn vector fields and means which are closer to the real data. Although the generator cannot satisfy relation (26) under large deviations, it still produces better results with the real data.

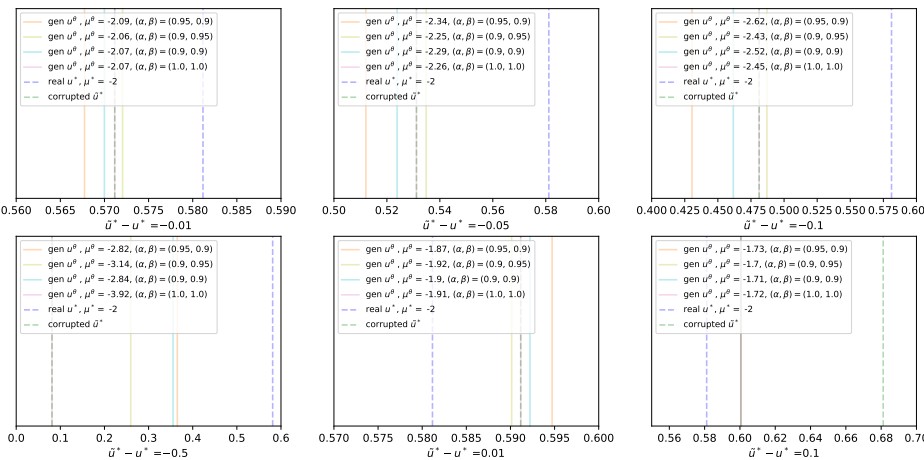

Figure 4: Learned generators for RealUID loss between 1D-Gaussians with corrupted teachers.

## A.2 GENERAL REALUID LOSS

**Extending our real data incorporation.** We recall that UID loss (Theorem 1) can be restated via linearization technique with $\delta = f^* - f$ as:

$$\mathcal{L}_{\text{UID}}(\delta, p_0^\theta) = \mathbb{E}_{\substack{t \sim [0,T], x_0^\theta \sim p_0^\theta, \\ x_t^\theta \sim p_t^\theta(\cdot|x_0^\theta)}}\left\{-\|\delta_t(x_t^\theta)\|^2 + 2\langle\delta_t(x_t^\theta), f_t^*(x_t^\theta)\rangle - 2\langle\delta_t(x_t^\theta), f_t^\theta(x_t^\theta|x_0^\theta)\rangle\right\}.$$

Following alternative definition of RealUID loss from Lemma 3, one can incorporate real data into data-free loss by splitting each term in the linearized form between generated and real data as:

$$\mathcal{L}_{\text{R-UID}}^{\alpha,\beta}(\delta, p_0^\theta) = \mathbb{E}_{\substack{t\sim[0,T], x_0^\theta \sim p_0^\theta, \\ x_t^\theta \sim p_t^\theta(\cdot|x_0^\theta)}} [-\alpha\|\delta_t(x_t^\theta)\|^2 + 2\alpha\langle\delta_t(x_t^\theta), f_t^*(x_t^\theta)\rangle - 2\beta\langle\delta_t(x_t^\theta), f_t^\theta(x_t^\theta|x_0^\theta)\rangle]$$

$$+ \mathbb{E}_{\substack{t\sim[0,T], x_0^*\sim p_0^*, \\ x_t^*\sim p_t^*(\cdot|x_0^*)}} [-(1-\alpha)\|\delta_t(x_t^*)\|^2 + 2(1-\alpha)\langle\delta_t(x_t^*), f_t^*(x_t^*)\rangle - 2(1-\beta)\langle\delta_t(x_t^*), f_t^*(x_t^*|x_0^*)\rangle].$$

In RealUID loss (17), its three terms are split with proportions $\alpha$ and $1-\alpha$, $\alpha$ and $1-\alpha$ and $\beta$ and $1-\beta$, respectively. We can go even further and split the first quadratic coefficient $-\|\delta_t(\cdot)\|^2$ using a new parameter $\gamma$ to create one more degree of freedom. Moreover, we can use other parameterization of $\delta$, since its form does not change the proof of distance lemma.

**Definition 3.** *We introduce **General RealUID loss** $\mathcal{L}_{R\text{-}UID}^{\alpha,\beta,\gamma}(\delta, p_0^\theta)$ on generated data $p_0^\theta \in \mathcal{P}(\mathbb{R}^D)$ with coefficients $\alpha, \beta, \gamma$:*

$$\mathcal{L}_{R\text{-}UID}^{\alpha,\beta,\gamma}(\delta, p_0^\theta) := \mathbb{E}_{\substack{t\sim[0,T], x_0^\theta\sim p_0^\theta, \\ x_t^\theta\sim p_t^\theta(\cdot|x_0^\theta)}} [-\gamma\|\delta_t(x_t^\theta)\|^2 + 2\alpha\langle\delta_t(x_t^\theta), f_t^*(x_t^\theta)\rangle - 2\beta\langle\delta_t(x_t^\theta), f_t^\theta(x_t^\theta|x_0^\theta)\rangle]$$

$$+ \mathbb{E}_{\substack{t\sim[0,T], x_0^*\sim p_0^*, \\ x_t^*\sim p_t^*(\cdot|x_0^*)}} [-(1-\gamma)\|\delta_t(x_t^*)\|^2 + 2(1-\alpha)\langle\delta_t(x_t^*), f_t^*(x_t^*)\rangle - 2(1-\beta)\langle\delta_t(x_t^*), f_t^*(x_t^*|x_0^*)\rangle].$$

*Optionally, one can change default parameterization $\delta = f^* - f$ (e.g., with $\delta = \beta(f^* - f)$), and substitute sampled real data term $f_t^*(x_t^*|x_0^*)$ with the unconditional teacher $f_t^*(x_t^*)$ and vice versa.*

**Theoretical properties.** In case of $\delta = f^* - f$ and $\gamma \neq \alpha$, the General RealUID loss cannot be expressed as inverse min-max problem (15) for simple losses, since some scalar products do not eliminate each other. Nevertheless, min-max optimization of $\mathcal{L}_{\text{R-UID}}^{\alpha,\beta,\gamma}$ still minimizes the squared $\ell_2$-distance between the weighted teacher and generator functions, attaining minimum when $p_0^\theta = p_0^*$.

**Lemma 4** (**Distance minimized by General RealUID loss**). *Maximization of General RealUID loss $\mathcal{L}_{R\text{-}UID}^{\alpha,\beta,\gamma}$ over $\delta$ represents the weighted squared $\ell_2$-distance between the student function $f^\theta := \arg\min_f \mathcal{L}_{UM}(f, p_0^\theta)$ and the teacher $f^* := \arg\min_f \mathcal{L}_{UM}(f, p_0^*)$:*

$$\max_\delta \mathcal{L}_{R\text{-}UID}^{\alpha,\beta,\gamma}(\delta, p_0^\theta) = \mathbb{E}_{\substack{t\sim[0,T], \\ x_t^*\sim p_t^*}} \left[ \frac{\|\frac{\beta}{\alpha}[p_t^*(x_t^*)f_t^*(x_t^*) - p_t^\theta(x_t^*)f_t^\theta(x_t^*)] + (p_t^\theta(x_t^*) - p_t^*(x_t^*))f_t^*(x_t^*)\|^2}{p_t^*(x_t^*) \cdot \max\{0, (1-\gamma)p_t^*(x_t^*) + \gamma p_t^\theta(x_t^*)\}/\alpha^2} \right].$$

$$(28)$$

The distances being minimized for RealUID (Lemma 2) and General RealUID (Lemma 4) are almost identical except the scale factor. Thus, we keep the same recommendations for choosing coefficients $\alpha, \beta$ as we discuss in Section 3.4. The factor $\beta/\alpha$ still has the largest impact within the distance, while $\alpha$ and $\gamma$ set the scaling. Values $\beta/\alpha$ and $\gamma$ should be chosen close to 1, but not exactly 1.

*Proof.* First, we take math expectation over data points $x_0^*$. Since the expectation can be taken in a reverse order, i.e., $\mathbb{E}_{x_0^*\sim p_0^*, x_t^*\sim p_t^*(\cdot|x_0^*)} = \mathbb{E}_{x_t^*\sim p_t^*, x_0^*\sim p_0^*(\cdot|x_t^*)}$, we see that

$$\mathbb{E}_{x_0^*\sim p_0^*, x_t^*\sim p_t^*(\cdot|x_0^*)} [\langle\delta_t(x_t^*), f_t^*(x_t^*|x_0^*)\rangle] = \mathbb{E}_{x_t^*\sim p_t^*} \langle\delta_t(x_t^*), \mathbb{E}_{x_0^*\sim p_0^*(\cdot|x_t^*)}[f_t^*(x_t^*|x_0^*)]\rangle$$

$$= \mathbb{E}_{x_t^*\sim p_t^*} [\langle\delta_t(x_t^*), f_t^*(x_t^*)\rangle]. \qquad (29)$$

For the term $\mathbb{E}_{x_0^\theta\sim p_0^\theta, x_t^\theta\sim p_t^\theta(\cdot|x_0^\theta)}[\langle\delta_t(x_t^\theta), f_t^\theta(x_t^\theta|x_0^\theta)\rangle] = \mathbb{E}_{x_t^\theta\sim p_t^\theta}[\langle\delta_t(x_t^\theta), f_t^\theta(x_t^\theta)\rangle]$, the reasoning is similar. Thus, we write down General RealUID loss (Def. 3) in an explicit form with $\delta_t = f_t^* - f_t$

$$\mathcal{L}_{\text{R-UID}}^{\alpha,\beta}(\delta, p_0^\theta) = \mathbb{E}_{t\sim[0,T]}\mathbb{E}_{x_t^\theta\sim p_t^\theta}[-\gamma\|\delta_t(x_t^\theta)\|^2 + 2\alpha\langle\delta_t(x_t^\theta), f_t^*(x_t^\theta)\rangle - 2\beta\langle\delta_t(x_t^\theta), f_t^\theta(x_t^\theta)\rangle]$$

$$+\mathbb{E}_{t\sim[0,T]}\mathbb{E}_{x_t^*\sim p_t^*}[-(1-\gamma)\|\delta_t(x_t^*)\|^2 + 2(1-\alpha)\langle\delta_t(x_t^*), f_t^*(x_t^*)\rangle - 2(1-\beta)\langle\delta_t(x_t^*), f_t^*(x_t^*)\rangle].$$

Then, we rescale the generated data terms in the General RealUID loss using the equality $p_t^\theta(x_t) = \frac{p_t^\theta(x_t)}{p_t^*(x_t)}p_t^*(x_t)$ for $x_t \in \mathbb{R}^D$ (we assume $p_t^*(x_t) > 0, \forall x_t, t$) leaving only math expectation w.r.t. the real data, i.e,

$$\mathcal{L}_{\text{R-UID}}^{\alpha,\beta,\gamma}(\delta, p_0^\theta) = \mathbb{E}_{\substack{t\sim[0,T], \\ x_t^*\sim p_t^*}} \left[ -[(1-\gamma) + \gamma\frac{p_t^\theta(x_t^*)}{p_t^*(x_t^*)}]\|\delta_t(x_t^*)\|^2 \right.$$

$$+ \mathbb{E}_{\substack{t \sim [0,T], \\ x_t^* \sim p_t^*}} \left[ 2[(\beta - \alpha) + \alpha \frac{p_t^\theta(x_t^*)}{p_t^*(x_t^*)}] \langle \delta_t(x_t^*), f_t^*(x_t^*) \rangle - 2\beta \frac{p_t^\theta(x_t^*)}{p_t^*(x_t^*)} \langle \delta_t(x_t^*), f_t^\theta(x_t^*) \rangle \right].$$

Next we maximize the loss w.r.t. $\delta_t(x_t^*)$ for each $x_t^*$ and $t$ as a quadratic function. If $(1 - \gamma) \cdot p_t^*(x_t^*) + \gamma \cdot p_t^\theta(x_t^*) \leq 0$, then the maximum tends to $+\infty$. Otherwise, the maximum is achieved when

$$\delta_t(x_t^*) = \frac{[(\beta - \alpha) + \alpha \frac{p_t^\theta(x_t^*)}{p_t^*(x_t^*)}] f_t^*(x_t^*) - \beta \frac{p_t^\theta(x_t^*)}{p_t^*(x_t^*)} f_t^\theta(x_t^*)}{[(1 - \gamma) + \gamma \frac{p_t^\theta(x_t^*)}{p_t^*(x_t^*)}]}. \tag{30}$$

The maximum itself equals to

$$\max_{\delta} \mathcal{L}_{\text{R-UID}}^{\alpha, \beta, \gamma}(\delta, p_0^\theta) = \mathbb{E}_{t \sim [0,T]} \mathbb{E}_{x_t^* \sim p_t^*} \left[ \frac{\| f_t^*(x_t^*) \cdot ((\beta - \alpha) + \alpha \frac{p_t^\theta(x_t^*)}{p_t^*(x_t^*)}) - f_t^\theta(x_t^*) \cdot \beta \frac{p_t^\theta(x_t^*)}{p_t^*(x_t^*)} \|^2}{(1 - \gamma) + \gamma \frac{p_t^\theta(x_t^*)}{p_t^*(x_t^*)}} \right].$$

$\square$

**Alternative parameterization.** In the proximity of the solution, when generated data approaches real one, i.e., $p_t^\theta \approx p_t^*$, the optimal $\delta_t$ (30) approaches

$$\delta_t(x_t^*) \approx \frac{[(\beta - \alpha) + \alpha \cdot 1] f_t^*(x_t^*) - \beta \cdot 1 \cdot f_t^\theta(x_t^*)}{[(1 - \gamma) + \gamma \cdot 1]} \approx \beta(f_t^*(x_t^*) - f_t^\theta(x_t^*)).$$

Thus, the parameterization $\delta_t = \beta(f_t^* - f_t)$ may naturally help reach the solution without making the fake model learn extra information about the teacher near the optimum.

In experiments in Tables 1 and 7, this parameterization with the corresponding coefficients $\gamma = \alpha$ and $\beta$ yields slightly better metrics from +0.02 to +0.04.

**Extra ranges for coefficients** $\alpha, \beta, \gamma$   New perspective on our RealUID loss allows us to expand the range of feasible configurations for the parameters $\alpha$, $\beta$, and $\gamma$. Specifically, it is now possible to set $\alpha = 1$ for any $\beta$, whereas in the original loss (16) this configuration is unavailable due to division by zero in the real data term. Additionally, one can now use values $\alpha, \beta, \gamma > 1$.

However, we observe that in the experiments reported in Tables 1 and 7, these extra configurations are highly unstable and lead to degraded results. This degradation occurs due to out-of-domain generated samples and negative quadratic summands leading to infinite losses and metric (28). Hence, we stick to the original ranges $\alpha, \beta, \gamma \in (0, 1]$.

### A.3   GENERAL SID WITH REAL DATA

**Our real data incorporation.** We recall that data-free UID loss (Theorem 1) can be restated via linearization technique with $\delta = f - f^*$ as:

$$\mathcal{L}_{\text{UID}}(\delta, p_0^\theta) = \mathbb{E}_{\substack{t \sim [0,T], x_0^\theta \sim p_0^\theta, \\ x_t^\theta \sim p_t^\theta(\cdot | x_0^\theta)}} \left[ -\| \delta_t(x_t^\theta) \|^2 + 2\langle \delta_t(x_t^\theta), f_t^*(x_t^\theta) \rangle - 2\langle \delta_t(x_t^\theta), f_t^\theta(x_t^\theta | x_0^\theta) \rangle \right]. \tag{31}$$

Following alternative definition of our RealUID loss from Lemma 3, one can incorporate real data into data-free loss by splitting each term in the linearized form between generated and real data as:

$$\mathcal{L}_{\text{R-UID}}^{\alpha, \beta}(\delta, p_0^\theta) = \mathbb{E}_{\substack{t \sim [0,T], x_0^\theta \sim p_0^\theta, \\ x_t^\theta \sim p_t^\theta(\cdot | x_0^\theta)}} [-\alpha \| \delta_t(x_t^\theta) \|^2 + 2\alpha \langle \delta_t(x_t^\theta), f_t^*(x_t^\theta) \rangle - 2\beta \langle \delta_t(x_t^\theta), f_t^\theta(x_t^\theta | x_0^\theta) \rangle]$$

$$+ \mathbb{E}_{\substack{t \sim [0,T], x_0^* \sim p_0^*, \\ x_t^* \sim p_t^*(\cdot | x_0^*)}} [-(1 - \alpha) \| \delta_t(x_t^*) \|^2 + 2(1 - \alpha) \langle \delta_t(x_t^*), f_t^*(x_t^*) \rangle - 2(1 - \beta) \langle \delta_t(x_t^*), f_t^*(x_t^* | x_0^*) \rangle]. \tag{32}$$

**General data-free SiD.** The authors of the SiD framework (Zhou et al., 2024a;b) for diffusion models empirically notice that scaling the first coefficient $-\| \delta_t(x_t^\theta) \|^2$ by the factor $2\alpha_{\text{SiD}}$ in the UID loss (31) for generator updates yields better performance. Hence, we generalize the SiD loss to

other matching models. Namely, the **General SiD loss** for the generator is the following loss with $\delta = f - f^*$ and parameter $\alpha_{\text{SiD}} \in [0.5, 1.2]$:

$$\mathcal{L}_{\text{SiD}}(\theta) := \mathbb{E}_{\substack{t \sim [0,T], x_0^\theta \sim p_0^\theta, \\ x_t^\theta \sim p_t^\theta(\cdot | x_0^\theta)}} \left[ -2\alpha_{\text{SiD}} \|\delta_t(x_t^\theta)\|^2 + 2\langle \delta_t(x_t^\theta), f_t^*(x_t^\theta) \rangle - 2\langle \delta_t(x_t^\theta), f^\theta(x_t^\theta | x_0^\theta) \rangle \right], \tag{33}$$

while the UM loss (Def. 1) for the fake model remains intact. The same positive effect is observed in experiments with flow matching models in FGM (Huang et al., 2024), where the authors do not calculate the gradient through some loss terms and obtain the General SiD loss (33) with $\alpha_{\text{SiD}} = 1$, achieving better performance.

**General SiD with real data.** Following the structure of the General SiD loss (33), we propose to scale the first coefficient in our RealUID loss (32) during generator updates. The whole **General SiD pipeline with real data (RealSiD)**, defined by coefficients $\alpha, \beta \in (0, 1], \alpha_{\text{SiD}} \in [0.5, 1.2]$ and teacher $f^*$, is two alternating steps:

1. Make one or several fake model $f$ update steps, minimizing UM loss with real data $\mathcal{L}_{\text{R-UM}}^{\alpha,\beta}(f, p_0^\theta)$:

$$L_{\text{R-UM}}^{\alpha,\beta}(f, p_0^\theta) \quad := \quad \underbrace{\alpha \cdot \mathbb{E}_{t \sim [0,T]} \mathbb{E}_{x_0^\theta \sim p_0^\theta, x_t^\theta \sim p_t^\theta(\cdot | x_0^\theta)} \left[ \|f_t(x_t^\theta) - \frac{\beta}{\alpha} f_t^\theta(x_t^\theta | x_0^\theta)\|^2 \right]}_{\text{generated data } p_0^\theta \text{ term}}$$

$$+ \quad \underbrace{(1 - \alpha) \cdot \mathbb{E}_{t \sim [0,T]} \mathbb{E}_{x_0^* \sim p_0^*, x_t^* \sim p_t^*(\cdot | x_0^*)} \left[ \|f_t(x_t^*) - \frac{1 - \beta}{1 - \alpha} f_t^*(x_t^* | x_0^*)\|^2 \right]}_{\text{real data } p_0^* \text{ term}}.$$

2. Make a generator update step, minimizing the loss $\mathcal{L}_{\text{R-SiD}}^{\alpha,\beta}(\theta) :=$

$$\mathbb{E}_{\substack{t \sim [0,T], x_0^\theta \sim p_0^\theta, \\ x_t^\theta \sim p_t^\theta(\cdot | x_0^\theta)}} \left[ -2\alpha_{\text{SiD}} \cdot \alpha \|\delta_t(x_t^\theta)\|^2 + 2\alpha \langle \delta_t(x_t^\theta), f_t^*(x_t^\theta) \rangle - 2\beta \langle \delta_t(x_t^\theta), f_t^\theta(x_t^\theta | x_0^\theta) \rangle \right], \tag{34}$$

where $\delta_t = f_t - f_t^*$.

In the SiD framework for diffusion models, the data-free generator SiD loss (33) is additionally normalized, and the SiD loss with real data (34) should be normalized the same way. For more details on normalization, time sampling, weighting, etc., refer to the original articles (Zhou et al., 2024a;b).

**Experimental validation.** We modify the data-free SiD loss in the official SiD implementation with real data and conduct a short ablation study on unconditional CIFAR-10. The SiD codebase for diffusion models can be found in



https://github.com/mingyuanzhou/SiD.



We compare the data-free SiD loss (33) and our RealSiD loss (34) with the best coefficients $\alpha, \beta$ from Table 1 for both the theoretically justified $\alpha_{\text{SiD}} = 0.5$ and the best practical heuristic $\alpha_{\text{SiD}} = 1.2$. We do not change anything else and use the default hyperparameters and training pipeline as described in (Zhou et al., 2024b). The results are presented in Figure 5.

For the accurate $\alpha_{\text{SiD}} = 0.5$, the RealSiD results for diffusion models are similar to those for flow models. Configurations with $\beta/\alpha = 1.02$ boost convergence compared to the data-free baseline ($\alpha = \beta = 1$), whereas in the case of $\alpha = \beta \neq 1$, the convergence speed remains close to the baseline.

However, for the heuristic $\alpha_{\text{SiD}} = 1.2$, our best configurations with $\beta/\alpha \neq 1.0$ either degrade performance compared to the baseline or become unstable. Thus, the heuristical SiD may require a different approach to incorporate real data, or a more careful tuning of the coefficients $\alpha, \beta$ and other hyperparameters, due to differing architectures and training pipelines.

*We would like to highlight that all our analyses and recommendations were justified only for $\alpha_{SiD} = 0.5$. For other $\alpha_{SiD}$ values, this justification may not hold true.*

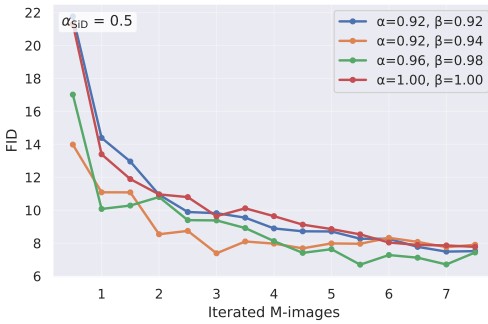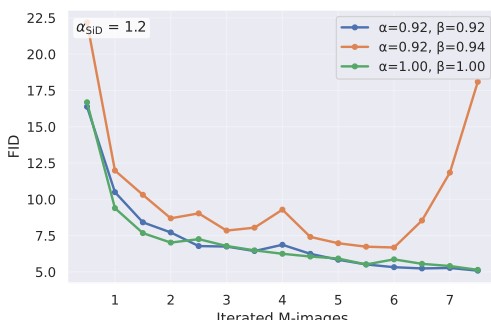

Figure 5: Evolution of FID during unconditional CIFAR-10 distillation for the data-free SiD loss ($\alpha = \beta = 1.0$) and our RealSiD loss for $\alpha_{\text{SiD}} = 0.5$ (left) and $\alpha_{\text{SiD}} = 1.2$ (right).

### A.4 NORMALIZED UID AND REALUID LOSSES FOR MINIMIZING $\ell_2$-DISTANCE

Using the linearization technique from (§3.1), we can estimate the non-squared $\ell_2$-distance between the teacher $f^* := \arg\min_f \mathcal{L}_{\text{UM}}(f, p_0^*)$ and student $f^\theta := \arg\min_f \mathcal{L}_{\text{UM}}(f, p_0^\theta)$ functions. In this case, the connection with the inverse optimization disappears.

For a fixed point $x_t^\theta \sim p_t^\theta$ and time $t \sim [0, T]$, we derive:

$$\|f_t^*(x_t^\theta) - f_t^\theta(x_t^\theta)\| = \max_{\delta_t(x_t^\theta)} \left\{ \langle \frac{\delta_t(x_t^\theta)}{\|\delta_t(x_t^\theta)\|}, f_t^*(x_t^\theta) - f_t^\theta(x_t^\theta) \rangle \right\}$$

$$= \max_{\delta_t(x_t^\theta)} \mathbb{E}_{x_0^\theta \sim p_0^\theta(\cdot|x_t^\theta)} \left[ \langle \frac{\delta_t(x_t^\theta)}{\|\delta_t(x_t^\theta)\|}, f_t^*(x_t^\theta) \rangle - \langle \frac{\delta_t(x_t^\theta)}{\|\delta_t(x_t^\theta)\|}, f_t^\theta(x_t^\theta|x_0^\theta) \rangle \right]. \quad (35)$$

With the parameterization $\delta_t = f_t^* - f_t$, the **Normalized UID loss** $\hat{\mathcal{L}}_{\text{UID}}(f, p_0^\theta)$ for solving $\min_\theta \mathbb{E}_{t \sim [0,T]} \mathbb{E}_{x_t^\theta \sim p_t^\theta} [\|f_t^*(x_t^\theta) - f_t^\theta(x_t^\theta)\|]$ is

$$\min_\theta \max_f \left\{ \hat{\mathcal{L}}_{\text{UID}}(f, p_0^\theta) := \mathbb{E}_{\substack{t \sim [0,T], x_0^\theta \sim p_0^\theta, \\ x_t^\theta \sim p_t^\theta(\cdot|x_0^\theta)}} \left[ \langle \frac{f_t^*(x_t^\theta) - f_t(x_t^\theta)}{\|f_t^*(x_t^\theta) - f_t(x_t^\theta)\|}, f_t^*(x_t^\theta) - f_t^\theta(x_t^\theta|x_0^\theta) \rangle \right] \right\}. \quad (36)$$

**Adding real data.** Following alternative definition of RealUID loss from Lemma 3, we can incorporate real data in Normalized UID loss (36) as well. We need to split two terms in the linearized form (35) into generated and real data parts with weights $\alpha, (1 - \alpha)$ and $\beta, (1 - \beta)$.

**Definition 4.** *We introduce **Normalized RealUID loss** $\hat{\mathcal{L}}_{R\text{-}UID}^{\alpha,\beta}(f, p_0^\theta)$ on generated data $p_0^\theta \in \mathcal{P}(\mathbb{R}^D)$ with coefficients $\alpha, \beta \in (0, 1]$:*

$$\hat{\mathcal{L}}_{R\text{-}UID}^{\alpha,\beta}(f, p_0^\theta) := \mathbb{E}_{\substack{t \sim [0,T], x_0^\theta \sim p_0^\theta, \\ x_t^\theta \sim p_t^\theta(\cdot|x_0^\theta)}} \left[ \langle \frac{f_t^*(x_t^\theta) - f_t(x_t^\theta)}{\|f_t^*(x_t^\theta) - f_t(x_t^\theta)\|}, \alpha \cdot f_t^*(x_t^\theta) - \beta \cdot f_t^\theta(x_t^\theta|x_0^\theta) \rangle \right]$$

$$+ \mathbb{E}_{\substack{t \sim [0,T], x_0^* \sim p_0^*, \\ x_t^* \sim p_t^*(\cdot|x_0^*)}} \left[ \langle \frac{f_t^*(x_t^*) - f_t(x_t^*)}{\|f_t^*(x_t^*) - f_t(x_t^*)\|}, (1 - \alpha) \cdot f_t^*(x_t^*) - (1 - \beta) \cdot f_t^*(x_t^*|x_0^*) \rangle \right].$$

Similar to the proof of RealUID distance Lemma 2, we can show that min-max optimization of Normalized RealUID loss minimizes the non-squared $\ell_2$-norm between the similar weighted student $f^\theta$ and teacher $f^*$ functions:

$$\max_f \hat{\mathcal{L}}_{\text{R-UID}}^{\alpha,\beta}(f, p_0^\theta) = \mathbb{E}_{t \sim [0,T]} \mathbb{E}_{x_t^* \sim p_t^*} \left[ \|((\beta - \alpha) + \alpha \frac{p_t^\theta(x_t^*)}{p_t^*(x_t^*)}) \cdot f_t^*(x_t^*) - \beta \frac{p_t^\theta(x_t^*)}{p_t^*(x_t^*)} \cdot f_t^\theta(x_t^*)\| \right].$$

This distance attains minimum when $p_0^\theta = p_0^*$, justifying the procedure.

## A.5 DMD AND INVERSE OPTIMIZATION

**Distribution Matching Distillation** (Luo et al., 2023; Wang et al., 2023; Yin et al., 2024b;a, **DMD**) approach distills Gaussian diffusion models with forward process $x_t = x_0 + \sigma_t \epsilon, \epsilon \sim \mathcal{N}(0, I)$.

This approach minimizes KL divergence $\mathbb{E}_{t \sim [0,T]}[D_{\text{KL}}(p_t^\theta || p_t^*)] = \mathbb{E}_{t \sim [0,T]} \mathbb{E}_{x_t^\theta \sim p_t^\theta} \left[ \log \left( \frac{p_t^\theta(x_t^\theta)}{p_t^*(x_t^\theta)} \right) \right]$ between the generated data $p_t^\theta$ and the real data $p_t^*$. The authors meticulously derive that the true gradient of $\mathbb{E}_{t \sim [0,T]}[D_{\text{KL}}(p_t^\theta || p_t^*)]$ w.r.t. $\theta$ can be computed via the score functions:

$$\mathbb{E}_{t \sim [0,T]} \left[ \frac{dD_{\text{KL}}(p_t^\theta || p_t^*)}{d\theta} \right] = \mathbb{E}_{\substack{t \sim [0,T], z \sim p^{\mathcal{Z}}, \\ x_0^\theta = G(z), x_t^\theta \sim p_t^\theta}} \left[ (\nabla_{x_t^\theta} \log p_t^{sg[\theta]}(x_t^\theta) - \nabla_{x_t^\theta} \log p_t^*(x_t^\theta)) \frac{dG_\theta(z)}{d\theta} \right].$$

Then, this true gradient is estimated with the teacher score function $s^* := \arg \min_s \mathcal{L}_{\text{DSM}}(s, p_0^*)$ and student score $s^\theta = \arg \min_s \mathcal{L}_{\text{DSM}}(s, p_0^\theta)$ obtained via minimizing DSM loss (1):

$$\mathbb{E}_{t \sim [0,T]} \left[ \frac{dD_{\text{KL}}(p_t^\theta || p_t^*)}{d\theta} \right] = \mathbb{E}_{\substack{t \sim [0,T], z \sim p^{\mathcal{Z}}, \\ x_0^\theta = G(z), x_t^\theta \sim p_t^\theta}} \left[ (s_t^\theta(x_t^\theta) - s_t^*(x_t^\theta)) \frac{dG_\theta}{d\theta} \right].$$

The final algorithm alternates updates for the fake model and the generator similar to SiD approach.

*Below we show that DMD fits our UID framework, but with another loss, different from the UM loss. In the case of diffusions, the UM loss is the $\mathcal{L}_{\text{DSM}}(s, p_0^\theta)$ loss, and with this loss, the resulting UID loss becomes exactly the SiD loss, not the DMD.*

**Inverse optimization view.** We decompose KL divergence as the difference of two KL losses:

$$\mathbb{E}_{t \sim [0,T]} \left[ D_{\text{KL}}(p_t^\theta || p_t^*) \right] = \mathbb{E}_{\substack{t \sim [0,T], \\ x_t^\theta \sim p_t^\theta}} \left[ \log \left( p_t^\theta(x_t^\theta) \right) \right] - \mathbb{E}_{\substack{t \sim [0,T], \\ x_t^\theta \sim p_t^\theta}} \left[ \log p_t^*(x_t^\theta) \right]$$

$$= \mathcal{L}_{\text{KL}}(p_t^\theta, p_t^\theta) - \mathcal{L}_{\text{KL}}(p_t^*, p_t^\theta),$$

where $\mathcal{L}_{\text{KL}}(q_t, p_t) := \mathbb{E}_{t \sim [0,T], x_t \sim p_t} [\log q_t(x_t)]$. We can differentiate through the generated samples $x_t^\theta \sim p_t^\theta$ from the second arguments of $\mathcal{L}_{\text{KL}}$ losses; however, the term $\mathcal{L}_{\text{KL}}(p_t^\theta, p_t^\theta) = \mathbb{E}_{t \sim [0,T], x_t^\theta \sim p_t^\theta} \left[ \log \left( p_t^\theta(x_t^\theta) \right) \right]$ remains intractable as it involves explicit generated data density $\log(p_t^\theta(\cdot))$ from the first argument. To make this term tractable, we apply the *linearization trick*. Due to non-negativity of KL divergence, we have the relation:

$$\mathbb{E}_{t \sim [0,T]} [D_{\text{KL}}(p_t || q_t)] = \mathcal{L}_{\text{KL}}(p_t, p_t) - \mathcal{L}_{\text{KL}}(q_t, p_t) \geq 0, \forall q_t \Rightarrow \mathcal{L}_{\text{KL}}(p_t, p_t) = \max_{q_t} \mathcal{L}_{\text{KL}}(q_t, p_t),$$

where maximum is attained when $q_t = p_t$. Thus, we substitute the intractable term $\mathcal{L}_{\text{KL}}(p_t^\theta, p_t^\theta)$ with the maximization problem parametrized by the *fake* distribution $q_t$:

$$\min_\theta \mathbb{E}_{t \sim [0,T]} \left[ D_{\text{KL}}(p_t^\theta || p_t^*) \right] = \min_\theta \{ \mathcal{L}_{\text{KL}}(p_t^\theta, p_t^\theta) - \mathcal{L}_{\text{KL}}(p_t^*, p_t^\theta) \}$$

$$= \min_\theta \{ \underbrace{\max_{q_t} \{ \mathcal{L}_{\text{KL}}(q_t, p_t^\theta) \} - \mathcal{L}_{\text{KL}}(p_t^*, p_t^\theta)}_{\geq 0} \} = \min_\theta \max_{q_t} \{ \mathcal{L}_{\text{KL}}(q_t, p_t^\theta) - \mathcal{L}_{\text{KL}}(p_t^*, p_t^\theta) \}. \quad (37)$$

Our min-max formulation of DMD (37) is the special case of the inverse optimization scheme (15) with the KL loss and teacher $p_t^* = \arg \max_{q_t} \mathcal{L}_{\text{KL}}(q_t, p_t^*)$. The generated data density $p_t^\theta$ appears only during sampling and we can efficiently backpropagate through it. Next, we easily calculate the gradient w.r.t. generator parameters, since we do not need to differentiate through independent $q_t$ which is equal to $p_t^{sg[\theta]}$ in the optimum:

$$\mathbb{E}_{t \sim [0,T]} \left[ \frac{dD_{\text{KL}}(p_t^\theta || p_t^*)}{d\theta} \right] = \frac{d}{d\theta} [\mathcal{L}_{\text{KL}}(p_t^{sg[\theta]}, p_t^\theta)] - \frac{d}{d\theta} [\mathcal{L}_{\text{KL}}(p_t^*, p_t^\theta)]$$

$$= \mathbb{E}_{\substack{t \sim [0,T], z \sim p^{\mathcal{Z}}, \\ x_0^\theta = G(z), x_t^\theta \sim p_t^\theta}} \left[ \frac{d}{d\theta} [\log p_t^{sg[\theta]}(x_t^\theta)] - \frac{d}{d\theta} [\log p_t^*(x_t^\theta)] \right]$$

$$= \mathbb{E}_{\substack{t \sim [0,T], z \sim p^{\mathcal{Z}}, \\ x_0^\theta = G(z), x_t^\theta \sim p_t^\theta}} \left[ (\underbrace{\nabla_{x_t^\theta} \log p_t^{sg[\theta]}(x_t^\theta)}_{=s_t^\theta(x_t^\theta)} - \underbrace{\nabla_{x_t^\theta} \log p_t^*(x_t^\theta)}_{=s_t^*(x_t^\theta)}) \frac{dG_\theta(z)}{d\theta} \right],$$

where teacher score $s^* := \arg \min_s \mathcal{L}_{\text{DSM}}(s, p_0^*)$ and fake score $s^\theta = \arg \min_s \mathcal{L}_{\text{DSM}}(s, p_0^\theta)$ are obtained via minimizing DSM loss (1) instead of maximizing the KL loss directly.

**Adding real data.** Following the logic from §3.4, we can modify the data-free KL loss $\mathcal{L}_{\text{KL}}(q_t, p_t) = \mathbb{E}_{t \sim [0,T], x_t \sim p_t} [\log q_t(x_t)]$ with the real data and put it back into the inverse optimization scheme (37). We propose the following modified loss with a single parameter $\alpha \in (0, 1]$:

$$\mathcal{L}^{\alpha}_{\text{R-KL}}(q_t, p_t^{\theta}) := \alpha \cdot \mathcal{L}_{\text{KL}}(q_t, p_t^{\theta}) + (1 - \alpha) \cdot \mathcal{L}_{\text{KL}}(q_t, p_t^*)$$

$$= \underbrace{\alpha \cdot \mathbb{E}_{t \sim [0,T], x_t^{\theta} \sim p_t^{\theta}} \left[ \log q_t(x_t^{\theta}) \right]}_{\text{generated data } p_t^{\theta} \text{ term}} + \underbrace{(1 - \alpha) \cdot \mathbb{E}_{t \sim [0,T], x_t^* \sim p_t^*} \left[ \log q_t(x_t^*) \right]}_{\text{real data } p_t^* \text{ term}}$$

$$= \mathcal{L}_{\text{KL}}(q_t, \alpha \cdot p_t^{\theta} + (1 - \alpha) \cdot p_t^*). \tag{38}$$

With this loss structure, the teacher distribution is preserved: $p_t^* = \arg\max_{q_t} \mathcal{L}^{\alpha}_{\text{R-KL}}(q_t, p_t^*) = \arg\max_{q_t} \mathcal{L}_{\text{KL}}(q_t, p_t^*)$. The KL loss $\mathcal{L}_{\text{KL}}(q_t, p_t^{\theta})$ is linear in $p_t^{\theta}$, thus, adding extra $\beta$ coefficient as we did in UM loss (16) is not working:

$$\alpha \cdot \mathcal{L}_{\text{KL}}(q_t, \frac{\beta}{\alpha} p_t^{\theta}) + (1 - \alpha) \cdot \mathcal{L}_{\text{KL}}(q_t, \frac{1 - \beta}{1 - \alpha} p_t^*) = \beta \cdot \mathcal{L}_{\text{KL}}(q_t, p_t^{\theta}) + (1 - \beta) \cdot \mathcal{L}_{\text{KL}}(q_t, p_t^*).$$

Putting the $\mathcal{L}_{\text{R-KL}}$ loss back into the inverse scheme (37), we explicitly obtain the KL divergence which minimizes the difference between the real data, and the mix of real and generated data:

$$\min_{\theta} \max_{q_t} \{ \mathcal{L}^{\alpha}_{\text{R-KL}}(q_t, p_t^{\theta}) - \mathcal{L}^{\alpha}_{\text{R-KL}}(p_t^*, p_t^{\theta}) \}$$

$$= \min_{\theta} \max_{q_t} \{ \mathcal{L}_{\text{KL}}(q_t, \alpha \cdot p_t^{\theta} + (1 - \alpha) \cdot p_t^*) - \mathcal{L}_{\text{KL}}(p_t^*, \alpha \cdot p_t^{\theta} + (1 - \alpha) \cdot p_t^*) \}$$

$$= \min_{\theta} \{ \mathcal{L}_{\text{KL}}(\alpha \cdot p_t^{\theta} + (1 - \alpha) \cdot p_t^*, \alpha \cdot p_t^{\theta} + (1 - \alpha) \cdot p_t^*) - \mathcal{L}_{\text{KL}}(p_t^*, \alpha \cdot p_t^{\theta} + (1 - \alpha) \cdot p_t^*) \}$$

$$= \min_{\theta} \mathbb{E}_{t \sim [0,T]} \left[ D_{\text{KL}}(\alpha \cdot p_t^{\theta} + (1 - \alpha) \cdot p_t^* || p_t^*) \right]. \tag{39}$$

Similarly, we calculate the gradient over generator parameters of our modified DMD (39):

$$\mathbb{E}_{t \sim [0,T]} \left[ \frac{d D_{\text{KL}}(\alpha \cdot p_t^{\theta} + (1 - \alpha) \cdot p_t^* || p_t^*)}{d\theta} \right]$$

$$= \frac{d}{d\theta} [\mathcal{L}_{\text{KL}}(\alpha \cdot p_t^{sg[\theta]} + (1 - \alpha) \cdot p_t^*, \alpha \cdot p_t^{\theta} + (1 - \alpha) \cdot p_t^*)] - \frac{d}{d\theta} [\mathcal{L}_{\text{KL}}(p_t^*, \alpha \cdot p_t^{\theta} + (1 - \alpha) \cdot p_t^*)]$$

$$= \alpha \cdot \mathbb{E}_{\substack{t \sim [0,T], z \sim p^{\mathcal{Z}}, \\ x_0^{\theta} = G(z), x_t^{\theta} \sim p_t^{\theta}}} \left[ \frac{d}{d\theta} [\log(\alpha \cdot p_t^{sg[\theta]}(x_t^{\theta}) + (1 - \alpha) \cdot p_t^*(x_t^{\theta}))] - \frac{d}{d\theta} [\log p_t^*(x_t^{\theta})] \right]$$

$$= \alpha \cdot \mathbb{E}_{\substack{t \sim [0,T], z \sim p^{\mathcal{Z}}, \\ x_0^{\theta} = G(z), x_t^{\theta} \sim p_t^{\theta}}} \left[ (\underbrace{\nabla_{x_t^{\theta}} \log(\alpha \cdot p_t^{sg[\theta]}(x_t^{\theta}) + (1 - \alpha) \cdot p_t^*(x_t^{\theta}))}_{=s_t^{\theta, \alpha}(x_t^{\theta})} - \underbrace{\nabla_{x_t^{\theta}} \log p_t^*(x_t^{\theta})}_{=s_t^*(x_t^{\theta})}) \frac{dG_{\theta}(z)}{d\theta} \right].$$

The score $s_t^{\theta, \alpha}$ of the mixed data $\alpha \cdot p_t^{\theta} + (1 - \alpha) \cdot p_t^*$ can be found using the DSM loss modified with the real data:

$$\mathcal{L}^{\alpha}_{\text{R-DSM}}(s, p_0^{\theta}) := \underbrace{\alpha \cdot \mathbb{E}_{t \sim [0,T]} \mathbb{E}_{x_0^{\theta} \sim p_0^{\theta}, x_t^{\theta} \sim p_t^{\theta}(\cdot|x_0)} \left[ \gamma_t \| s_t(x_t^{\theta}) - s^{\theta}(x_t^{\theta}|x_0^{\theta}) \|^2 \right]}_{\text{generated data } p_0^{\theta} \text{ term}}$$

$$+ \underbrace{(1 - \alpha) \cdot \mathbb{E}_{t \sim [0,T]} \mathbb{E}_{x_0^* \sim p_0^*, x_t^* \sim p_t^*(\cdot|x_0^*)} \left[ \gamma_t \| s_t(x_t^*) - s_t^*(x_t^*|x_0^*) \|^2 \right]}_{\text{real data } p_0^* \text{ term}}.$$

We sum up all conclusions in the lemma below.

**Lemma 5 (DMD with real data).** *Consider real data $p_0^* \in \mathcal{P}(\mathbb{R}^D)$ and generated data $p_0^{\theta} \in \mathcal{P}(\mathbb{R}^D)$. Then, KL divergence between mixed and real data for $\alpha \in (0, 1]$ has the following gradient with modified student score $s_t^{\theta, \alpha} := \arg\min_s \mathcal{L}^{\alpha}_{DSM}(s, p_0^{\theta})$ and teacher score $s_t^* := \arg\min_s \mathcal{L}_{DSM}(s, p_0^*)$:*

$$\mathbb{E}_{t \sim [0,T]} \left[ \frac{d D_{KL}(\alpha \cdot p_t^{\theta} + (1 - \alpha) \cdot p_t^* || p_t^*)}{d\theta} \right] = \mathbb{E}_{\substack{t \sim [0,T], z \sim p^{\mathcal{Z}}, \\ x_0^{\theta} = G_{\theta}(z), x_t^{\theta} \sim p_t^{\theta}}} \left[ \alpha (s_t^{\theta, \alpha}(x_t^{\theta}) - s_t^*(x_t^{\theta})) \frac{dG_{\theta}}{d\theta} \right].$$

Even though this approach is theoretically justified, it requires coefficients $\alpha = \beta$ which work poorly for our RealUID; see Table 1.

**Extension to Bregman divergences.** KL divergence belongs to a large family of distance functions called Bregman divergences. A divergence $D_\Psi(p_t^\theta, p_t^*)$ is determined by a convex function $\Psi$:

$$D_\Psi(p_t^\theta, p_t^*) := \Psi(p_t^\theta) - \Psi(p_t^*) - \langle \nabla\Psi(p_t^*), p_t^\theta - p_t^* \rangle \geq 0. \tag{40}$$

For KL divergence, we have the function $\Psi(p) = \int \log(p(x))p(x)dx$. In (40), the only non-linear and intractable term w.r.t. $p_t^\theta$ is the function $\Psi(p_t^\theta)$, but we can similarly make it tractable via the linearization trick:

$$\Psi(p_t^\theta) = \max_{q_t}\{\Psi(q_t) + \langle \nabla\Psi(q_t), p_t^\theta - q_t \rangle\},$$

and build a general inverse distillation scheme:

$$\min_\theta \mathbb{E}_t[D_\Psi(p_t^\theta, p_t^*)] = \min_\theta \max_{q_t}\{\underbrace{\mathbb{E}_t[\Psi(q_t) + \langle \nabla\Psi(q_t), p_t^\theta - q_t \rangle]}_{=:\mathcal{L}_\Psi(q_t, p_t^\theta)} - \underbrace{\mathbb{E}_t[\Psi(p_t^*) + \langle \nabla\Psi(p_t^*), p_t^\theta - p_t^* \rangle]}_{=:\mathcal{L}_\Psi(p_t^*, p_t^\theta)}\}.$$

# B REALUID ALGORITHM FOR FLOW MATCHING MODELS

We provide a practical implementation of our RealUID approach for flow matching models in Algorithm 1. In the loss functions, we retain only the terms dependent on the target parameters. For the fake model, we reformulate the maximization objective as a minimization. We use alternating optimization, updating the fake model $K$ times per one student update for stability.

---

**Algorithm 1** Real data modified Unified Inversion Distillation (RealUID) for Flow Matching

---

**Input:** teacher drift $u^*$, student generator $G_\theta$, fake drift $u_\psi$, real data $p_0^*$, coefficients $\alpha, \beta \in (0, 1]$, generator update steps $K$, number of iterations $N$, batch size $B$, fake drift minimizer $Opt_{st}$, generator minimizer $Opt_{gen}$, latent distribution $p^\mathcal{Z}$, noise distribution $p_1$.

1: **for** $n = 0, \ldots, N - 1$ **do**
2:     Sample generated batch $\{x_{0,i}^\theta = G_\theta(z_i)\}_{i=1}^B$, $z_i \sim p^\mathcal{Z}$ and noise batch $\{x_{1,i}\}_{i=1}^B \sim p_1$;
3:     Sample time batch $\{t_i\}_{i=1}^B \sim Uniform[0, 1]$ and calculate $x_{t_i,i}^\theta = (1 - t_i)x_{0,i}^\theta + t_i x_{1,i}$;
4:     **if** student step $(n \% (K + 1) \neq 0)$ **then**
5:         Sample real data batch $\{x_{0,i}^*\}_{i=1}^B \sim p_0^*$ and calculate $x_{t_i,i}^* = (1 - t_i)x_{0,i}^* + t_i x_{1,i}$;
6:         Update fake drift parameters $\psi$ via minimizer $Opt_{st}$ step with gradients of

$$\frac{1}{B} \sum_{i=1}^B \left[ \alpha \| u_\psi(t_i, x_{t_i,i}^{sg[\theta]}) - \frac{\beta}{\alpha}(x_{1,i} - x_{0,i}^{sg[\theta]}) \|^2 + (1 - \alpha)\| u_\psi(t_i, x_{t_i,i}^*) - \frac{1-\beta}{1-\alpha}(x_{1,i} - x_{0,i}^*) \|^2 \right];$$

7:     **else**
8:         Update generator parameters $\theta$ via minimizer $Opt_{gen}$ step with gradients of

$$\frac{1}{B} \sum_{i=1}^B \left[ \alpha \| u^*(t_i, x_{t_i,i}^\theta) - \frac{\beta}{\alpha}(x_{1,i} - x_{0,i}^\theta) \|^2 - \alpha \| u_{sg[\psi]}(t_i, x_{t_i,i}^\theta) - \frac{\beta}{\alpha}(x_{1,i} - x_{0,i}^\theta) \|^2 \right];$$

9:     **end if**
10: **end for**

---

## C UNIFIED INVERSE DISTILLATION WITH REAL DATA FOR BRIDGE MATCHING AND STOCHASTIC INTERPOLANTS

### C.1 BRIDGE MATCHING

Bridge Matching (Liu et al., 2022; Peluchetti, 2023) is an extension of diffusion models specifically design to solve data-to-data, e.g., image-to-image problems. Typically, the distribution $p_T$ is the distribution of "corrupted data" and $p_0$ is the distribution of clean data, furthermore, there is some coupling of clean and corrupted data $\pi(x_0, x_T)$ with marginals $p_0(x_0)$ and $p_T(x_T)$. To construct the diffusion which recovers clean data given a corrupted data, one first needs to build prior process (which often is the same forward process used in diffusions):

$$dx_t = f_t(x_t)dt + g_t d\mathrm{w}_t,$$

where $f_t(\cdot)$ is a drift function, $g_t$ is a time-dependent scalar noise scheduler and $\mathrm{w}_t$ is a standard Wiener process. This prior process defines conditional density $p_t(x_t|x_0)$ and the posterior density $p_t(x_t|x_0, x_T)$ called "diffusion bridge". To recover $p_0$ from $p_T$, one can use reverse-time SDE with a reverse-time Wiener process $\bar{\mathrm{w}}_t$:

$$dx_t = \left(f_t(x_t) - g_t^2 \cdot u_t(x_t)\right) dt + g_t d\bar{\mathrm{w}}_t,$$

where the drift $u_t(x_t)$ is learned via solving of the bridge matching problem:

$$\mathcal{L}_{\mathrm{BM}}(v, \pi) = \mathbb{E}_{t\sim[0,T],(x_0,x_T)\sim\pi, x_t\sim p_t(\cdot|x_0,x_T)} \left[\|v_t(x_t) - \nabla_{x_t} \log p_t(x_t|x_0)\|^2\right]. \quad (41)$$

However, this reverse-time diffusion in general does not guarantee that the produced samples come from the same coupling $\pi(x_0, x_T)$ used for training. It happens only if $\pi(x_0, x_T)$ solves entropic optimal transport between $p_0$ and $p_T$. To guarantee the preservance of the coupling $\pi(x_0, x_T)$, there exists another version of Bridge Matching called either Augmented Bridge Matching or Conditional Bridge Matching (De Bortoli et al., 2023), which differs only by addition of a condition on $x_T$ to the trainable drift $v_t(x_t, x_T)$:

$$\mathcal{L}_{\mathrm{ABM}}(v, \pi) = \mathbb{E}_{t\sim[0,T],(x_0,x_T)\sim\pi, x_t\sim p(\cdot|x_0,x_T)} \left[\|v_t(x_t, x_T) - \nabla_{x_t} \log p_t(x_t|x_0)\|_2^2\right].$$

The learned conditional drift $u(x_t, x_T)$ is then used for sampling via the reverse-time SDE starting from a given $x_T \sim p_T$:

$$dx_t = \left(f_t(x_t) - g_t^2 \cdot u_t(x_t, x_T)\right) dt + g_t d\bar{\mathrm{w}}_t.$$

### C.2 STOCHASTIC INTERPOLANTS

The Stochastic Interpolants framework generalizes Flow Matching and diffusion models, constructing a diffusion or flow between two given distributions $p_0$ and $p_T$. To do so, one needs to consider the interpolation between any pair of points $(x_0, x_T)$ which are sampled from the coupling $\pi(x_0, x_T)$ with marginals $p_0$ and $p_T$. The interpolation itself is given by formula

$$x_t = I(t, x_0, x_T) + \gamma_t \epsilon, \quad \epsilon \sim \mathcal{N}(0, \mathbf{I}), \quad t \in [0, T],$$

where $I(0, x_0, x_T) = x_0$, $I(T, x_0, x_T) = x_T$, $\gamma_0 = \gamma_T = 0$ and $\gamma_t > 0$ for all $t \in (0, T)$. This interpolant defines a conditional Gaussian path $p_t(x_t|x_0, x_T)$. Note that in the original paper (Albergo et al., 2023), the authors consider the time interval $[0, 1]$, but those two intervals are interchangeable by using a change of variable $t' = \frac{T}{t}$. Thus, the ODE interpolation between $p_0$ and $p_T$ is given by:

$$dx_t = u_t(x_t)dt, \quad x_0 \sim p_0,$$

where $u_t(x, x_T) := \mathbb{E}[\dot{x}_t|x_t = x] = \mathbb{E}[\partial_t I(t, x_0, x_T) + \dot{\gamma}_t \epsilon | x_t = x]$ is the unique minimizer of the quadratic objective:

$$\mathcal{L}_{\mathrm{SI}}(v, \pi) = \mathbb{E}_{\substack{t\sim[0,T],(x_0,x_T)\sim\pi, \\ (x_t,\epsilon)\sim p(\cdot|x_0,x_T)}} \left[\|v_t(x_t, x_T) - (\partial_t I(t, x_0, x_T) + \dot{\gamma}_t \epsilon)\|^2\right]. \quad (42)$$

The authors also provide a way of matching the score and the SDE drift of the reverse process by solving similar MSE matching problems.

### C.3 OBJECTIVE FOR GENERAL DATA COUPLING

The essential difference of Bridge Matching and Stochastic Interpolants from diffusion models and Flow Matching with a Gaussian path is that they additionally introduce coupling $\pi(x_0, x_T)$ used to sample $x_t$ and can work with conditional drifts.

This difference can be easily incorporated to our RealUID distillation framework just by parametrizing the generator $G_\theta$ to output not the samples from the initial distribution $p_0^\theta$, but from the coupling $\pi^\theta$. One can do it by setting $\pi^\theta(x_0, x_T) = p_T(x_T)\pi_0^\theta(x_0|x_T)$, where conditional data distribution $\pi_0^\theta(x_0|x_T)$ is parametrized by the *student generator* $G_\theta : \mathcal{Z} \times \mathbb{R}^D \to \mathbb{R}^D$ conditioned on a sample $x_T \sim p_T$. This approach is specifically used in Inverse Bridge Matching Distillation (IBMD) (Gushchin et al., 2024). Hence, our Universal Inverse Distillation objective can be written just by substituting student distribution $p_0^\theta$ by student coupling $\pi^\theta$, substituting real data $p_0^*$ by real data coupling $\pi^*$ and adding extra conditions.

**Definition 5.** *We define **Universal Matching loss with real data for general coupling** on generated data coupling $\pi^\theta \in \mathcal{P}(\mathbb{R}^D \times \mathbb{R}^D)$ with $\alpha, \beta \in (0,1]$:*

$$\mathcal{L}_{\text{R-UM-coup}}^{\alpha,\beta}(f, \pi^\theta) = \alpha \cdot \underbrace{\mathbb{E}_{t\sim[0,T]}\mathbb{E}_{\substack{x_T\sim p_T, x_0^\theta\sim\pi_0^\theta(\cdot|x_T), \\ x_t^\theta\sim p_t^\theta(\cdot|x_0^\theta, x_T)}} \left[\|f_t(x_t^\theta, x_T) - \frac{\beta}{\alpha} f^\theta(x_t^\theta|x_0^\theta, x_T)\|^2\right]}_{\text{generated data } \pi^\theta \text{ term}}$$

$$+ (1-\alpha) \cdot \underbrace{\mathbb{E}_{t\sim[0,T]}\mathbb{E}_{\substack{x_T\sim p_T, x_0^*\sim\pi_0^*(\cdot|x_T), \\ x_t^*\sim p_t^*(\cdot|x_0, x_T)}} \left[\|f_t(x_t^*, x_T) - \frac{1-\beta}{1-\alpha} f_t^*(x_t^*|x_0^*, x_T)\|^2\right]}_{\text{real data } \pi^* \text{ term}}.$$

*And the corresponding **Universal Inverse Distillation loss with real data for general coupling** is:*

$$\min_\theta \max_f \{\mathcal{L}_{\text{R-UID-coup}}^{\alpha,\beta}(f, \pi^\theta) := \mathcal{L}_{\text{R-UM-coup}}^{\alpha,\beta}(f^*, \pi^\theta) - \mathcal{L}_{\text{R-UM-coup}}^{\alpha,\beta}(f, \pi^\theta)\}.$$

In case of coupling match $\pi^\theta = \pi^*$, the RealUID loss for couplings attains its minimum, i.e.,

$$\min_\theta \max_f \mathcal{L}_{\text{R-UID-coup}}^{\alpha,\beta}(f, \pi^\theta) = \min_\theta \{\underbrace{\mathcal{L}_{\text{R-UM-coup}}^{\alpha,\beta}(f^*, \pi^\theta) - \min_f \{\mathcal{L}_{\text{R-UM-coup}}^{\alpha,\beta}(f, \pi^\theta)\}}_{\geq 0}\}$$

$$= \mathcal{L}_{\text{R-UM-coup}}^{\alpha,\beta}(f^*, \pi^*) - \underbrace{\min_f \{\mathcal{L}_{\text{R-UM-coup}}^{\alpha,\beta}(f, \pi^*)\}}_{=\mathcal{L}_{\text{R-UM-coup}}^{\alpha,\beta}(f^*, \pi^*)} = 0.$$

It is worth noting that the inverse distillation from the IBMD framework can be extended to discrete-time Bridge Matching models, as is done in **Residual Shifting Distillation** (Selikhanovych et al., 2025, **RSD**). Likewise, our RealUID objective can be extended to discrete-time models as well.

# D    EXPERIMENTAL DETAILS AND ADDITIONAL RESULTS

## D.1    CIFAR-10 DISTILLATION FROM SCRATCH

**Codebase, dataset and teachers.**    Building on the reference codebase and network architectures of (Tong et al., 2024), we implement the training algorithm described in our Algorithm 1. We evaluate the resulting approach on CIFAR-10 (32×32), under both conditional and unconditional settings, benchmarking against established baselines. The codebase implementation is publicly available in

https://github.com/atong01/conditional-flow-matching.

Note that in this codebase, the time flow is reversed, i.e., the time $t = 0$ corresponds to the pure noise, while the time $t = 1$ is the real data. As an unconditional teacher, we use already trained Conditional Flow Matching checkpoints from the above repository. For conditional setup, we slightly modify the original code and train our own teacher. Our trained checkpoints, along with the code, are located in

https://github.com/David-cripto/RealUID.

**Training hyperparameters.**    We train both our models with Adam (Kingma & Ba, 2014), using the same momentums $(\beta_1, \beta_2) = (0, 0.999)$, learning rate $3 \times 10^{-5}$ and a 500-step linear warm-up. Similar to SiD framework (Zhou et al., 2024a), we do not recommend setting momentum $\beta_1 \neq 0$ as it is crucial for a successful convergence in our min-max optimization.

To regulate adaptation between the generator and the fake model, the generator is updated once for every $K = 5$ updates of the fake model, following DMD2 (Yin et al., 2024a). While the SiD framework leverages an EDM architecture (Karras et al., 2022) and updates the generator after a single update of the fake model ($K = 1$), our RealUID approach becomes unstable for values $K < 3$ due to the different (Tong et al., 2024) architecture.

We do not use dropout in generator and fake models. We set a batch size of 256 and maintain an EMA of the generator parameters with decay 0.999 (Hunter, 1986). Additionally, at each optimization step we apply $\ell_2$ gradient-norm clipping with threshold 1.0 to both the generator and the fake model.

**Training time.**    All distillation experiments were trained for 500,000 gradient updates, corresponding to approximately 5 days. The experiments were executed on a single Ascend910B NPU with 65 GB of VRAM memory.

**Generator parameterization and models initialization.**    We parameterize generator $G_\theta(\cdot)$ using a time-dependent U-Net $g_\theta(0, \cdot)$ with a fixed time input $t = 0$ and a one-step integration scheme:

$$G_\theta(z) = z + g_\theta(0, z).$$

We initialize the model $g_\theta$ with a teacher model, and the fake model with random weights. Empirically, we observe that this initialization strategy lead to improved performance on the considered datasets.

**GAN details.**    We integrate a GAN loss into our framework in line with SiD$^2$A and DMD2 (Zhou et al., 2024a; Yin et al., 2024a). In the original setup of (Zhou et al., 2024a), the adversarial loss employs a coefficient ratio of $\lambda_{\text{adv}}^D / \lambda_{\text{adv}}^{G_\theta} = 10^2$ (see Table 6 in Zhou et al. (2024a)), a choice that poses practical difficulties due to the extreme imbalance between the generator and discriminator losses. To mitigate this issue, we adopt the formulation of (Yin et al., 2024a), where the ratio is $\approx 3$, and evaluate different coefficient scales (see the results in Table 1). Additionally, we can select the range of times within which adversarial loss is applied between noised generated and real data samples. We found that the best choice is not to take only clear real data or the whole interval [0,1], but rather to take the range of not severely corrupted data, namely times from 0.8 to 1.

**Evaluation protocol.**    We evaluate image quality using the Fréchet Inception Distance (FID; Heusel et al., 2017), computed from 50,000 generated samples following (Karras et al., 2022; 2020; 2019). In line with SiD (Zhou et al., 2024b), we periodically compute FID during distillation and select the checkpoint achieving the minimum value. To ensure statistical reliability, we repeat the evaluation over 3 independent runs, rather than 10 as in SiD, because the empirical variance of FID in our experiments was below 0.01.

**Efficiency comparison.** In terms of efficiency, RealUID leverages a lightweight architecture based on (Tong et al., 2024). Therefore, as summarized in Table 5, it achieves nearly $2\times$ faster inference, lower memory usage, and reduced model size compared to recent distillation approaches (Zhou et al., 2024b;a; Huang et al., 2024).

| Methods | Inference Time (ms) | # Total Param (M) | Max GPU Mem Alloc (MB) | Max GPU Mem Reserved (MB) |
|---|---|---|---|---|
| RealUID (**Ours**) | **18.636** | **36.784** | **165** | **172** |
| FGM (Huang et al., 2024) | 30.745 | 55.734 | 242 | 276 |
| SiD (Zhou et al., 2024b;a) | | | | |

Table 5: Inference complexity on an Ascend 910B3 (65 GB) NPU. All methods require only 1 NFE. For each method, we report **(i)** the mean inference time per image (bs=1, fp32), averaged over 10,000 iterations; **(ii)** the total number of parameters (Millions); and **(iii)** peak NPU memory usage (maximum allocated and reserved, in MB). Best values are **bolded**.

## D.2 CIFAR-10 DISTILLATION FINE-TUNING

This section presents an ablation study of the fine-tuning stage over the loss-balancing coefficients for GANs and our RealUID on CIFAR-10. In this stage, the generator is initialized from the best-performing checkpoint obtained during training from scratch of the corresponding framework, while the fake model is initialized from the teacher model. In the unconditional setup, the best configuration are RealUID with ($\alpha = 0.92, \beta = 0.94$) and FID 2.22, and GAN with ($\lambda_{\text{adv}}^{G_\theta} = 0.3, \lambda_{\text{adv}}^D = 1$) and FID 2.29. In the unconditional setup, it is RealUID with ($\alpha = 0.98, \beta = 0.96$) and FID 2.02, and GAN with ($\lambda_{\text{adv}}^{G_\theta} = 0.3, \lambda_{\text{adv}}^D = 1$) and FID 2.12. Fine-tuning then proceeds with new values $\alpha_{\text{FT}}$ and $\beta_{\text{FT}}$ for our RealUID and $\lambda_{\text{FT}}^{G_\theta}$ and $\lambda_{\text{FT}}^D$ for GANs. The results are summarized in Table 6.

Table 6: Ablation of the fine-tuning parameters ($\alpha_{\text{FT}}, \frac{\beta_{\text{FT}}}{\alpha_{\text{FT}}}$) for our RealUID and fine-tuning scales ($\lambda_{\text{FT}}^{G_\theta}, \lambda_{\text{FT}}^D$) for GANs for unconditional (left) and conditional (right) generation. All values report FID $\downarrow$, where lower is better. The mark "–" indicates that configuration is infeasible, and the mark "–" shows that the method did not converge. Best results for each method are **bolded**.

| $\alpha_{\text{FT}}\backslash\frac{\beta_{\text{FT}}}{\alpha_{\text{FT}}}$ | 0.92 | 0.94 | 0.96 | 0.98 | 1.0 | 1.02 | 1.04 | 1.06 | 1.08 |
|---|---|---|---|---|---|---|---|---|---|
| 0.92 | 1.99 | **1.98** | 2.02 | – | – | – | 2.04 | 2.04 | 2.02 |
| 0.94 | 2.02 | 2.02 | 2.04 | – | – | – | 2.07 | 2.06 | - |
| 0.96 | 2.06 | 2.04 | 2.09 | – | – | – | 2.08 | - | - |
| 0.98 | 2.07 | 2.05 | 2.07 | – | – | – | - | - | - |

| $\alpha_{\text{FT}}\backslash\frac{\beta_{\text{FT}}}{\alpha_{\text{FT}}}$ | 0.92 | 0.94 | 0.96 | 0.98 | 1.0 | 1.02 | 1.04 | 1.06 | 1.08 |
|---|---|---|---|---|---|---|---|---|---|
| 0.92 | 1.92 | 1.91 | 1.99 | – | – | – | 1.96 | 1.94 | 1.92 |
| 0.94 | 1.92 | 1.90 | 1.88 | – | – | – | 1.96 | 1.91 | - |
| 0.96 | 1.93 | 1.94 | **1.87** | – | – | – | 1.96 | - | - |
| 0.98 | 1.91 | 1.95 | 1.95 | – | – | – | - | - | - |

| $\lambda_{\text{FT}}^{G_\theta}$ | 0.1 | 0.3 | 1.0 | 5.0 | 25.0 | 100.0 |
|---|---|---|---|---|---|---|
| $\lambda_{\text{FT}}^D$ | 0.3 | 1.0 | 3.0 | 15.0 | 75.0 | 300.0 |
| FID $\downarrow$ | – | – | – | 2.25 | **2.10** | 2.12 |

| $\lambda_{\text{FT}}^{G_\theta}$ | 0.1 | 0.3 | 1.0 | 5.0 | 25.0 | 100.0 |
|---|---|---|---|---|---|---|
| $\lambda_{\text{FT}}^D$ | 0.3 | 1.0 | 3.0 | 15.0 | 75.0 | 300.0 |
| FID $\downarrow$ | – | – | – | 1.94 | **1.88** | 2.04 |

We observe that fine-tuning is highly sensitive to the choice of factor $\frac{\beta_{\text{FT}}}{\alpha_{\text{FT}}}$ which still brings the main impact. The best factors $\frac{\beta_{\text{FT}}}{\alpha_{\text{FT}}} = 0.94$ or $\frac{\beta_{\text{FT}}}{\alpha_{\text{FT}}} = 1.06$ are much farther from $1.0$ compared to training from scratch (Table 1), i.e., fine-tuning relies more on information from real data rather than on guidance from a teacher. Meanwhile, configurations closer to 1.0 are unstable, underscoring the crucial role of real data. In the case of GANs, small adversarial losses similarly fail to converge, and only high scales which particularly emphasize real data achieve improvement.

**Training details.** We run fine-tuning with a smaller learning rate $1 \times 10^{-5}$ and without warm-up. All other details remain the same as described in Appendix D.1 for training from scratch.

**Training time.** All fine-tuning experiments were conducted for 100,000 gradient updates, which took a little more than 1 day, starting from the best distillation checkpoints. The experiments were executed on a single Ascend910B NPU with 65 GB of VRAM memory.

## D.3 CELEBA DISTILLATION

In this section, we present the results of the same ablation study from (§4.2) on the CelebA dataset with higher $64 \times 64$ resolution (Liu et al., 2015). The results are summarized in Table 7. Similar to

Table 1 for CIFAR10, the same pairs of coefficients with $\beta/\alpha = 1.02$ or $\beta/\alpha = 0.98$ yield a significant improvement in quality over the baseline ($\alpha = 1.0, \beta = 1.0$), reaching a level comparable to GANs.

| $\alpha \backslash \frac{\beta}{\alpha}$ | 0.96 | 0.98 | 1.00 | 1.02 | 1.04 |
|---|---|---|---|---|---|
| 0.88 | **1.03** | 1.08 | 1.36 | 1.14 | 1.45 |
| 0.90 | 1.06 | **1.03** | 1.38 | 1.06 | 1.48 |
| 0.92 | 1.12 | 1.04 | 1.28 | 1.10 | 1.69 |
| 0.94 | 1.13 | **1.03** | 1.18 | 1.10 | 1.64 |
| 0.96 | 1.24 | 1.11 | 1.25 | 1.07 | 1.69 |
| 0.98 | 1.65 | 1.26 | 1.22 | 1.29 | - |
| 1.0 | - | - | 1.20 | - | - |

| $\lambda_{\text{adv}}^{G_\theta}$ | $\lambda_{\text{adv}}^{D}$ | FID ($\downarrow$) |
|---|---|---|
| 0.1 | 0.3 | 1.14 |
| 0.3 | 1 | 1.18 |
| 1 | 3 | 1.10 |
| 5 | 15 | **1.04** |
| 25 | 75 | 3.31 |

Table 7: Ablation studies of our $(\alpha, \frac{\beta}{\alpha})$ parameters in the left table and adversarial weighting parameters $(\lambda_{\text{adv}}^{G_\theta}, \lambda_{\text{adv}}^{D})$ in the right table for CelebA, 800,000 iterations. The baseline RealUID ($\alpha = 1.0, \beta = 1.0$) does not use real data. Configurations that substantially outperform the baseline are highlighted. All values report FID$\downarrow$, where lower is better. The best configuration is **bolded**. The mark "–" denotes infeasible configurations.

**Training hyperparameters and details.** We take the same architecture (Tong et al., 2024) as for CIFAR-10, but adapt it to a larger resolution. We train both models with Adam (Kingma & Ba, 2014) for 800,000 iterations, using the same momentums $(\beta_1, \beta_2) = (0, 0.999)$, learning rate $5 \times 10^{-6}$ and a 500-step linear warm-up. Similar to SiD framework (Zhou et al., 2024a), we do not recommend setting momentum $\beta_1 \neq 0$ as it is crucial for a successful convergence in our min-max optimization.

To regulate adaptation between the generator and the fake model, the generator is updated once for every $K = 5$ updates of the fake model, following DMD2 (Yin et al., 2024a). While the SiD framework leverages an EDM architecture (Karras et al., 2022) and updates the generator after a single update of the fake model ($K = 1$), our RealUID approach becomes unstable for values $K < 3$ due to the different (Tong et al., 2024) architecture.

We do not use dropout in generator and fake models. We set a batch size of 64 and maintain an EMA of the generator parameters with decay 0.999 (Hunter, 1986). Additionally, at each optimization step we apply $\ell_2$ gradient-norm clipping with threshold 1.0 to both the generator and the fake model.

All other details remain the same as described in Appendix D.1 for CIFAR-10.

**Teacher training.** For CelebA, we train our own teacher model based on the official implementation of the conditional flow matching procedure from (Tong et al., 2024). We use the same pipeline, architectures, and hyperparameters, but with larger networks and a different dataset. The adapted code for teacher training and final checkpoints for distillation can be found in our repository:

<p align="center">https://github.com/David-cripto/RealUID.</p>

**Fine-tuning.** For fine-tuning, we hold the data-free UID baseline (our RealUID with $\alpha = 1.0, \beta = 1.0$) and all highlighted GAN and RealUID setups from Table 7 for twice as long, i.e., for 1,600,000 iterations. The best-found configurations and results are reported in Table 8 and Figure 6. According to it, our RealUID still outperforms data-free UID baseline, reaching the same performance as GANs.

**Training time.** All experiments were executed on a single Ascend910B NPU with 65 GB of VRAM memory. Regular 800,000 gradient updates took approximately 5 days, while longer fine-tuning with 1,600,000 iterations took 10 days.

Table 8: This table presents the results of ablation study of our RealUID framework, evaluated using the FID metric on CelebA dataset, 1,600,000 iterations. The Teacher Flow model with 100 NFE is reported as a reference. The performance of the UID (FGM) baseline without real-data incorporation is indicated in *italic*. For emphasis, we underline the two counterparts that incorporate real data: the GAN-based and our RealUID methods. The best-performing configuration is highlighted in **bold**. Qualitative results are presented in Appendix D.5.2.

| Model | FID ($\downarrow$) |
|---|---|
| Teacher Flow (NFE=100) | 2.46 |
| UID (FGM) | *0.96* |
| UID + GAN ($\lambda_{\text{adv}}^{G_\theta} = 1.0, \lambda_{\text{adv}}^{D} = 3.0$) | **0.87** |
| RealUID ($\alpha = 0.88, \beta = 0.90$) (**Ours**) | 0.89 |

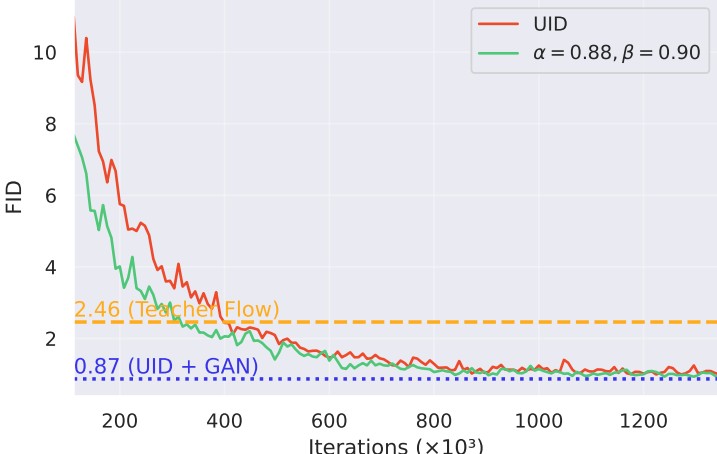

Figure 6: Evolution of FID during CelebA distillation for the data-free UID baseline and the best-performing RealUID configuration. The performances of Teacher Flow and UID+GAN are indicated by horizontal lines in their respective colors.

## D.4 FURTHER HYPERPARAMETERS GRIDSEARCH

The primary goal across all experiments in this paper was to study RealUID framework, focusing on the effects of the coefficients $\alpha$ and $\beta$, and provide a fair comparison with GANs. For this reason, we kept all other hyperparameters fixed at their standard values. Now that we have identified the optimal settings for RealUID, we can explore other hyperparameters. Below, we provide a list of useful findings, while the latest hyperparameters sets and training pipelines are described in our repository

https://github.com/David-cripto/RealUID.

**EMA decays.** One can track not only a single EMA decay but a range of values, e.g., [0.999, 0.9996, 0.9999], during a single training run. In long-distance training, larger EMA decays can lead to more stable convergence dynamics and better metrics, whether training from scratch or fine-tuning.

## D.5 EXAMPLE OF SAMPLES FOR VARIOUS METHODS

This section presents representative sample outputs from various studies conducted within the RealUID framework.

### D.5.1 CIFAR-10 GENERATED IMAGES

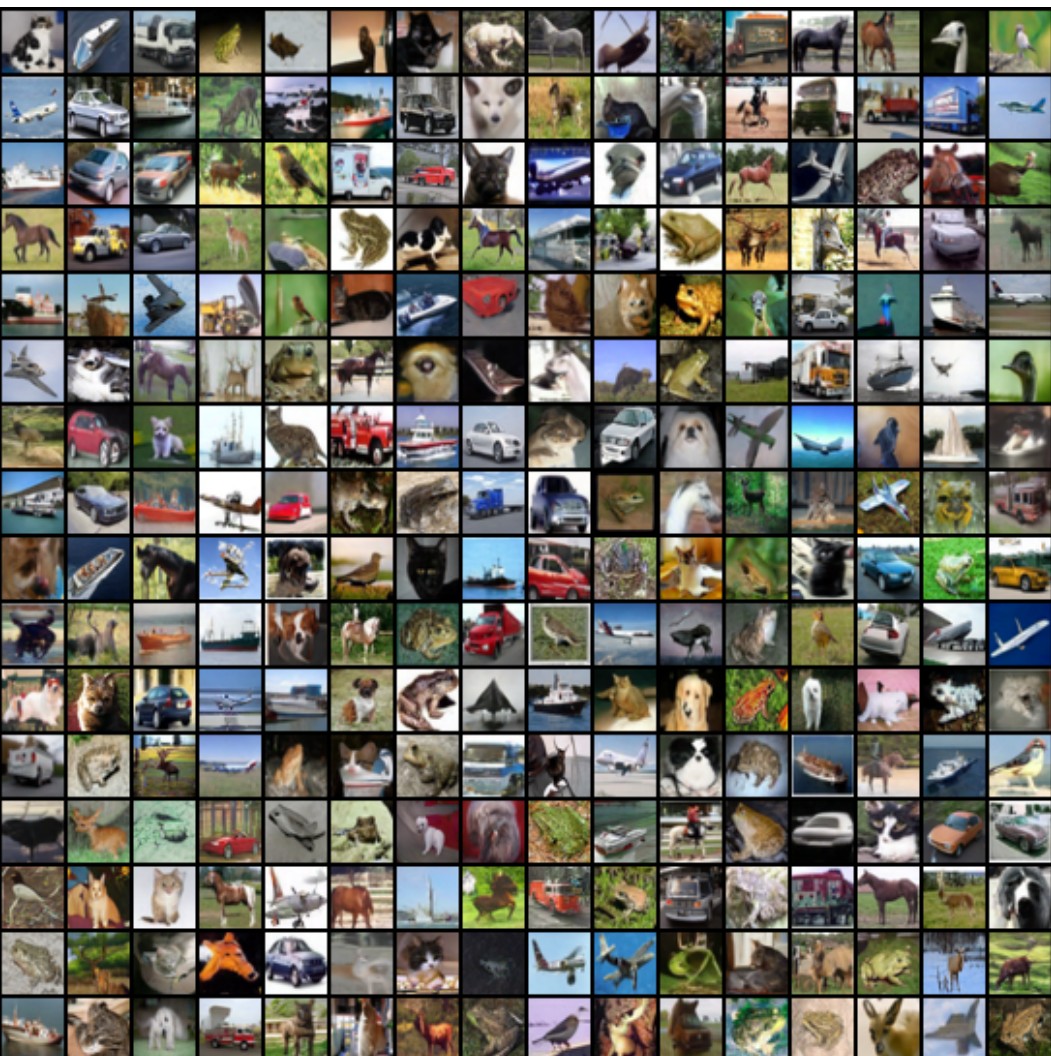

Figure 7: Uncurated samples for *unconditional* generation by the one-step data-free baseline UID trained on CIFAR-10. Quantitative results are reported in Table 2.

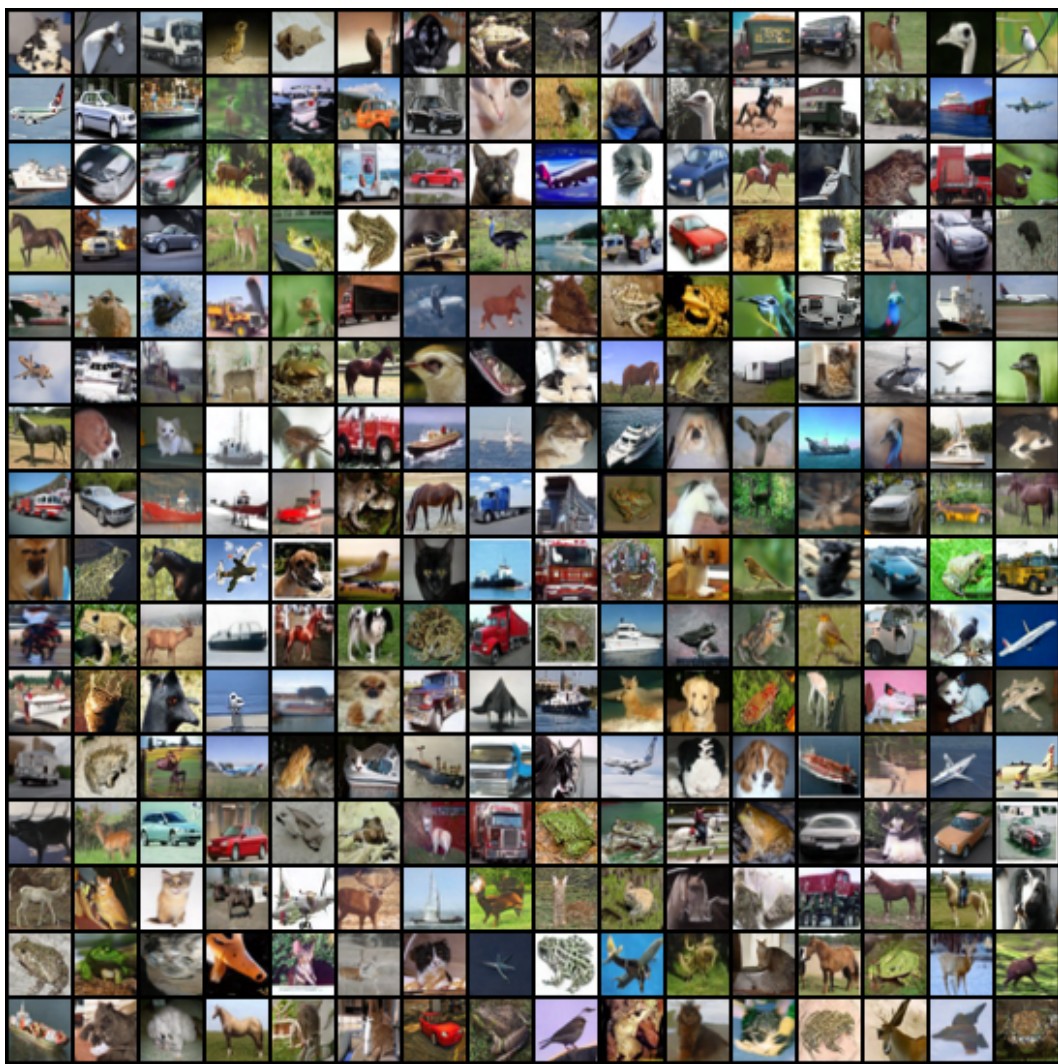

Figure 8: Uncurated samples for *unconditional* generation by the one-step UID + GAN ($\lambda_{\text{adv}}^{G_\theta} = 0.3, \lambda_{\text{adv}}^{D} = 1 | \lambda_{\text{FT}}^{G_\theta} = 25, \lambda_{\text{FT}}^{D} = 75$) trained on CIFAR-10. Quantitative results are reported in Table 2.

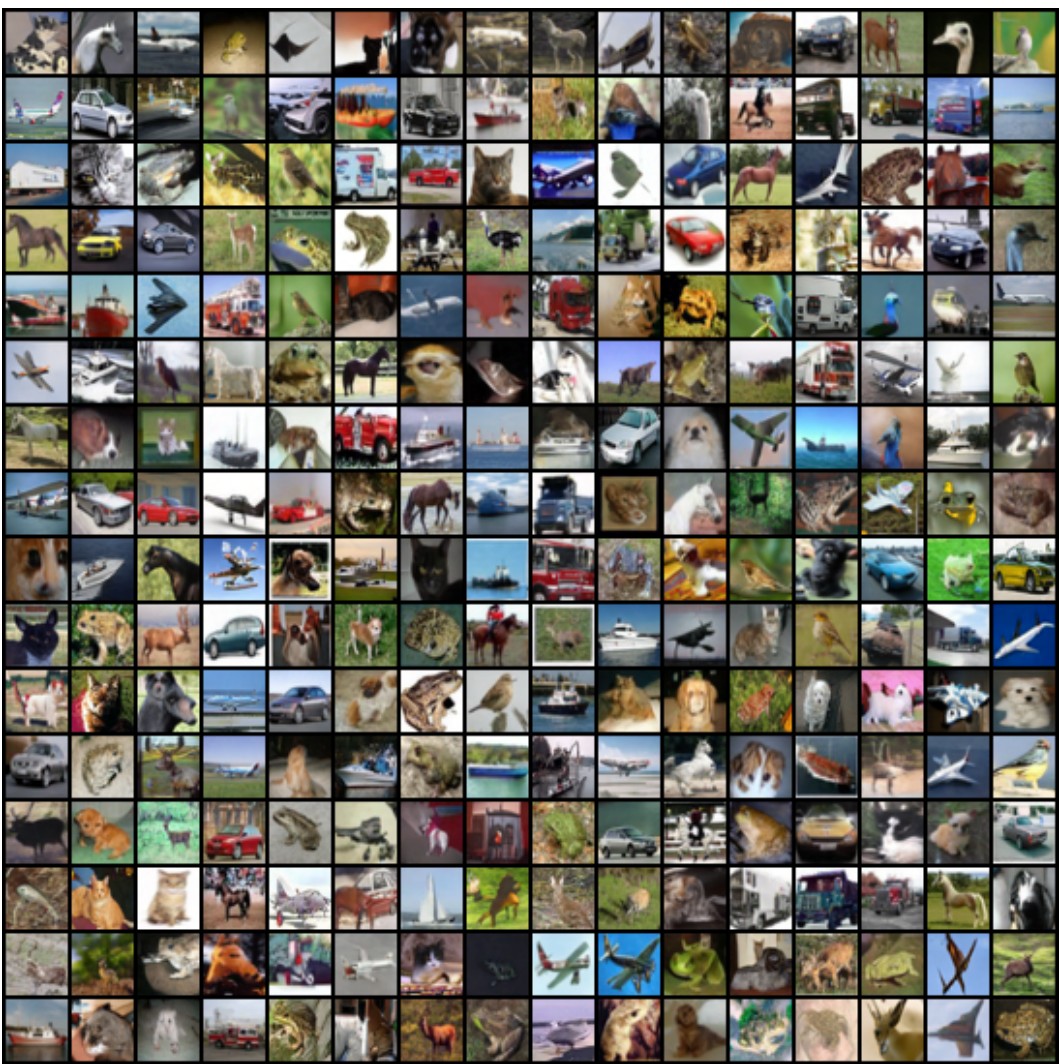

Figure 9: Uncurated samples for *unconditional* generation by **our** one-step RealUID ($\alpha = 0.92, \beta = 0.94 \mid \alpha_{\text{FT}} = 0.92, \beta_{\text{FT}} = 0.86$) trained on CIFAR-10. Quantitative results are reported in Table 2.

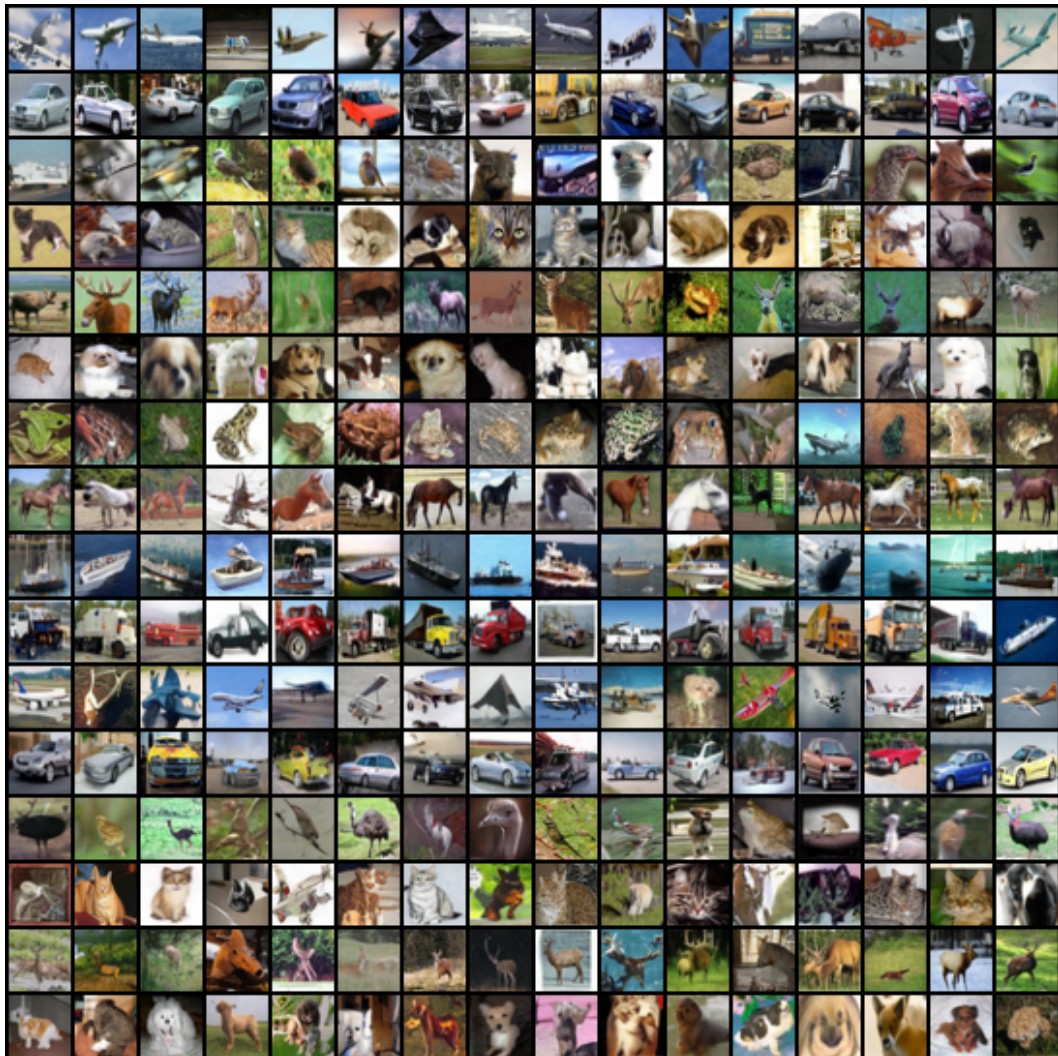

Figure 10: Uncurated samples for *conditional* generation by the one-step data-free baseline UID trained on CIFAR-10. Quantitative results are reported in Table 2.

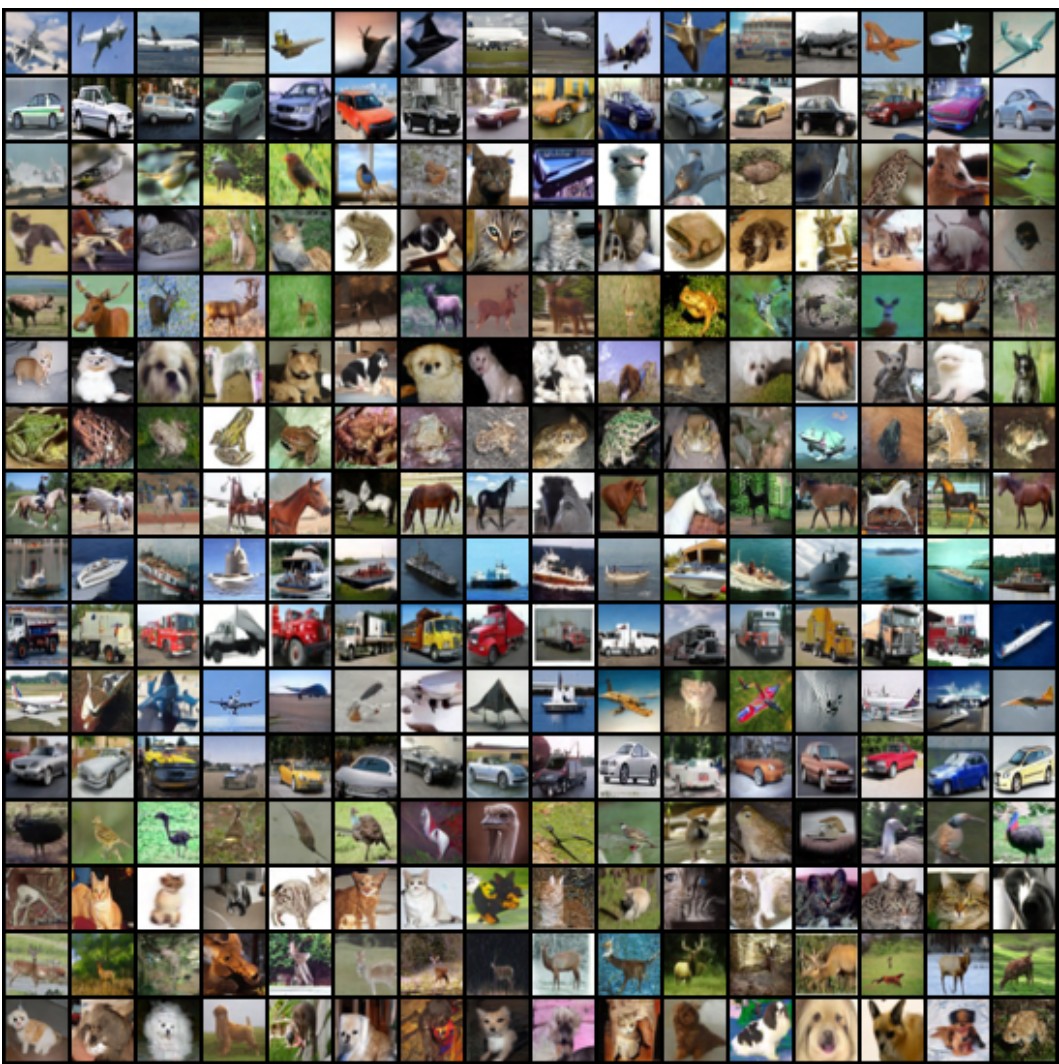

Figure 11: Uncurated samples for *conditional* generation by the one-step UID + GAN ($\lambda_{\text{adv}}^{G_\theta} = 0.3, \lambda_{\text{adv}}^{D} = 1 | \lambda_{\text{FT}}^{G_\theta} = 25, \lambda_{\text{FT}}^{D} = 75$) trained on CIFAR-10. Quantitative results are reported in Table 2.

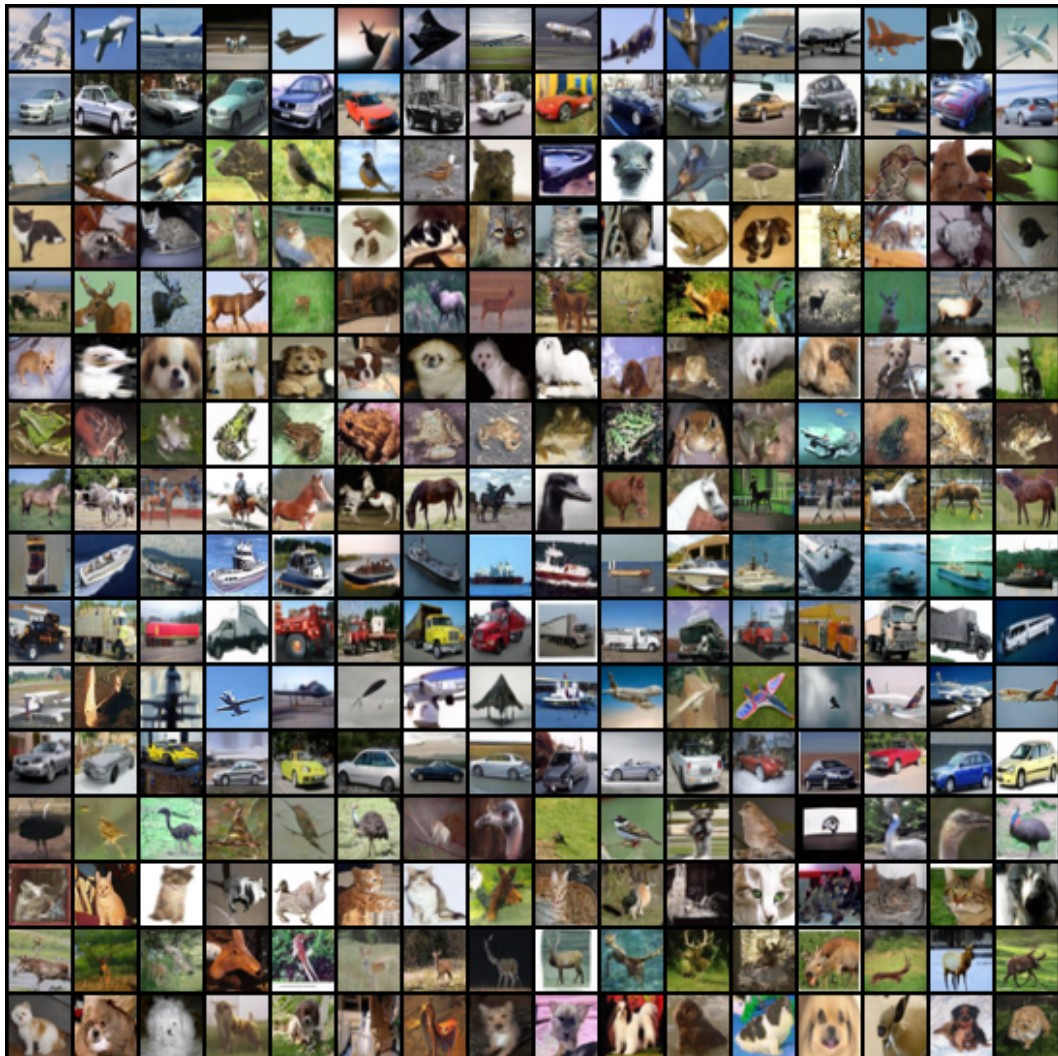

Figure 12: Uncurated samples for *conditional* generation by **our** one-step RealUID ($\alpha = 0.98, \beta = 0.96 \mid \alpha_{\text{FT}} = 0.96, \beta_{\text{FT}} = 0.92$) trained on CIFAR-10. Quantitative results are reported in Table 2.

### D.5.2 CELEBA GENERATED IMAGES

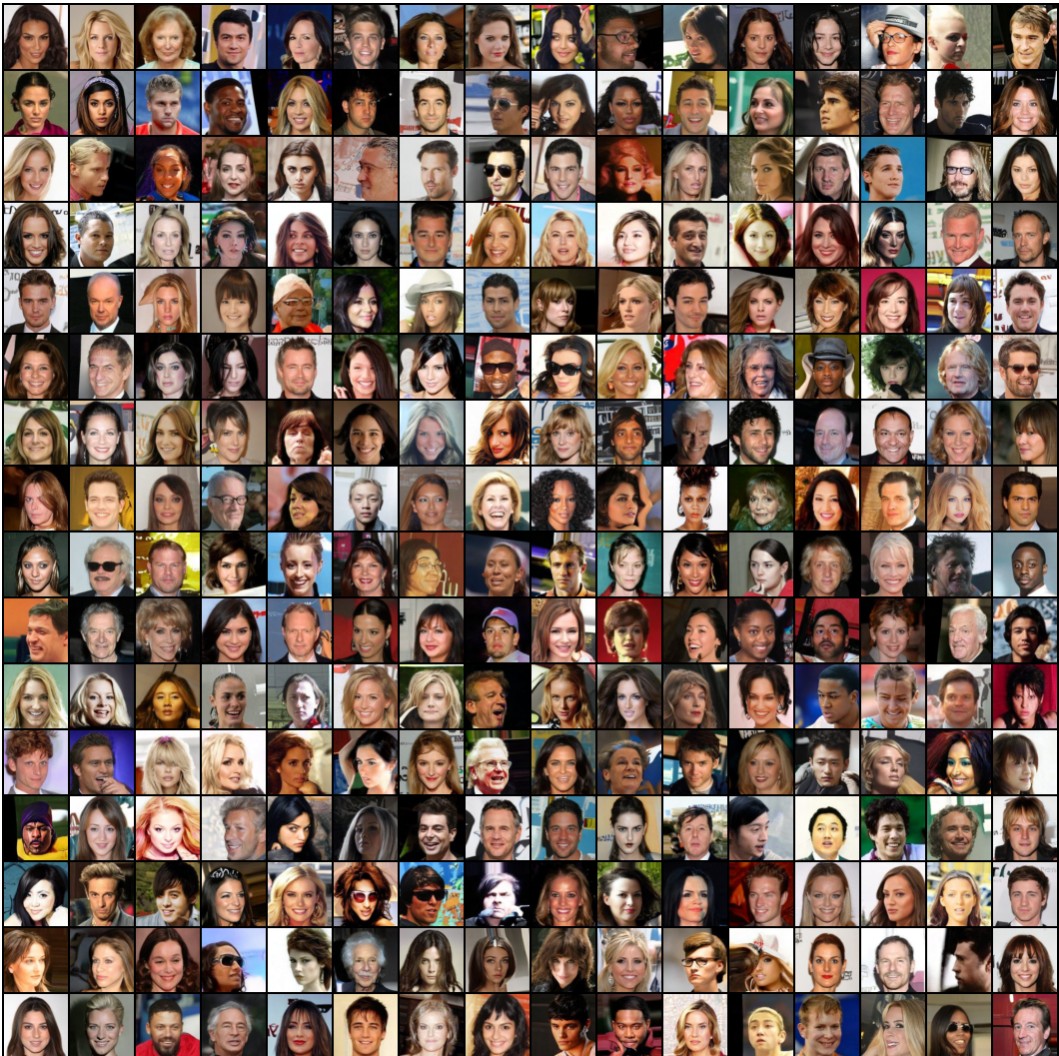

Figure 13: Uncurated samples by the one-step data-free baseline UID trained on CelebA. Quantitative results are reported in Table 7.

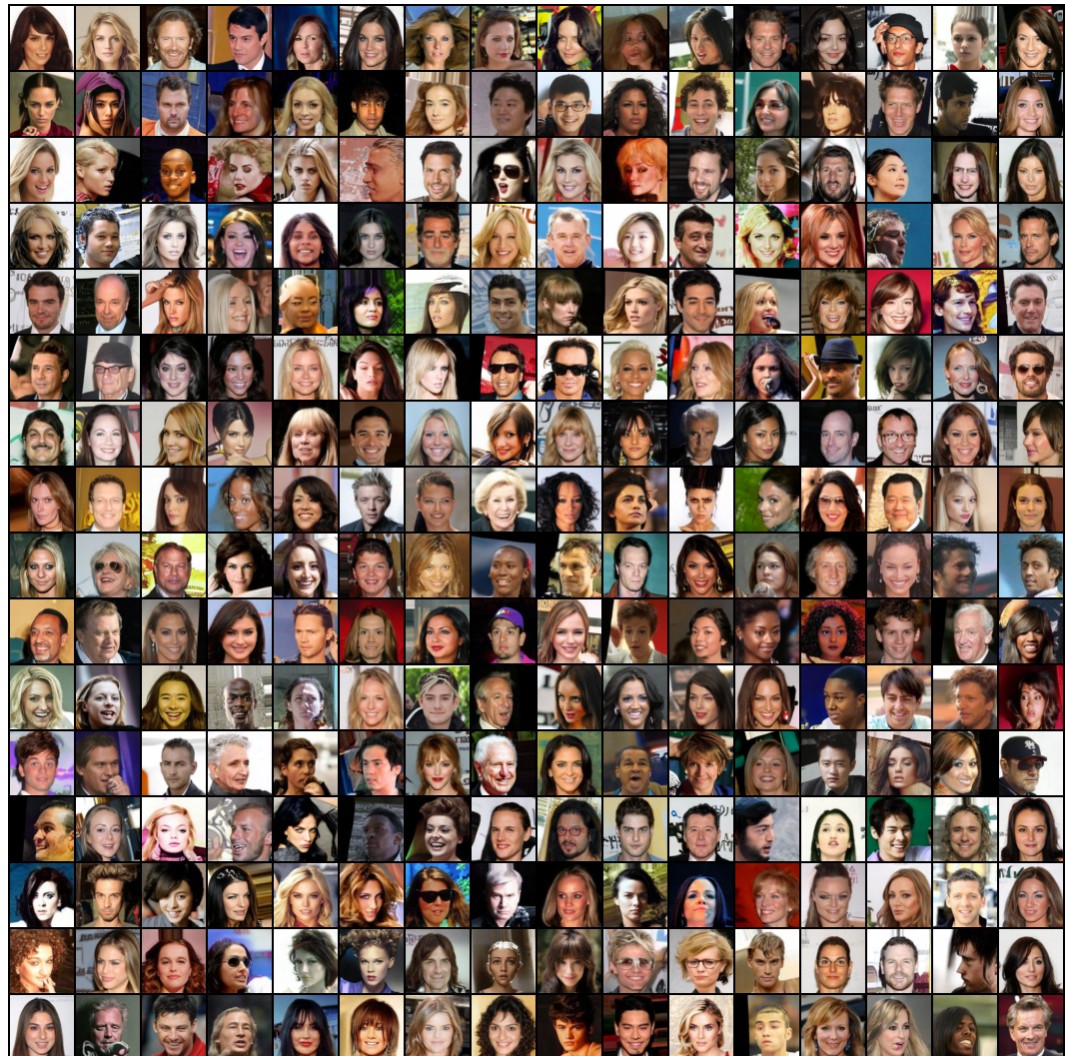

Figure 14: Uncurated samples by the one-step UID + GAN ($\lambda_{\text{adv}}^{G_\theta} = 1.0, \lambda_{\text{adv}}^{D} = 3.0$) trained on CelebA. Quantitative results are reported in Table 7.

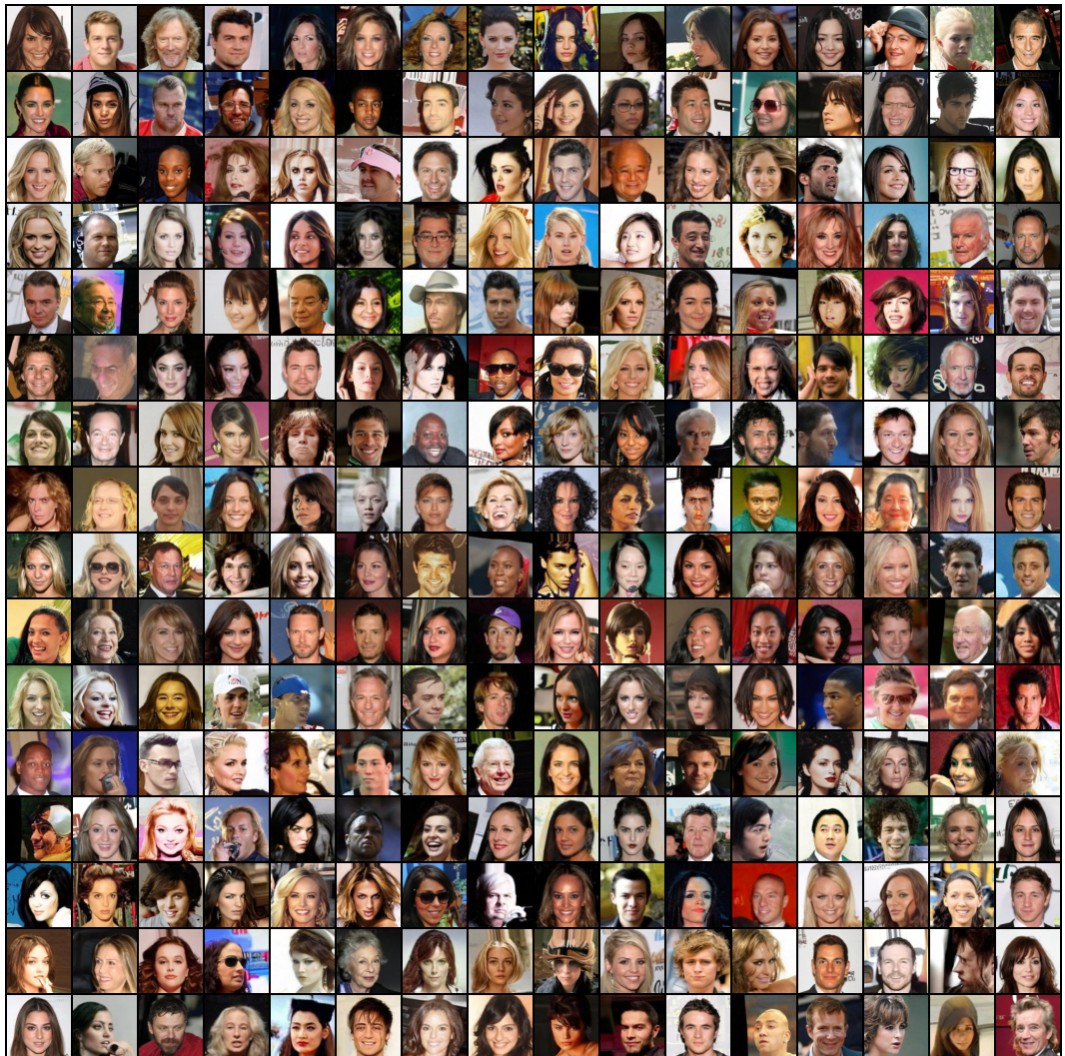

Figure 15: Uncurated samples by **our** one-step RealUID ($\alpha = 0.88, \beta = 0.9$) trained on CelebA. Quantitative results are reported in Table 7.

