# OpenReview forum: "Universal Inverse Distillation for Matching Models with Real-Data Supervision (No GANs)"
_ICLR.cc/2026/Conference — ICLR 2026 Oral_

### Official Review · Reviewer_pZHj · 2025-10-27

**Soundness:** 3
**Presentation:** 3
**Contribution:** 3
**Rating:** 8
**Confidence:** 3

**Summary:**

This is a paper about generative modeling, in which the main contribution is the incorporation of real data into the loss function or the training process. The goal is to overcome the slow sampling process in diffusion models and flow matching models, which are collectively called match models in the paper. The proposed method incorporates real data into the distillation loss, thus, it removes the need to use GAN for assessing the adversarial loss. Experiments were performed on CIFAR-10 and showed that the proposed method did have faster inference.

**Strengths:**

Stengths include the incorporation of real data into the distillation loss, thus avoiding the need for extra adversarial loss, which in turn, requires a discriminator network as in GAN. This is a logic and novel idea of implementing the distillation loss, as compared with existing approaches. Another strength is the reduced inference time, as shown in Table 3, second row from the bottom.

**Weaknesses:**

It seems Equation 17 is the key formula of the work as it defines the new distillation loss, which weighs the loss or contribution from real data by hyperparameters alpha and beta. It is unclear why alpha and beta are, first, confined to a small range such as between 0.94 and 1.0, and, second, why alpha and beta are so close to each other in most cases. What does that mean for understanding the distillation loss? Does it mean that the contribution from the real data is typically given a small weight since it is multiplied by (1-alpha)?
In the ablation study, Table 2, second row, when alpha = 1, beta=1 in this case, shouldn't the RealUID's performance equal that of the reference (the first row, Teacher Flow) since according to equation 17, when alpha=1, the second term is 0?

**Questions:**

Please see weaknesses.
It is unclear why on line 294, the paper requires that "A key constraint is that the loss must still yield the same teacher upon minimization on the real data." With the inclusion of real data, shouldn't the teacher model be somehow different from the teacher model without using the real data?
The goal of equation 13 is to maximize it, which is the last term of equation 10. Does maximizing equation 13 guarantee that equation 10, as a whole, will be minimized as equation 10 has two other terms that depend on f_t^star and f_t^theta?

---

> ### Author Response · Authors · 2025-11-22
>
> Dear Reviewer pZHj, thank you for positive evaluation of our work. Please find below the answers to your questions and comments.
>
> **W1.  It is unclear why alpha and beta are, first, confined to a small range such as between 0.94 and 1.0, and, second, why alpha and beta are so close to each other in most cases.**
>
> Following your question, we explained more thoroughly why $\alpha,\beta$ are chosen this way in the revision Section 3.4.
>
> Our Lemma 2 shows that, instead of $\alpha$ and $\beta$, actually factors $\alpha$ and $\beta/\alpha$ determine the balance between generated and real data in the minimized distance:
> $$\mathcal{L}^{\alpha, \beta}\_{\text{R-UID}}(f, p\_0^\theta)  = \mathbb{E}\_{t \sim [0,T]}\mathbb{E}\_{x\_t \sim p\_t^\*}[ \frac{||  \frac{\beta}{\alpha}\cdot [  p\_t^\*(x\_t^\*)f^\*\_t(x\_t^\*) -   p\_t^\theta(x\_t^\*) f^\theta\_t(x\_t^\*)] + (p\_t^\theta(x\_t^\*) - p\_t^*(x\_t^\*)) \cdot f^\*\_t(x\_t^\*)||^2}{p\_t^\*(x\_t^\*)((1-\alpha)p\_t^\*(x\_t^\*) + \alpha p\_t^\theta(x\_t^\*))/\alpha^2 }].$$
>
> Furthermore, coefficient $\alpha$ only sets the general scaling of the distance, while \textit{$\frac{\beta}{\alpha}$ plays the most important role}, as it determines the relation between $f^\theta_t$ and $f^*_t$ inside the distance.
>
>  1) Too far away values $ \frac{\beta}{\alpha} \ll 1$ diminish the effect of the generated data term $f_t^\theta$, while  $\frac{\beta}{\alpha} \gg 1$ diminish the effect of the right real data term $f^*_t$.
>
>  2)  The parameters $\alpha, \beta$ are confined to small range to control the effect of real data during fake model training. The fake model learns both generator current state and real data simultaneously. Too much real data can distract the fake model from learning  generator gradients which prevents effective backpropagation. Theoretically, the optimal fake model is expressed via Equation (19) in the revision:
> $$f\_t(x\_t) = \frac{ f^\*\_t(x\_t) \cdot (1 - \beta) + f^\theta\_t(x\_t) \cdot \beta \frac{p\_t^\theta(x\_t)}{p\_t^\*(x\_t)} }{(1-\alpha) + \alpha \frac{p\_t^\theta(x\_t)}{p\_t^*(x\_t)}}.$$
>  With small  $\alpha, \beta$, the generator-dependent terms diminish as $\alpha, \beta \to 0$, and the fake model cannot capture the gradients with respect to the generator's parameters.
>
>
>
>  Thus, efficient mix of real and generated data happens only when $\alpha, \beta$ are close to $1$ with small non-zero difference.
>
>
>
> **W2. In the ablation study, Table 2, second row, when alpha = 1, beta=1 in this case, shouldn't the RealUID's performance equal that of the reference?**
>
> Even in the data-free setup $(\alpha = \beta = 1)$, the SiD-like distillation methods are known to perform better than the teacher, as was noted in the original SiD paper [1]. The teacher  learns a complex vector field over the entire time and space interval, not the data distribution itself. Moreover, it accumulates errors due to its multi-step sampling procedure which considerably reduces output quality. In contrast, the generator produces data directly.
>
>  [1] Mingyuan Zhou, et. al. Score identity
> distillation: Exponentially fast distillation of pretrained diffusion models for one-step generation.
>
> **Q1. With the inclusion of real data, shouldn't the teacher model be somehow different from the teacher model without using the real data?**
>
> Could you please clarify this question?
>
>
> **Q2.Does maximizing equation 13 guarantee that equation 10, as a whole, will be minimized as equation 10 has two other terms that depend on $f_t^*$ and $f_t^\theta$?**
>
>  We believe there may be a misunderstanding regarding Equations (10) and (13). The Equation (13) when maximized equals to the entire Equation (10), not only its last term.

---

### Official Review · Reviewer_WywQ · 2025-10-31

**Soundness:** 2
**Presentation:** 2
**Contribution:** 2
**Rating:** 4
**Confidence:** 3

**Summary:**

This paper introduces the RealUID framework, with two primary contributions: (1) it theoretically unifies prior distillation methods for different matching models (e.g., SiD, FGM) and extends them to Bridge Matching and Stochastic Interpolants via a novel UID loss based on a linearization technique; (2) it proposes a novel method for incorporating real data into the distillation process without relying on GANs. Experiments on CIFAR-10 provide empirical support for these theoretical claims.

**Strengths:**

1.The authors constructed a concise unified form through a linearization technique, and the theoretical derivation is fundamentally sound;
2.The extension of the framework to bridge matching and stochastic interpolants is commendable;
3.Replacing complex GANs with simple loss interpolation to incorporate real data is a straightforward yet effective approach.

**Weaknesses:**

1.The proposed unified framework appears more as a formal re-formulation than a substantive theoretical breakthrough. This raises concerns that the core contribution may essentially be a "re-parameterization" of prior methods like SID and FGM, thereby undermining its claimed novelty and independence.
2.The proposed RealUID method suffers from a critical dependence on two newly introduced hyperparameters, α and β. Firstly, the selection of these parameters lacks clear theoretical guidance, and their design appears more as a heuristic afterthought rather than a principled derivation. Secondly, as evidenced in Table 1 and Appendix Table 6, the method exhibits extreme sensitivity and pronounced randomness to these parameters.
3.Lemma 2 relies on the assumption that the data probability density is positive everywhere (pt∗(xt) > 0, ∀xt, t). This assumption typically fails in high-dimensional spaces.
4.The most prominent weakness of this work is its narrow experimental setup:
1)The evaluation is narrowly based on small-scale datasets like CIFAR-10. The absence of results on standard large-scale, high-resolution benchmarks such as AFHQ and ImageNet severely limits the empirical grounding of the paper's claims.
2)The experimental evaluation is confined to  flow-based models, omitting tests on diffusion models. This limitation raises serious concerns about the method's competitive advantage and the framework's generalizability.
3)The experiments lack comparisons against state-of-the-art baselines from 2025.

**Questions:**

1.Beyond mathematical re-formulation, what are the new insights and practical improvements uniquely enabled by your unified UID loss perspective that were not apparent or easily achievable within the prior frameworks.
2.Is there any adaptive criterion for tuning α and β? What is the rationale behind using different parameter values during the training versus fine-tuning stages? Do these hyperparameters generalize across datasets, or do they require costly re-tuning for each new dataset?
3.Why does RealUID use a lightweight model architecture different from the baseline methods? Could increasing the parameter count further improve performance?
4.Could the authors provide additional results of RealUID on other datasets and models? This is my primary concern.

---

> ### Author Response · Authors · 2025-11-22
>
> Dear Reviewer WywQ, thank you for your feedback. Please find below the answers to your questions and comments.
>
> **W1. The evaluation is narrowly based on small-scale datasets like CIFAR-10.**
>
> To test the scalability of our RealUID, we conducted the same experiments on the larger and more complex CelebA 64x64 dataset and architecture. Due to limited time, we ran all experiments only for 400k iterations. They all can achieve better metrics, and we can train them longer if needed. The results are presented in the table below and in the Appendix E.2 in the revision.
>
> | α\β| 0.94 | 0.96 | 0.98 | 1.0|
> |---|---|--|--|--|
> | 0.94 | 2.60 | 1.93 | 2.13  | 2.53 |
> | 0.96| **1.70** | 2.77 | 2.00  | 2.47 |
> | 0.98| 2.16  | 2.04  | 2.62 | 2.42 |
> | 1.0 | 2.96  | 2.48  | 2.23 | 2.62  |
>
> Once again, our RealUID significantly outperforms its data-free counterpart. __The best parameters $\alpha, \beta$ remain the same for both Cifar10 and CelebA, demonstrating stability over various setups.__
>
> **W2. The proposed unified framework appears more as a formal re-formulation than a substantive theoretical breakthrough.**
>
> First, we generalize the distillation ideas, propose new perspective and provide a new, much simpler and more universal proof. Thanks to our work, one can easily build distillation methods for any existing or new matching models.
>
> Second, our new inverse optimization perspective allows us to incorporate real data in the distillation procedure despite the model type and theoretically justify it.
>
> Third, our new proofs techniques enable future loss modifications, see Appendix A for more details.
>
> Since our basic data-free UID includes SiD and FGM, it is identical to them in practice.
>
> **W3. New parameters lacks clear theoretical guidance, and their design appears more as a heuristic afterthought rather than a principled derivation.**
>
> The two coefficients and the RealUM loss structure arise from the purely theoretical requirement to recover the same teacher model  for all $\alpha, \beta$, so we could use pre-trained teacher models and not retrain them for each $\alpha, \beta$ individually. (Section 3.4.)
>
> In comparison, a GAN has numerous parameters to tune: a generator loss factor over a large range, a discriminator loss factor over a large range, a choice of discriminator architecture, and the range of timesteps used in the discriminator. *This makes our method significantly simpler to tune*, as it has fewer parameters with a clear interpretation.
>
>
> **W4. Secondly, as evidenced in Table 1 and Appendix Table 6, the method exhibits extreme sensitivity and pronounced randomness to these parameters.**
>
> Following your question, we explained more thoroughly how to choose good $\alpha,\beta$ in the revision Section 3.4.
>
> Our Lemma 2 shows that, instead of $\alpha$ and $\beta$, actually factors $\alpha$ and $\beta/\alpha$ determine the balance between generated and real data in the minimized distance:
>
> $$\mathcal{L}^{\alpha, \beta}\_{\text{R-UID}}(f, p\_0^\theta)  = \mathbb{E}\_{t \sim [0,T]}\mathbb{E}\_{x\_t \sim p\_t^\*}[ \frac{||  \frac{\beta}{\alpha}\cdot [  p\_t^\*(x\_t^\*)f^\*\_t(x\_t^\*) -   p\_t^\theta(x\_t^\*) f^\theta\_t(x\_t^\*)] + (p\_t^\theta(x\_t^\*) - p\_t^*(x\_t^\*)) \cdot f^\*\_t(x\_t^\*)||^2}{p\_t^\*(x\_t^\*)((1-\alpha)p\_t^\*(x\_t^\*) + \alpha p\_t^\theta(x\_t^\*))/\alpha^2 }].$$
>
> Furthermore, coefficient $\alpha$ only sets the general scaling of the distance, while *$\frac{\beta}{\alpha}$ plays the most important role*, as it determines the relation between $f^\theta_t$ and $f^*_t$ inside the distance.
>
> We provide theoretical analysis of this distance in Section 3.4. It states that the good $\alpha, \beta \in (0,1]$ can be chosen by first finding $\frac{\beta}{\alpha} \neq 1$, as it has the largest impact, and then adjusting $\alpha < 1$. Both $\frac{\beta}{\alpha}$ and $\alpha$ should be close to $1.$
>
> Hence, we reorganized the results table in terms of $\alpha$ and $\beta/\alpha$ to confirm that
>
> 1) $\beta/\alpha$ brings the largest impact (and can even be adjusted solely with a fixed $\alpha$),
>
> 2)  small non-zero values $\beta/\alpha -1$ consistently lead to a better performance, while  large values lead to a worse one.
>
> | Generation  | α \ (α/β) | 1.06  | 1.04  | 1.02  | 1.0   | 0.98  | 0.96  | 0.94  |
> |--|--|---|---|--|--|---|-|---|
> | **Unconditional** | **0.94** | -     | -     | -     | 2.66  | **2.28** | 2.58  | 2.98  |
> |  | **0.96** | -  | -  | 2.37  | 2.58  | 2.29  | 2.65  | - |
> | | **0.98** | - | 2.97  | 2.33  | 2.62  | 2.38  | - | - |
> | | **1.0**  | 5.81  | 4.51  | 3.29  | 2.58 | - | - | - |
> | **Conditional**  | **0.94** | -| - | - | 2.35  | 2.19  | 2.25  | 2.47  |
> |  | **0.96** | - | -| 2.09  | 2.32  | 2.13  | 2.27  | -  |
> |   | **0.98** | -| 2.34  | **2.02** | 2.26  | 2.05  | -     | - |
> |   | **1.0**  | 4.32  | 3.27  | 2.43  | 2.21  | -  | - | -|

---

> > ### Author Response · Authors · 2025-11-22
> >
> > **W5. Lemma 2 relies on the assumption (pt∗(xt) > 0, ∀xt, t).**
> >
> >
> >  This assumption is made only for the simplicity and conciseness of the obtained results. Lemma 2 can be proved without it, but the distance formula Equation (20) will have an additional mathematical expectation for the points $x_t$ that have $p_t^*(x_t) = 0$ and $p_t^\theta(x_t) > 0$. Theorems 1 and 2 can be proved without this assumption using inverse optimization perspective.
> >   Moreover, since real data is typically noised with Gaussian noise in the forward process,  the assumption holds for all $t \in (0,1]$ and $t=0$ is neglectable.
> >
> >
> >
> > **W6. The experimental evaluation is confined to flow-based models, omitting tests on diffusion models.**
> >
> > As shown in papers [1,2], diffusion models and flow matching are essentially the same. Due to our resource constraints, we decided to focus only on flow matching models as they are less studied in the distillation field.
> >
> > [1] Peter Holderrieth, Marton Havasi, Jason Yim, Neta Shaul, Itai Gat, Tommi Jaakkola, Brian Karrer,
> > Ricky TQ Chen, and Yaron Lipman. Generator matching: Generative modeling with arbitrary
> > markov processes. arXiv preprint arXiv:2410.20587, 2024
> >
> >
> > [2] Ruiqi Gao, Emiel Hoogeboom, Jonathan Heek, Valentin De Bortoli, Kevin Patrick Murphy, and Tim
> > Salimans. Diffusion models and gaussian flow matching: Two sides of the same coin. In The
> > Fourth Blogpost Track at ICLR 2025, 2025
> >
> >
> > **W7. The experiments lack comparisons against state-of-the-art baselines from 2025.**
> >
> >  We primarily compared our RealUID within the class of SiD-like distillation methods. In the revision, we added SOTA results for consistency and flowmap-models that we outperform.
> >
> >
> >
> > **Q1. Is there any adaptive criterion for tuning $\alpha, \beta$?**
> >
> > We do not know any adaptive criterion, but we propose and justify the simple guidance for $\alpha, \beta$ in the W4 reply.
> >
> > **Q2. What is the rationale behind using different parameter values during the training versus fine-tuning stages?**
> >
> >   The fine-tuning stage (Table 6) considerably differs from training from zero, since we use the teacher model as the initializer for the student model, rather than a model we pre-trained, and we use an EMA generator. These models are not well-aligned which requires much more real data (larger value $\beta/\alpha$) to align them. For example, in the data-free setup, the model fails to converge at all.
> >
> >
> > **Q3. Why does RealUID use a lightweight model architecture different from the baseline methods?**
> >
> >
> > With our resources, baseline SiDA needs 15 days to achieve our metrics (we ran it for only 2 days) and twice that to achieve its SOTA performance. Since the main goal of our methodological work is to demonstrate the universal theory behind all distillation methods and test a new approach to incorporate real data, we decided to take a faster architecture.
> >
> >
> > **Q4. Could increasing the parameter count further improve performance?**
> >
> > Since our main goal is not to achieve the best results on every setup, we leave this technical experiment for future work.
> >
> >
> >
> > **Concluding remarks.** We would be grateful if you could let us know if our explanations have been satisfactory. If so, we kindly ask that you consider increasing your rating. We are also open to discussing any other questions you may have.

---

### Official Review · Reviewer_ha8b · 2025-11-03

**Soundness:** 4
**Presentation:** 2
**Contribution:** 3
**Rating:** 6
**Confidence:** 5

**Summary:**

The paper tackles the challenge of distilling diffusion/flow-based models into fast one-step generators. The authors build on works such as SiD and FGM, which optimize the student by minimizing the $\ell_2$ distance between the teacher and student score/velocity functions, referred to as $f$-functions in the paper. The authors propose a simple but clever modification to the training objective of the student $f$-function that includes both real and generated data, allowing for an alternative way to incorporate real data into the distillation pipeline of these methods. They theoretically justify this change by proving that it has a proper global minimizer and experimentally demonstrate its usefulness on CIFAR-10, achieving impressive results.

**Strengths:**

- The premise of the paper is very interesting and important. Distillation methods that rely on Reverse-KL-like objectives (SiD, FGM, DMD, …) are widely used in practice but are known to suffer from a diversity collapse problem. This issue arises mainly because the loss is computed only on student samples (reverse-KL). Therefore, being able to effectively incorporate real data into training without resorting to GANs (which have famously unstable training dynamics) is potentially very valuable.
- The reasoning that leads to the RealUM and RealUID objectives is clever.
- The supplementary material is extensive and includes additional variants of the approach, as well as theorems and proofs.

**Weaknesses:**

- As mentioned above, loss of diversity is a common drawback of distillation approaches with Reverse-KL-like objectives. Incorporating real data into the training pipeline can, in principle, fix this issue to some extent. Therefore, including diversity metrics in the experiments would be very helpful and could strengthen the paper’s argument.
- The paper broadly argues that recent diffusion distillation methods either rely solely on synthetic data or use real data only in GAN-like objectives. While this is true for most Reverse-KL-like methods, it does not hold for consistency- or flowmap-based methods[1,2,3,4,5]. Including a discussion of such methods and adding them as additional baselines would help the paper better position itself within the literature.
- Although the authors explicitly note that the limited scope of the experiments is due to computational constraints, the fact remains that the proposed approach is primarily evaluated on a small dataset using a lightweight network architecture. As a result, it is unclear whether the conclusions drawn will generalize to larger datasets and models. For example, as shown in Table 4, the proposed approach performs considerably worse than prior SiD and SiDA baselines.
- The method seems highly sensitive to the values of $\alpha,\beta$, as shown in Table 1.


[1] Song, Yang, et al. "Consistency models." (2023). \
[2] Lu, Cheng, and Yang Song. "Simplifying, stabilizing and scaling continuous-time consistency models." arXiv preprint arXiv:2410.11081 (2024). \
[3] Frans, Kevin, et al. "One step diffusion via shortcut models." arXiv preprint arXiv:2410.12557 (2024). \
[4] Geng, Zhengyang, et al. "Mean flows for one-step generative modeling." arXiv preprint arXiv:2505.13447 (2025). \
[5] Sabour, Amirmojtaba, Sanja Fidler, and Karsten Kreis. "Align Your Flow: Scaling Continuous-Time Flow Map Distillation." arXiv preprint arXiv:2506.14603 (2025).

**Questions:**

- Why does the setting too low values for the coefficients $\alpha,\beta$ lead to collapse? Is this just experimentally seen or is there a theoretical justification for it?

---

> ### Author Response · Authors · 2025-11-22
>
> Dear Reviewer ha8b, thank you for your comments. Please find below the answers to your questions and comments.
>
> **W1. It is unclear whether the conclusions drawn will generalize to larger datasets and models.**
>
>
> To test the scalability of our RealUID, we conducted the same experiments on the larger and more complex CelebA 64x64 dataset and architecture. Due to limited time, we ran all experiments only for 400k iterations. They all can achieve better metrics, and we can train them longer if needed. The results are presented in the table below and in the Appendix E.2 in the revision.
>
>
> | α\β   | 0.94  | 0.96  | 0.98  | 1.0   |
> |-|-|-|--|-|
> | 0.94 | 2.60 | 1.93 | 2.13| 2.53 |
> | 0.96| **1.70** | 2.77 | 2.00  | 2.47  |
> | 0.98| 2.16  | 2.04  | 2.62 | 2.42 |
> | 1.0 | 2.96  | 2.48  | 2.23 | 2.62|
>
> Once again, our RealUID significantly outperforms its data-free counterpart. __The best parameters $\alpha, \beta$ remain the same for both Cifar10 and CelebA, demonstrating stability over various setups.__
>
> **W3. The method seems highly sensitive to the values of $\alpha, \beta$, as shown in Table 1.**
>
> Following your question, we explained more thoroughly how to choose good $\alpha,\beta$ in the revision Section 3.4.
>
> Our Lemma 2 shows that, instead of $\alpha$ and $\beta$, actually factors $\alpha$ and $\beta/\alpha$ determine the balance between generated and real data in the minimized distance:
>
> $$\mathcal{L}^{\alpha, \beta}\_{\text{R-UID}}(f, p\_0^\theta)  = \mathbb{E}\_{t \sim [0,T]}\mathbb{E}\_{x\_t \sim p\_t^\*}[ \frac{||  \frac{\beta}{\alpha}\cdot [  p\_t^\*(x\_t^\*)f^\*\_t(x\_t^\*) -   p\_t^\theta(x\_t^\*) f^\theta\_t(x\_t^\*)] + (p\_t^\theta(x\_t^\*) - p\_t^*(x\_t^\*)) \cdot f^\*\_t(x\_t^\*)||^2}{p\_t^\*(x\_t^\*)((1-\alpha)p\_t^\*(x\_t^\*) + \alpha p\_t^\theta(x\_t^\*))/\alpha^2 }].$$
>
> Furthermore, coefficient $\alpha$ only sets the general scaling of the distance, while *$\frac{\beta}{\alpha}$ plays the most important role*, as it determines the relation between $f^\theta_t$ and $f^*_t$ inside the distance.
>
> We provide theoretical analysis of this distance in Section 3.4. It states that the good $\alpha, \beta \in (0,1]$ can be chosen by first finding $\frac{\beta}{\alpha} \neq 1$, as it has the largest impact, and then adjusting $\alpha < 1$. Both $\frac{\beta}{\alpha}$ and $\alpha$ should be close to $1.$
>
> Hence, we reorganized the results table in terms of $\alpha$ and $\beta/\alpha$ to confirm that
>
> 1) $\beta/\alpha$ brings the largest impact (and can even be adjusted solely with a fixed $\alpha$),
>
> 2)  small non-zero values $\beta/\alpha -1$ consistently lead to a better performance, while  large values lead to a worse one.
>
> | Generation  | α \ (α/β) | 1.06  | 1.04  | 1.02  | 1.0   | 0.98  | 0.96  | 0.94  |
> |---|--|--|---|----|--|----|-------|-------|
> | **Unconditional** | **0.94** | -     | -     | -     | 2.66  | **2.28** | 2.58  | 2.98  |
> |  | **0.96** | -  | -  | 2.37  | 2.58  | 2.29  | 2.65  | - |
> | | **0.98** | - | 2.97  | 2.33  | 2.62  | 2.38  | - | - |
> | | **1.0**  | 5.81  | 4.51  | 3.29  | 2.58 | - | - | - |
> | **Conditional**  | **0.94** | -| - | - | 2.35  | 2.19  | 2.25  | 2.47  |
> |  | **0.96** | - | -| 2.09  | 2.32  | 2.13  | 2.27  | -  |
> |   | **0.98** | -| 2.34  | **2.02** | 2.26  | 2.05  | -     | - |
> |   | **1.0**  | 4.32  | 3.27  | 2.43  | 2.21  | -  | - | -|
>
> **W1. Including diversity metrics in the experiments would be very helpful.**
>
> In prior works on distillation and consistency models with real data [1-3], the FID score alone has been considered sufficient for fair model comparison.
>
>
> [1] Mingyuan Zhou, et al. Adversarial score identity
> distillation: Rapidly surpassing the teacher in one step.
>
> [2] Tianwei Yin, et al.  Improved distribution matching distillation for fast image synthesis.
>
> [3] Zhengyang Geng, et al.  Mean flows for
> one-step generative modeling.
>
> **W2.  Including a discussion of consistency and flowmap methods**
>
> Indeed, consistency and flowmap models use real data in the distillation mode along with the teacher. However, the role and effect of real data in these models are different from that in GANs or our RealUID.
>
> GANs and RealUID compare real and generated data directly via adversarial losses or fake model, forcing the generator to produce samples that are indistinguishable from real ones. On the other hand, the main goal of consistency models is to satisfy, at each point, a consistency equation defined by the teacher. The real data primarily determines which points are the most likely, but the final model depends only on the teacher's trajectory. For example, if the teacher is flawed, the consistency model will learn to reproduce all of its mistakes. In contrast, GANs and RealUID utilize data directly and can mitigate the errors of a flawed teacher. We have added a new analysis of the our RealUID's ability to correct teacher's errors in the revision Appendix A.1.3.
>
> In the revision, we also added new consistency baselines to the results in Table 1.

---

> > ### Author Response · Authors · 2025-11-22
> >
> > **W3. For example, as shown in Table 4, the proposed approach performs considerably worse than prior SiD and SiDA baselines.**
> >
> >
> > The SOTA performance of models is usually achieved through larger architectures, numerous practical heuristics, careful tuning of hyperparameters and significantly more compute. For example, with our resources, SiDA needs 15 days to achieve our metrics (we ran it for only 2 days) and twice that to achieve its SOTA performance.
> >
> > In our methodological work, the main goal is to demonstrate the universal theory behind all distillation methods and to test a new approach to incorporate real data. Moreover, we do not have enough resources to run SOTA experiments that will achieve even our own metrics within a reasonable time. That is why we did not try to achieve the SOTA results.
> >
> >
> > Nevertheless, for Flow Matching models, our UID achieves the SOTA performance with lightweight architecture, outperforming the FGM approach [4] (generalization of SiD for Flow Matching models) with heavy EDM architecture.
> >
> > [4] Huang, Z., Geng, Z., Luo, W., \& Qi, G. J. (2024). Flow generator matching. arXiv preprint arXiv:2410.19310.
> >
> > **Q1. Why does the setting too low values for the coefficients $\alpha, \beta$ lead to collapse? Is this just experimentally seen or is there a theoretical justification for it?**
> >
> > It can be seen theoretically as well. The optimal fake model is expressed via Equation (19) in the revision:
> > $$f\_t(x\_t) = \frac{ f^\*\_t(x\_t) \cdot (1 - \beta) + f^\theta\_t(x\_t) \cdot \beta \frac{p\_t^\theta(x\_t)}{p\_t^\*(x\_t)} }{(1-\alpha) + \alpha \frac{p\_t^\theta(x\_t)}{p\_t^\*(x\_t)}}.$$
> >  With small  $\alpha, \beta$, the generator-dependent terms diminish as $\alpha, \beta \to 0$, and the fake model cannot capture the gradients with respect to the generator's parameters. This prevents effective backpropagation.
> >
> > **Concluding remarks.** We would be grateful if you could let us know if our explanations have been satisfactory. If so, we kindly ask that you consider increasing your rating. We are also open to discussing any other questions you may have.

---

> > > ### Comment · Reviewer_ha8b · 2025-11-27
> > >
> > > I appreciate the authors’ response, especially the additional experiments and the clarification regarding the $\alpha,\beta$ hyperparameters. On the topic of diversity, while it is true that most distillation works do not report this metric, it remains a well-known weakness of DMD-style distillation. If RealUID can alleviate this issue, that would be a significant contribution. For this reason, if feasible, I would still strongly encourage the authors to include diversity metrics in their evaluations for the camera-ready version.
> > >
> > > Overall, my main concerns have been addressed, and I will be raising my score.

---

> > > > ### Author Response · Authors · 2025-11-28
> > > >
> > > > Dear Reviewer ha8b,
> > > >
> > > > To address your concern regarding diversity metrics, we have calculated Inception Score, Precision and Recall for the teacher model, the data-free model, the best GAN model, and our best RealUID. The results for both unconditional  and conditional CIFAR-10 are presented below.
> > > >
> > > > **Unconditional:**
> > > >
> > > > | model | Inception Score       | Precision           | Recall             | FID                |
> > > > |---------------|-----------------------|---------------------|--------------------|--------------------|
> > > > | RealUID 0.94 / 0.96   | **9.82 ± 0.06**       | 0.652 ± 0.002     | **0.611 ± 0.002**    | **2.26 ± 0.02**    |
> > > > | RealUID 1.0 / 1.0 + GAN 0.3 / 1.0 | 9.80 ± 0.04 | 0.657 ± 0.002     | 0.607 ± 0.002    | 2.28 ± 0.01    |
> > > > | RealUID 1.0 / 1.0     | 9.69 ± 0.03       | **0.660 ± 0.002**     | 0.601 ± 0.003    | 2.57 ± 0.02 |
> > > > | teacher       | 8.90     ± 0.05            | 0.655 ± 0.001              | 0.585 ± 0.001             | 4.43    ± 0.06         |
> > > >
> > > >
> > > > **Conditional:**
> > > >
> > > > | model     | Inception Score       | Precision           | Recall             | FID                |
> > > > |-------------------|-----------------------|---------------------|--------------------|--------------------|
> > > > | RealUID 0.98 / 0.96       | **9.90 ± 0.03**       | **0.650 ± 0.001**     | **0.617 ± 0.001**    | **2.01 ± 0.01**    |
> > > > | RealUID 1.0 / 1.0  + GAN 0.3 / 1.0 | 9.86 ± 0.03 | 0.645 ± 0.003     | **0.617 ± 0.001**    | 2.14 ± 0.01    |
> > > > | RealUID 1.0 / 1.0         | 9.81 ± 0.04       | 0.645 ± 0.002     | 0.616 ± 0.003    | 2.23 ± 0.01    |
> > > > | teacher           | 8.89      ± 0.02          | 0.637 ± 0.001              | 0.604 ± 0.001             | 5.50 ± 0.04             |
> > > >
> > > >
> > > > In most cases, *our RealUID achieves better or competitive metrics* compared to the data-free and GAN counterparts, and a significant improvement over the teacher model.

---

### Official Review · Reviewer_oamB · 2025-11-03

**Soundness:** 3
**Presentation:** 3
**Contribution:** 3
**Rating:** 6
**Confidence:** 4

**Summary:**

This paper addresses a core problem in generative model distillation: existing data-free methods (e.g., SiD, FGM) cannot leverage real data to improve the student model, and the only current way to do so relies on complex and unstable GANs.The paper's contributions are twofold:
1. **Theoretical Unification (UID)**: It introduces a general min-max distillation framework (UID) and demonstrates that recent methods like SiD and FGM are special cases of this unified structure.
2. **Methodological Innovation (RealUID)**: This is the paper's main contribution. The authors propose RealUID, a novel, non-adversarial (GAN-free) mechanism to integrate real-data supervision. This is achieved by cleverly redesigning the inner loss function of the min-max game ($\mathcal{L}_{R-UM}^{\alpha,\beta}$) to include terms from both the generated distribution ($p_0^\theta$) and the real-data distribution ($p_0^*$).

Experiments (on CIFAR-10) show that RealUID not only outperforms the data-free baseline in FID but also achieves comparable or superior results to GAN-based methods, all while significantly accelerating convergence and avoiding the complexities of adversarial training.

**Strengths:**

1. **Significant Methodological Innovation (GAN-free Supervision)**: The paper's strongest contribution is RealUID, a novel and elegant non-adversarial method to incorporate real-data supervision into the distillation process. This directly addresses a major pain point of previous work, avoiding the need for extra discriminator architectures and the instability of adversarial training.
2. **Clever Loss Function Design ($\mathcal{L}_{R-UM}^{\alpha,\beta}$)**: The mechanism of innovation is particularly commendable. Instead of simply adding a new top-level loss, the authors redesigned the inner-loop loss of the min-max game (Eq. 17). The "fake model" $f$ is forced to optimize on both $p_0^\theta$ (generated data) and $p_0^*$ (real data) during its $\max_f$ step. This allows $f$ to serve as a "bridge," passing the supervisory signal from the real data to the student generator $G_\theta$ in a natural, non-adversarial way.
3. **Strong Theoretical Unification (UID)**: Prior to RealUID, the paper's UID framework is a valuable theoretical contribution in its own right. It unifies seemingly separate works (SiD, FGM) under a single min-max perspective using a clear and general mathematical derivation (the linearization technique). It clarifies the underlying game structure: the "fake model" $f$ (the $\max_f$ player) tries to approximate the "theoretical student function" $f^\theta$, while the generator $G_\theta$ (the $\min_\theta$ player) tries to minimize the distance between $f^\theta$ and the teacher $f^*$.
4. **Excellent Empirical Efficiency**: The experimental results strongly support the practical utility of this new method. RealUID not only achieves strong FID scores but also demonstrates significantly faster convergence (approx. 3x) compared to the data-free baseline (Fig. 2), proving that this GAN-free supervision is both effective and efficient。

**Weaknesses:**

1. **Severely Limited Experimental Scope (Main Flaw)**: All experiments are conducted exclusively on the CIFAR-10 (32x32) dataset. While resource constraints are mentioned, this leaves the scalability of RealUID to higher resolutions (e.g., 256x256 or 512x512) completely unknown. The need for distillation is most acute for these larger, slower models.
2. **Not State-of-the-Art (SOTA) Performance**： While the results are strong, the best-reported FID (e.g., 1.91 for conditional generation) still lags behind SOTA adversarial distillation methods (e.g., $SiD^2A$ @ 1.39). The authors hypothesize this is due to their lightweight architecture and teacher model, but this is unproven. It is unclear if the performance gap is due to the architecture or an inherent ceiling of the RealUID method itself compared to adversarial approaches.
3. **New Hyperparameter Sensitivity**: RealUID avoids GAN instability but introduces new hyperparameters $\alpha$ and $\beta$. The fine-tuning ablation (Table 6) shows high sensitivity to these, with many configurations failing to converge (marked with "-"). This may offset the claimed simplicity, as tuning $\alpha$ and $\beta$ could be as difficult as tuning a GAN.

**Questions:**

1. **On Scalability (W1)**: Can the authors provide even preliminary results on a single higher-resolution dataset (e.g., FFHQ-64 or CelebA-256)? How does RealUID converge, and are the $\alpha, \beta$ hyperparameters stable across different resolutions?
2. **On $K=5$ (W3)**: The experimental details state the fake model $f$ is updated 5 times for every 1 generator update ($K=5$). How critical is this ratio? Does the RealUID framework become unstable at $K=1$? This seems to hide a 5x computational cost for the fake model's training loop relative to the generator's.
3. **On $\mathcal{L}_{R-UM}^{\alpha,\beta}$ Coefficient Design (S2)**: The specific coefficients $\frac{\beta}{\alpha}$ and $\frac{1-\beta}{1-\alpha}$ in Eq. 17 are clearly deliberate. The paper claims this is to ensure $f=f^\*$ is the optimal solution when $p_0^\theta = p_0^*$.
Is this specific mathematical form the only one that satisfies this property? Would the framework fail if simpler, unscaled $L_2$ losses were used (e.g., setting $\beta/\alpha = 1$ and $(1-\beta)/(1-\alpha) = 1$)? This seems highly relevant given the hyperparameter sensitivity seen in (W3).
4. **On "Correcting" the Teacher (S2)**: A potential benefit of RealUID is that the fake model $f$ sees both the teacher $f^\*$ and the real data $p_0^\*$. If the teacher $f^\*$ is flawed (e.g., suffers from mode collapse) but $p_0^\*$ is complete, $f$ faces a conflict during its training: should it match $f^\*$ or the function implied by $p_0^\*$?  Can RealUID truly "correct" a flawed teacher, or would this conflict lead to training instability?

---

> ### Author Response · Authors · 2025-11-22
>
> Dear Reviewer oamB, thank you for your comments. Please find below the answers to your questions and comments.
>
> **W1. Experimental Scope and Scalability**
>
> To test the scalability of our RealUID, we conducted the same experiments on the larger and more complex CelebA 64x64 dataset and architecture. Due to limited time, we ran all experiments only for 400k iterations. They all can achieve better metrics, and we can train them longer if needed. The results are presented in the table below and in the Appendix E.2 in the revision.
>
> | α\β| 0.94 | 0.96 | 0.98 | 1.0|
> |---|---|--|--|--|
> | 0.94 | 2.60 | 1.93 | 2.13  | 2.53 |
> | 0.96| **1.70** | 2.77 | 2.00  | 2.47 |
> | 0.98| 2.16  | 2.04  | 2.62 | 2.42 |
> | 1.0 | 2.96  | 2.48  | 2.23 | 2.62  |
>
> Once again, our RealUID significantly outperforms its data-free counterpart. __The best parameters $\alpha, \beta$ remain the same for both Cifar10 and CelebA, demonstrating stability over various setups.__
>
> **W2. Not SOTA Performance**
>
> The SOTA performance of models is usually achieved through larger architectures, numerous practical heuristics, careful tuning of hyperparameters and significantly more compute. For example, with our resources, SiDA needs 15 days to achieve our metrics (we ran it for only 2 days) and twice that to achieve its SOTA performance.
>
> In our methodological work, the main goal is to demonstrate the universal theory behind all distillation methods and to test a new approach to incorporate real data. Moreover, we do not have enough resources to run SOTA experiments that will achieve even our own metrics within a reasonable time. That is why we did not try to achieve the SOTA results.
>
> Nevertheless, for Flow Matching models, our UID achieves the SOTA performance with lightweight architecture, outperforming the FGM approach [1] (generalization of SiD for Flow Matching models) with heavy EDM architecture.
>
> [1] Huang, Z., Geng, Z., Luo, W., \& Qi, G. J. (2024). Flow generator matching. arXiv preprint arXiv:2410.19310.
>
> **W2. Performance gap is due to the architecture or an inherent ceiling of the RealUID?**
>
> We use the same architectures in all our experiments, testing our real data incorporation approach against GANs. Therefore, the comparison between the two approaches is fair.
>
> **W3. RealUID avoids GAN instability but introduces new hyperparameters.**
>
> Following your question, we explained more thoroughly how to choose good $\alpha,\beta$ in the revision Section 3.4.
>
> Our Lemma 2 shows that, instead of $\alpha$ and $\beta$, actually factors $\alpha$ and $\beta/\alpha$ determine the balance between generated and real data in the minimized distance:
>
> $$\mathcal{L}^{\alpha, \beta}\_{\text{R-UID}}(f, p\_0^\theta)  = \mathbb{E}\_{t \sim [0,T]}\mathbb{E}\_{x\_t \sim p\_t^\*}[ \frac{||  \frac{\beta}{\alpha}\cdot [  p\_t^\*(x\_t^\*)f^\*\_t(x\_t^\*) -   p\_t^\theta(x\_t^\*) f^\theta\_t(x\_t^\*)] + (p\_t^\theta(x\_t^\*) - p\_t^*(x\_t^\*)) \cdot f^\*\_t(x\_t^\*)||^2}{p\_t^\*(x\_t^\*)((1-\alpha)p\_t^\*(x\_t^\*) + \alpha p\_t^\theta(x\_t^\*))/\alpha^2 }].$$
>
> Furthermore, coefficient $\alpha$ only sets the general scaling of the distance, while *$\frac{\beta}{\alpha}$ plays the most important role*, as it determines the relation between $f^\theta_t$ and $f^*_t$ inside the distance.
>
> We provide theoretical analysis of this distance in Section 3.4. It states that the good $\alpha, \beta \in (0,1]$ can be chosen by first finding $\frac{\beta}{\alpha} \neq 1$, as it has the largest impact, and then adjusting $\alpha < 1$. Both $\frac{\beta}{\alpha}$ and $\alpha$ should be close to $1.$
>
> Hence, we reorganized the results table in terms of $\alpha$ and $\beta/\alpha$ to confirm that
>
> 1) $\beta/\alpha$ brings the largest impact (and can even be adjusted solely with a fixed $\alpha$),
>
> 2)  small non-zero values $\beta/\alpha -1$ consistently lead to a better performance, while  large values lead to a worse one.
>
> | Generation  | α \ (α/β) | 1.06  | 1.04  | 1.02  | 1.0   | 0.98  | 0.96  | 0.94  |
> |----------------|-----------|-------|-------|-------|-------|-------|-------|-------|
> | **Unconditional** | **0.94** | -     | -     | -     | 2.66  | **2.28** | 2.58  | 2.98  |
> |  | **0.96** | -  | -  | 2.37  | 2.58  | 2.29  | 2.65  | - |
> | | **0.98** | - | 2.97  | 2.33  | 2.62  | 2.38  | - | - |
> | | **1.0**  | 5.81  | 4.51  | 3.29  | 2.58 | - | - | - |
> | **Conditional**  | **0.94** | -| - | - | 2.35  | 2.19  | 2.25  | 2.47  |
> |  | **0.96** | - | -| 2.09  | 2.32  | 2.13  | 2.27  | -  |
> |   | **0.98** | -| 2.34  | **2.02** | 2.26  | 2.05  | -     | - |
> |   | **1.0**  | 4.32  | 3.27  | 2.43  | 2.21  | -  | - | -|
>
> In comparison, a GAN has numerous parameters to tune: a generator loss factor over a large range, a discriminator loss factor over a large range, a choice of discriminator architecture, and the range of timesteps used in the discriminator. *This makes our method significantly simpler to tune*, as it has fewer parameters with a clear interpretation.

---

> > ### Author Response · Authors · 2025-11-22
> >
> > **W3. The fine-tuning ablation (Table 6) shows high sensitivity to these.**
> >
> > In fine-tuning experiments, we model real-world scenarios, when only the best generator and teacher models are usually publicly available. Hence, we use EMA generator and the teacher model as the initializer for the fake model, rather than the models we pre-trained. These models are not well-aligned which causes the disconvergence, not the instability of the proposed RealUID. For example, without real data, the model fails to converge at all, and adding real data is the only way to improve the quality. For the GAN-based fine-tuning, the method also disconverges for the most parameter configurations.
> >
> > **Q1. Is this specific mathematical form the only one that satisfies this property?**
> >
> >
> > No, it is not. For example, one may replace MSE in loss in our inverse problem with any Bregman divergence $D(f_t^\theta, f_t) = \Psi(f_t^\theta) - \Psi(f_t) - \langle \nabla \Psi(f_t), f_t^\theta -f_t \rangle$ between model $f_t$ and data $f^\theta_t$. This is because the Bregman divergence can be used as a substitute to MSE in the forward matching problem as shown in [2]. However, other divergences but Euclidean ($\Psi(\cdot)=\frac{1}{2}||\cdot||^{2}$) are not widely used, and optimum analysis with real data for them might be complicated.
> >
> > [2] Holderrieth, P., Havasi, M., Yim, J., Shaul, N., Gat, I., Jaakkola, T., ... \& Lipman, Y. (2024). Generator matching: Generative modeling with arbitrary markov processes. arXiv preprint arXiv:2410.20587.
> >
> > **Q2.
> > Would the framework fail if simpler, unscaled  losses were used (e.g., setting $\alpha = \beta$ and $1- \alpha = 1-\beta$)?**
> >
> > We can set $\alpha = \beta$ in our loss, but this leads to inferior performance both in theory (see the discussion after Theorem 2) and in practice (Tables 1, 7). Thus, two distinct coefficients are necessary. However, as we showed before, it makes more sense to find good value $\frac{\beta}{\alpha} \neq 1$ with some fixed $\alpha$ first, and it can be already good enough.
> >
> >
> > **Q3. $K$ stability**
> >
> > To answer your question, we have conducted additional experiments with $K = 1,2,3,4, 5$ and the same number of total iterations on the best RealUID, and the results are presented below.
> >
> > | K             | 5       | 4       | 3       | 2       | 1       |
> > |---------------|---------|---------|---------|---------|---------|
> > | Min EMA FID   | 2.28 | 2.32   | 2.21   | 2.95   | 4.00   |
> >
> > They show that too small $K$ lead to worse results. Remarkably, $K=3$ achieves a bit better FID than original $K=5$.
> >
> > Large $K$ should not be considered as a wasted computation, as each update of the fake model refines the generator's gradients and better captures the real data. We believe that model's stability with respect to small $K$ can be achieved via architectures and loss modifications, e.g., loss normalization from SiD.
> >
> >
> >
> >
> > **Q4. Can RealUID truly "correct" a flawed teacher?**
> >
> >
> > Yes, our RealUID can correct teacher's errors. Following your question, we have conducted additional theoretical analysis in the revision Appendix A.1.3 which shows that *RealUID actually corrects the teacher's errors*. We derived the distance formula (23) minimized by RealUID and showed that its generator optimum can be guided toward the real data and away from the flawed teacher with the proper choice of $\alpha$ and $\beta/\alpha$.
> >
> >
> > **Concluding remarks.** We would be grateful if you could let us know if our explanations have been satisfactory. If so, we kindly ask that you consider increasing your rating. We are also open to discussing any other questions you may have.

---

> > > ### Comment · Reviewer_oamB · 2025-11-23
> > >
> > > Thanks the authors for the detailed explanation. My concern and questions are fully answered. As a distillation method introducing real data without GAN is interesting and the proving is solid. I would like to raise my score.

---

### Author Response · Authors · 2025-11-22

Dear Reviewers,

In this revision, we have addressed your comments by adding the following new materials in the revision, which are highlighted in blue text:

1) **[oamB, ha8b, WywQ];** **Additional experiments on the CelebA-64x64 dataset** in Appendix E.2, demonstrating the stability and strong performance of our RealUID across different setups.

2) **[oamB];** **New theoretical analysis** in Appendix A.1.3 dedicated to the ability of our RealUID to correct the teacher's errors.


3) **[ha8b, WywQ];** **New comparisons** with baseline consistency models in Table 1.

4) **[oamB, ha8b, WywQ];** **An expanded guide** on selecting the parameters $\alpha$ and $\beta$ in Section 3.4.

---

### Author Response · Authors · 2025-12-03
**Final summary**

In our work, we unify previously known distillation methods for diffusion and flow matching models under a universal, simple yet powerful theory, and expand it further to other types of matching models such as bridge matching and stochastic interpolants. Using our proof techniques, we propose a novel way to naturally incorporate real data into the distillation loss directly, avoiding any adversarial losses and extra GAN models, while achieving comparable or better results in practice.

*All Reviewers acknowledge the importance, clarity, expandability, and universality of our theory. The data incorporation approach is also clever, sound, and effective.*

Below, we summarize the question-response discussion:

| Question | Reviewers | Answer |
| :--- | :--- | :--- |
| **Scalability to larger datasets and architectures** | **[oamB, ha8b, WywQ]** | We have added experiments on the CelebA 64×64 dataset (Appendix E.2), in which our RealUID outperforms its data-free counterpart while exhibiting stability w.r.t. new hyperparameters across different setups. |
| **Sensitivity to new hyperparameters** | **[oamB, ha8b, WywQ]** | We have provided a more detailed explanation in both theory (Section 3.4) and practice for selecting good parameters. This includes switching from $\alpha, \beta$ to $\beta/\alpha, \alpha$ which have more interpretable effect. |
| **Diversity metrics** | **[ha8b]** | We calculated Recall, Precision, and Inception Score for all models. Our RealUID consistently yields the best results. |
| **SOTA performance** | **[oamB, ha8b, WywQ]** | We emphasized that the main goal of our work is not to achieve SOTA results, as these are typically achieved via extensive practical heuristics and computational resources, which we lack. Nevertheless, in our experiments, all methods are fairly compared under a single pipeline, and among Flow Matching models, we do achieve SOTA performance.|
| **New consistency baselines** | **[ha8b, WywQ]** | We discussed the differences between our method and consistency models, adding new baselines for a more comprehensive comparison. |
| **Theory-related questions** | **[pZHj, WywQ]** | We have addressed the theory-related questions; please see the discussion for details. |
| **Experiments with diffusion models and large architectures** | **[WywQ]** | We believe these experiments are rather technical than conceptual and require significant preparation to execute properly. Given the limited rebuttal time and resources, we decided to focus on the less-studied Flow Matching models. |

*As a result of the discussion, Reviewers pZHj and oamB gave us an 8 (the first from the initial review and the second raised the score shortly after the rebuttal began). Reviewer ha8b intended to raise the score from 6, and Reviewer WywQ did not respond.*

---

### Meta-Review · Area_Chair_ta1W · 2026-01-08

**Summary:**

This paper proposes RealUID, a unified distillation framework for diffusion- or flow-based models. The approach utilizes real-data supervision without utilizing adversarial training. The major contribution is the unification of existing data-free distillation methods (SiD, FGM) and introduces a new loss function to utilize the real data.

**Reviewer Concerns:**

Reviewers raised the following issues initially

1. Larger datasets (oamB, ha8b, WywQ): The reviewers mentioned that the experiments were limited to CIFAR-10. The reviewers asked ImageNet of AFHQ benchmark results.
2. Sensitivity (oamB, ha8b, WywQ): The reviewers asked about the stability of the approach and the theoretical justification of the method.
3. Evaluation metrics (ha8b): The reviewer requested diversity metrics to check the possible mode collapse problem.
4. Additional comparison (ha8b, WywQ): The additional baselines, such as consistency models, needed to be compared

**Reviewer Scores:**

This paper initially received 6(oamB), 6(ha8b), 4(WywQ), 8(pZHj).

Regarding concern 1, the authors provided experiments on CelebA 64x64. The reviewer oamB explicitly stated the major concerns were resolved. Regarding 2, the authors provided a new theoretical analysis \frac{\beta}{\alpha} is the critical factor and provided a guide for selection. Regarding 3, the authors provided inception score, precision, and recall, which show RealUID yields better results. Regarding 4, the authors added an experiment with consistency models that explained the difference in data utilization.

Overall, AC confirms that most of the issues were properly addressed in the rebuttal. In light of the unified framework for the various distillation frameworks, AC notes the value of this paper. AC notes a few recommendations: (1) Please try to add complete experiments with further training steps, since the rebuttal provided early experiments due to the time limit. (2) Consider changing the title of the paper, since terms such as "(No GANs)" are not a common choice. A quick suggestion is "Universal Inverse Distillation for Matching Models without Adversarial Training". AC hopes the paper will be further polished to have a greater impact in the field.

---

### Decision · Program_Chairs · 2026-01-26

Accept (Oral)